# COME: Continuous Motion Diffusion for High Quality Text to Motion Generation

## Abstract

Text-to-Motion generation aims to synthesize realistic 3D human motion from natural language descriptions. Although continuous diffusion models naturally align with the temporal and spatial continuity of motion, they have underperformed discrete token-based approaches in generation quality. However, as T2M tasks evolve to include motion editing, personalization, and multimodal control, they increasingly demand fine-grained semantics, compositionality, and diverse sampling—capabilities better supported by continuous frameworks. Motivated by these real-world demands and the inherent continuity of motion, we revisit continuous diffusion modeling and identify two core limitations: (1) motion representations are often crowded and poorly separable, which increases the difficulty of generation and denoising; (2) suboptimal generative modeling that further degrades generation quality. To address these challenges, we propose COME, a continuous diffusion framework that enhances both motion representation and generative modeling. COME comprises two main components: the Motion Contrastive Masked Autoencoder (MoCMAE) and the Cross-Condition Diffusion Transformer (ccDIT). MoCMAE employs an asymmetric hybrid architecture that integrates Masked Motion Modeling to extract key spatio-temporal features and Contrastive Learning to further enhance feature discriminability, thereby providing an expressive latent space. Meanwhile, ccDIT incorporates ccDIT block for global and fine-grained semantic comprehension and then utilizes Stable-Min-SNR-$\gamma$ to address training-inference inconsistencies and the conflicts across different timesteps, thus boosting generation quality. Extensive experiments show that COME achieves SOTA performance while improving inference and training efficiency, highlighting the effectiveness of our approach.

## 1 Introduction

Text-to-motion (T2M) generation aims to produce realistic and semantically coherent human motion sequences from natural language descriptions. This task is foundational for a variety of applications, including embodied AI, animation, and human–robot interaction. Recent progress in T2M has been largely driven by two distinct paradigms: discrete token-based generation and continuous diffusion-based generation.

In the discrete paradigm, motion sequences are first quantized into tokens via vector-quantized autoencoders (e.g., VQ-VAE (Van Den Oord et al., 2017)), and then modeled using autoregressive or masked transformers (Zhang et al., 2023a; Guo et al., 2024). These approaches have achieved impressive results in terms of generation quality and inference speed, and currently dominate standard T2M benchmarks. In contrast, continuous approaches—particularly diffusion models (Tevet et al., 2022; Chen et al., 2023)—directly generate 3D human motions or latent motion representations. While conceptually more natural for capturing the temporal and spatial continuity of human motion, these methods have underperformed discrete models.

However, the evolving demands of T2M tasks are beginning to expose the limitations of discrete paradigms and underscore the untapped potential of continuous diffusion models. Emerging applications—such as motion editing (Athanasiou et al., 2024), style transfer (Zhong et al., 2024; Raab et al., 2024), personalization (Kim et al., 2025), and multimodal conditioning (e.g., sketch, trajectory, or video prompts) (Wang et al., 2025; Yu et al., 2025)—increasingly require flexible semantic control, compositional generalization, and diverse sampling. These requirements are inherently more

compatible with the stochastic and continuous nature of diffusion-based models, which offer greater adaptability than discrete token-based approaches (Athanasiou et al., 2024; Kim et al., 2025).

Motivated by both real-world demands and the natural continuity of human motion, we revisit continuous diffusion frameworks for motion generation and identify two key limitations: First, weak motion representations: latent features from continuous models (e.g., VAEs) are often poorly separated and entangled, making it difficult to distinguish between different motions. Compared to discrete models like VQ-VAE or RVQ-VAE (see Tab. 7, Figs. 6, 5), this lack of inter-sample separability hinders denoising in diffusion models and often leads to low-quality motion generation. Second, suboptimal diffusion training strategies, which include training–inference mismatch (Lin et al., 2024), where training does not reach pure noise, but inference starts from pure noise, and gradient conflicts across timesteps (Hang et al., 2023), which hinder convergence stability.

To address these challenges, we propose COME, a fully continuous diffusion framework for T2M generation. Our method is designed with two core components: MoCMAE, a masked contrastive motion autoencoder that learns a high-quality latent representation. The model uses an asymmetric encoder–decoder architecture: the encoder employs CNNs for local pattern extraction and transformers for global spatiotemporal reasoning, while the decoder remains lightweight (CNN-only), promoting the encoder to produce expressive and disentangled features. Masked motion modeling enhances spatial-temporal abstraction, and contrastive learning improves inter-sample discriminability. Overall, this design improves feature quality without adding decoding burdens to the generation process. ccDIT, a diffusion transformer tailored for text-conditioned motion generation. The main design principle is to fully capture both global sentence semantics and fine-grained word semantics. We incorporate sentence-level features via AdaLN-Zero(Peebles & Xie, 2023) and integrate word-level features using cross-attention, enabling both global and fine-grained semantic control. Additionally, we employ skip connections to further enhance training stability. To solve the training–inference mismatch issue, we unify Zero-SNR (Lin et al., 2024) to enforce noise to pure noise, while addressing the conflicts across timesteps using Min-SNR-$\gamma$ (Hang et al., 2023) to rescale loss weights of different timesteps. These two strategies conflict with each other, and we propose a simple improvement, Stable-Min-SNR-$\gamma$, to reconcile them.

In summary, we introduce COME, a continuous diffusion framework that revisits T2M generation through the lens of representation learning and modeling design. (1) To address the limitations of conventional continuous features, we propose asymmetric MoCMAE to learn expressive and well-separated motion representations without increasing decoding cost. (2) We develop ccDIT, a diffusion transformer that integrates sentence- and word-level semantics, and propose Stable-Min-SNR-$\gamma$ to reconcile training-inference mismatch and gradient instability. (3) Extensive comparative, ablation, and generalization studies show that COME achieves SOTA performance while improving inference and training efficiency, highlighting the effectiveness and generalizability of our approach.

## 2 RELATED WORK

**Continuous T2M Generation.** The continuous paradigm directly generates 3D human motions or their latent representations from textual descriptions using frameworks such as VAEs (Kingma & Welling, 2013), GANs (Goodfellow et al., 2014), and diffusion models (Zhang et al., 2024; Dabral et al., 2023; Yuan et al., 2023; Kim et al., 2023; Wang et al., 2023; Zhang et al., 2023c; Jin et al., 2024; Huang et al., 2024; Hong et al., 2025). Among these, diffusion models have emerged as the dominant approach. MDM (Tevet et al., 2022) first introduced them into T2M generation, while MLD (Chen et al., 2023) improved the paradigm by applying latent diffusion, compressing motion into a latent space to enhance quality and efficiency. Building on these foundations, subsequent works such as ReMoDiffuse (Zhang et al., 2023b) (retrieval-augmented generation), MotionMamba (Mamba)(Zhang et al., 2025b), and MotionLCM(Dai et al., 2025) (Latent Consistency Model) have further advanced the field. Beyond text-to-motion generation, recent efforts have focused on broader aspects of the task. These include dataset refinement with fine-grained annotations (FineMoGen (Zhang et al., 2023c), Fg-T2M (Wang et al., 2023), GraphMotion (Jin et al., 2024)), motion control(OmniControl (Xie et al., 2024),PriorMDM (Shafir et al., 2024),GMD (Karunratanakul et al., 2023)), motion editing (MotionFix (Athanasiou et al., 2024)), and motion style transfer (Smoodi (Zhong et al., 2024), Monkey (Raab et al., 2024)).

**Discrete T2M Generation.** The discrete paradigm operates in two stages: discretization and generation. Continuous motion data is first tokenized using tokenizers like VQ-VAE (Van Den Oord

et al., 2017). Subsequently, tokens are predicted using either AR (Guo et al., 2022b; Zhang et al., 2023a) or NAR (Guo et al., 2024; Pinyoanuntapong et al., 2024; Yuan et al., 2025). By transforming the regression problem into a classification task, this approach simplifies the generation process and leverages advancements in natural language processing, resulting in improved performance (Jiang et al., 2023; Zhou et al., 2024). TM2T was the first to introduce the discrete paradigm. T2M-GPT further optimized VQ-VAE training with techniques such as code reset and EMA updates, boosting performance. MoMask employed residual vector quantization (RVQ) to reduce quantization errors, combined with two NAR generation models, achieving improvements in both quality and speed.

## 3 PREMALIERAIES

Diffusion models consist of two processes: a forward noising process and a reverse denoising process. We denote the distribution of training data as $p(\mathbf{x}_0)$. The forward process is a Gaussian transition, gradually adding noise with different scales to a real data point $\mathbf{x}_0 \sim p(\mathbf{x}_0)$ to obtain a series of noisy latent variables $\{\mathbf{x}_1, \mathbf{x}_2, \ldots, \mathbf{x}_T\}$:

$$q(\mathbf{x}_t|\mathbf{x}_0) = \mathcal{N}(\mathbf{x}_t; \alpha_t\mathbf{x}_0, \sigma_t^2\mathbf{I}) \tag{1}$$

$$\mathbf{x}_t = \alpha_t\mathbf{x}_0 + \sigma_t\boldsymbol{\epsilon} \tag{2}$$

where $\boldsymbol{\epsilon}$ is the noise sampled from Gaussian distribution $\mathcal{N}(0, \mathbf{I})$. The noise schedule $\sigma_t$ denotes the magnitude of noise added to the clean data at $t$ timestep. It increases monotonically with $t$. $\alpha_t = \sqrt{1 - \sigma_t^2}$. Signal-to-noise ratio (SNR) can be calculated as: $\mathrm{SNR}(t) := \frac{\alpha_t^2}{\sigma_t^2}$

The reverse process is parameterized by another Gaussian transition, gradually denoises the latent variables and restores the real data $\mathbf{x}_0$ from a Gaussian noise:

$$p_\theta(\mathbf{x}_{t-1}|\mathbf{x}_t) = \mathcal{N}(\mathbf{x}_{t-1}; \hat{\mu}_\theta(\mathbf{x}_t), \hat{\Sigma}_\theta(\mathbf{x}_t)). \tag{3}$$

$\hat{\mu}_\theta$ and $\hat{\Sigma}_\theta$ are predicted statistics.

Common optimization objectives include three types: the noise $\epsilon$ Ho et al. (2020), the original sample $\mathbf{x}_0$ Ramesh et al. (2022), and $\mathbf{v}$-prediction Salimans & Ho (2022). The velocity $\mathbf{v}$ is defined as a linear combination of $\mathbf{x}_0$ and $\boldsymbol{\epsilon}$: $\mathbf{v} = \alpha_t\boldsymbol{\epsilon} - \sigma_t\mathbf{x}_0$. This $\mathbf{v}$-prediction unifies noise and data prediction, stabilizing training and enhancing generative quality.

Latent diffusion models (LDMs) Rombach et al. (2022); Peebles & Xie (2023) reduce the high computational cost by conducting the diffusion process in the latent space. First, the autoencoder is utilized to compress the raw data to latent space, then the diffusion models work on the latent space.

## 4 METHOD

Our goal is to efficiently generate high-quality and diverse human motions, denoted as $\mathbf{X}_{1:N}$ with a sequence length of $N$, conditioned on text inputs $c$. As illustrated in Fig. 1, we present COME, an innovative framework consisting of two primary components: a novel continuous motion tokenizer (MoCMAE), which constructs a representative latent space (Sec. 4.1), and a conditional motion diffusion model (ccDIT) that generates motions within this latent space (Sec. 4.2).

### 4.1 MOCMAE

We systematically improve both the model architecture and training strategy to enhance motion representations for motion generation. We propose MoCMAE, a masked contrastive motion autoencoder that learns a structured and expressive latent space tailored for generative modeling. Its encoder combines CNNs for local motion extraction and Transformer blocks for long-range spatiotemporal modeling, while the decoder is lightweight and CNN-based. This asymmetric design enables informative encoding without increasing inference cost. In addition, we introduce two complementary discriminative representation technologies within the generative pipeline. Masked motion modeling (MMM) randomly masks input frames, encouraging the encoder to capture essential spatiotemporal patterns. Contrastive learning (CL) aligns latent codes from full and masked sequences, improving inter-sample separability and stabilizing representations, directly benefiting motion generation. MoCMAE uses a dual-branch pipeline where one branch processes the full motion and the other the masked input, sharing a CNN decoder. We also explore a VAE variant, MoCMVAE, with

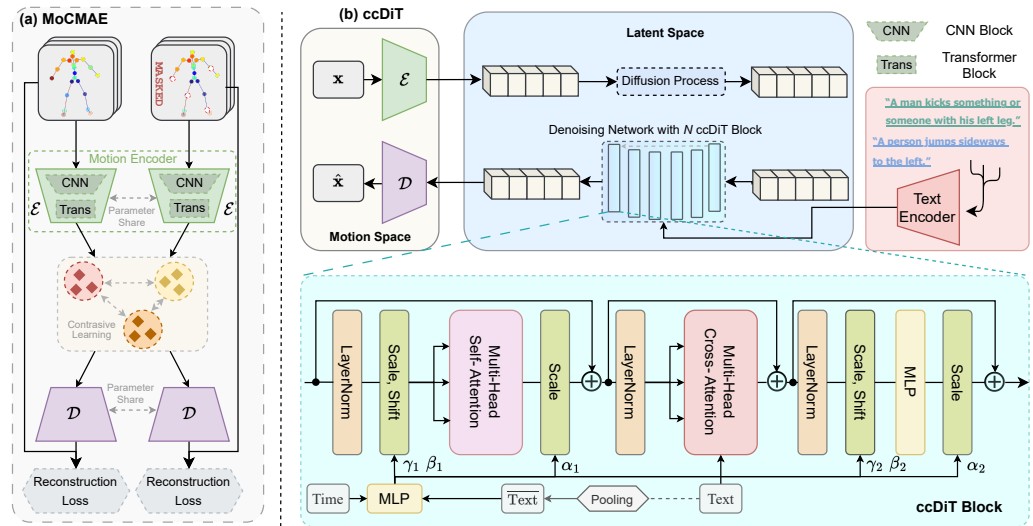

Figure 1: Overview of COME. It consists of: (a) MoCMAE, an asymmetric hybrid model to extract high-quality motion features without introducing additional decoding overhead; and (b) ccDIT, a conditional diffusion model that captures both global sentence-level semantics and fine-grained word-level cues to generate high-quality motion sequences.

a KL penalty toward a standard Gaussian prior. The detailed architecture is provided in Appendix Fig. 15.

## 4.2 ccDIT

Our core goal is to revive continuous motion diffusion models for text-to-motion generation under standard experimental settings, enhancing generation quality, training efficiency, and inference speed. Inspired by advances in text-to-image and discrete T2M models, we systematically analyzed architectural modifications and retained only the most effective design choices. To enable effective semantic conditioning, ccDIT integrates two complementary mechanisms. First, it adopts AdaLN-Zero Peebles & Xie (2023) to inject sentence-level embeddings and diffusion timestep signals into each transformer block, providing global semantic alignment and stage-aware modulation during denoising. Second, it incorporates cross-attention layers at each block to attend to word-level language features, facilitating fine-grained temporal grounding aligned with individual tokens. To improve training stability and representation consistency, ccDIT further incorporates skip connections as in U-DiT Tian et al. (2024), which enhance gradient flow and accelerate convergence. Together, these components enable ccDIT to generate temporally consistent and semantically rich motions conditioned on complex natural language, while maintaining scalability and training efficiency.

## 4.3 TRAINING STRATEGY

**MoCMAE.** MoCMAE is trained with a joint objective combining reconstruction loss $\mathcal{L}_{\text{rec}}$ and contrastive loss $\mathcal{L}_{\text{contrast}}$, ensuring accurate motion reconstruction and robust latent alignment. The reconstruction loss includes a smooth L1 term on motion sequences and a velocity regularization term to promote temporal coherence:

$$\mathcal{L}_{\text{rec}} = \mathcal{L}_1^{\text{smooth}}(\mathbf{X}, \mathbf{X}_{\text{rec}}) + \alpha \, \mathcal{L}_1^{\text{smooth}}(V(\mathbf{X}), V(\mathbf{X}_{\text{rec}})), \tag{4}$$

where $\alpha$ balances the position and velocity terms, and $V(\cdot)$ computes frame-wise velocity.

The contrastive loss aligns masked and full embeddings by maximizing similarity for matched pairs $(\bar{Z}_m^i, \bar{Z}_f^i)$ while minimizing it for unmatched ones:

$$\mathcal{L}_{\text{contrast}} = -\frac{1}{N} \sum_{i=1}^{N} \log \frac{\exp(\text{sim}(\bar{Z}_m^i, \bar{Z}_f^i)/\tau)}{\sum_{j=1}^{N} \exp(\text{sim}(\bar{Z}_m^i, \bar{Z}_f^j)/\tau)}, \tag{5}$$

where $\tau$ is a temperature hyperparameter. The full MoCMAE objective is:

$$\mathcal{L}_{\text{MoCMAE}} = \mathcal{L}_{\text{rec}} + \lambda \, \mathcal{L}_{\text{contrast}}, \tag{6}$$

with $\lambda$ weighting the contrastive loss.

MoCMVAE extends MoCMAE by adding a KL-penalty to encourage a structured latent distribution:

$$\mathcal{L}_{\text{MoCMVAE}} = \mathcal{L}_{\text{MoCMAE}} + \eta\,\mathcal{L}_{\text{KL}}, \tag{7}$$

where $\eta$ controls the strength of regularization.

**ccDIT.** Training diffusion-based T2M models presents two key challenges. First, a mismatch exists between training and inference, as inference starts from pure noise while training rarely encounters such conditions. To mitigate this discrepancy, we adopt the Zero-SNR strategy Lin et al. (2024), which enforces signal-to-noise ratio (SNR) to be zero at the final timestep during training.

Second, conflicting loss magnitudes across timesteps can hamper convergence Hang et al. (2023), as training overly emphasizes denoising at specific noise levels. To address this, we incorporate the Min-SNR-$\gamma$ strategy, which reweights the loss at each timestep based on clamped SNR:

$$w_t = \frac{\min\{\text{SNR}(t), \gamma\}}{\text{SNR}(t)}, \tag{8}$$

where $\gamma$ denotes the maximum allowed SNR. However, this formulation becomes undefined when $\text{SNR}(t) = 0$. To ensure numerical stability, we propose a stabilized variant, Stable-Min-SNR-$\gamma$:

$$w_t = \frac{\min\{\text{SNR}(t), \gamma\}}{\text{SNR}(t) + \phi}, \tag{9}$$

where $\phi > 0$ is a small constant that prevents division by zero.

To further enhance generation quality, we employ Classifier-Free Guidance (CFG) Ho & Salimans (2022) by training the model with both conditioned and unconditioned inputs (randomly setting $c = \emptyset$ for 10% of training samples). The final ccDIT objective uses an SNR-weighted MSE loss:

$$\mathcal{L}_{\text{ccDIT}} = w_t \cdot \mathcal{L}_{\text{mse}}. \tag{10}$$

### 4.4 Sampling

The generation process begins with pure noise, and the model iteratively denoises under text conditions to produce the motion. To balance diversity and quality, we employ CFG to combine both conditional and unconditional outputs. Furthermore, we investigate various sampling schedulers Song et al. (2020); Lu et al. (2022) to optimize both inference efficiency and generation quality.

## 5 Experiments

**Datasets.** We conduct experiments on two standard datasets: HumanML3D Guo et al. (2022a) and KIT-ML Plappert et al. (2016). HumanML3D includes 14,616 motion sequences and 44,970 textual descriptions from AMASS Mahmood et al. (2019) and HumanAct12 Guo et al. (2020). KIT-ML contains 3,911 motion sequences and 6,278 descriptions.

**Evaluation Metrics.** We evaluate the performance using standard metrics proposed by T2M Guo et al. (2022a). Fréchet Inception Distance (FID) measures the distributional difference between the high-level features of generated and real motions, reflecting the overall motion quality. R-Precision and Multimodal Distance (MM-Dist) assess the semantic alignment between input text and generated motions, with R-Precision reporting Top-1, Top-2, and Top-3 accuracies based on retrieval tasks. Diversity is calculated as the average Euclidean distance between randomly sampled pairs of 300 motion sequences, indicating the variability in generated motions. Multimodality (MModality) evaluates the variance of motions generated from the same text prompt.

**Implementation Details.** All models are implemented in PyTorch Paszke et al. (2019). MoCMAE adopts an asymmetric architecture with ResBlocks He et al. (2016) (downscale/upscale factor of 4) and 4 transformer Vaswani (2017) layers in the encoder for enhanced spatio-temporal modeling. We use a mask ratio of 0.5 and a contrastive loss weight of 0.1. The model is trained for 500 epochs using the AdamW optimizer and a cosine scheduler, with a batch size of 64 on both datasets. ccDIT is composed of 10 transformer layers with a latent dimension of 512. Language features are extracted using CLIP-ViT-B Radford et al. (2021). We apply a linear warm-up to a peak learning rate of 1e-4 over 100 iterations, followed by cosine decay. Training uses AdamW with EMA (rate

| Paradigms | Methods | FID↓ | R Precision↑ | | | MM-Dist↓ | Diversity→ | MModality↑ |
|---|---|---|---|---|---|---|---|---|
| | | | Top 1 | Top 2 | Top 3 | | | |
| | Ground Truth | $0.002^{\pm.000}$ | $0.511^{\pm.003}$ | $0.703^{\pm.003}$ | $0.797^{\pm.002}$ | $2.974^{\pm.008}$ | $9.503^{\pm.065}$ | - |
| Continuous | Seq2Seq Plappert et al. (2018) | $11.75^{\pm.035}$ | $0.180^{\pm.002}$ | $0.300^{\pm.002}$ | $0.396^{\pm.002}$ | $5.529^{\pm.007}$ | $6.223^{\pm.061}$ | - |
| | JL2P Lucas et al. (2022) | $11.02^{\pm.046}$ | $0.246^{\pm.002}$ | $0.387^{\pm.002}$ | $0.486^{\pm.002}$ | $5.296^{\pm.008}$ | $7.676^{\pm.058}$ | - |
| | T2G Bhattacharya et al. (2021) | $7.664^{\pm.030}$ | $0.165^{\pm.001}$ | $0.267^{\pm.002}$ | $0.345^{\pm.002}$ | $6.030^{\pm.008}$ | $6.409^{\pm.071}$ | - |
| | Hier Ghosh et al. (2021) | $6.532^{\pm.024}$ | $0.301^{\pm.002}$ | $0.425^{\pm.002}$ | $0.552^{\pm.004}$ | $5.012^{\pm.018}$ | $8.332^{\pm.042}$ | - |
| | TEMOS Petrovich et al. (2022) | $3.734^{\pm.028}$ | $0.424^{\pm.002}$ | $0.612^{\pm.002}$ | $0.722^{\pm.002}$ | $3.703^{\pm.008}$ | $8.973^{\pm.071}$ | $0.368^{\pm.018}$ |
| | T2M Guo et al. (2022a) | $1.087^{\pm.021}$ | $0.455^{\pm.003}$ | $0.636^{\pm.003}$ | $0.736^{\pm.002}$ | $3.347^{\pm.008}$ | $9.175^{\pm.083}$ | $2.219^{\pm.074}$ |
| | MDM Tevet et al. (2022) | $0.544^{\pm.044}$ | - | - | $0.611^{\pm.007}$ | $5.566^{\pm.027}$ | $9.559^{\pm.086}$ | $\underline{2.799^{\pm.072}}$ |
| | MLD Chen et al. (2023) | $0.473^{\pm.013}$ | $0.481^{\pm.003}$ | $0.673^{\pm.003}$ | $0.772^{\pm.002}$ | $3.196^{\pm.010}$ | $9.724^{\pm.082}$ | $2.413^{\pm.079}$ |
| | MotionDiffuse Zhang et al. (2024) | $0.630^{\pm.001}$ | $0.491^{\pm.001}$ | $0.681^{\pm.001}$ | $0.782^{\pm.001}$ | $3.113^{\pm.001}$ | $9.410^{\pm.049}$ | $1.553^{\pm.042}$ |
| | PhysDiff Yuan et al. (2023) | $0.433$ | - | - | $0.631$ | - | - | - |
| | Fg-T2M Wang et al. (2023) | $0.243^{\pm.019}$ | $0.492^{\pm.002}$ | $0.683^{\pm.003}$ | $0.783^{\pm.002}$ | $3.109^{\pm.007}$ | $9.278^{\pm.072}$ | $1.614^{\pm.049}$ |
| | ReMoDiffuse Zhang et al. (2023b) | $0.103^{\pm.004}$ | $0.510^{\pm.005}$ | $0.698^{\pm.006}$ | $0.795^{\pm.004}$ | $2.974^{\pm.016}$ | $9.018^{\pm.075}$ | $1.795^{\pm.043}$ |
| | GraphMotion Jin et al. (2023) | $0.116^{\pm.007}$ | $0.504^{\pm.003}$ | $0.699^{\pm.002}$ | $0.785^{\pm.002}$ | $3.070^{\pm.008}$ | $9.692^{\pm.067}$ | $2.766^{\pm.096}$ |
| | FineMoGen Zhang et al. (2023c) | $0.151^{\pm.008}$ | $0.504^{\pm.003}$ | $0.690^{\pm.002}$ | $0.784^{\pm.002}$ | $2.998^{\pm.008}$ | $9.263^{\pm.067}$ | $2.696^{\pm.079}$ |
| | EMDM Zhou et al. (2025) | $0.112^{\pm.019}$ | $0.498^{\pm.007}$ | $0.684^{\pm.006}$ | $0.786^{\pm.006}$ | $3.110^{\pm.027}$ | $\underline{9.551^{\pm.078}}$ | $1.641^{\pm.078}$ |
| | MotionLCM Dai et al. (2025) | $0.304^{\pm.012}$ | $0.502^{\pm.003}$ | $0.698^{\pm.002}$ | $0.798^{\pm.002}$ | $3.012^{\pm.007}$ | $9.607^{\pm.066}$ | $2.259^{\pm.092}$ |
| | MotionMamba Zhang et al. (2025b) | $0.281^{\pm.009}$ | $0.502^{\pm.003}$ | $0.693^{\pm.002}$ | $0.792^{\pm.002}$ | $3.060^{\pm.058}$ | $9.871^{\pm.084}$ | $2.294^{\pm.058}$ |
| | COME (Ours) | $\textbf{0.041}^{\pm.002}$ | $\textbf{0.526}^{\pm.003}$ | $\textbf{0.723}^{\pm.002}$ | $\textbf{0.816}^{\pm.002}$ | $\textbf{2.898}^{\pm.006}$ | $9.532^{\pm.062}$ | $1.704^{\pm.059}$ |
| Discrete | TM2T Guo et al. (2022b) | $1.501^{\pm.017}$ | $0.424^{\pm.003}$ | $0.618^{\pm.002}$ | $0.729^{\pm.002}$ | $3.467^{\pm.011}$ | $8.589^{\pm.076}$ | $2.424^{\pm.093}$ |
| | T2M-GPT Zhang et al. (2023a) | $0.141^{\pm.005}$ | $0.492^{\pm.003}$ | $0.679^{\pm.002}$ | $0.775^{\pm.002}$ | $3.121^{\pm.009}$ | $9.761^{\pm.081}$ | $1.831^{\pm.048}$ |
| | M2DM Kong et al. (2023) | $0.352^{\pm.005}$ | $0.497^{\pm.003}$ | $0.682^{\pm.002}$ | $0.763^{\pm.003}$ | $3.134^{\pm.010}$ | $9.926^{\pm.073}$ | $\textbf{3.587}^{\pm.072}$ |
| | MotionGPT Jiang et al. (2023) | $0.567$ | - | - | - | $3.775$ | $9.006$ | - |
| | AvatarGPT Zhou et al. (2024) | $0.168^{\pm.008}$ | $0.510^{\pm.005}$ | $0.702^{\pm.005}$ | $0.796^{\pm.003}$ | - | $9.624^{\pm.055}$ | - |
| | CoMo Huang et al. (2025) | $0.262^{\pm.004}$ | $0.502^{\pm.002}$ | $0.692^{\pm.007}$ | $0.790^{\pm.002}$ | $3.032^{\pm.015}$ | $9.936^{\pm.066}$ | $1.013^{\pm.046}$ |
| | AttT2M Zhong et al. (2023) | $0.112^{\pm.006}$ | $0.499^{\pm.003}$ | $0.690^{\pm.002}$ | $0.786^{\pm.002}$ | $3.038^{\pm.007}$ | $9.700^{\pm.090}$ | $2.452^{\pm.051}$ |
| | ParCo Zou et al. (2025) | $0.109^{\pm.005}$ | $0.515^{\pm.003}$ | $0.706^{\pm.003}$ | $0.801^{\pm.002}$ | $2.927^{\pm.008}$ | $9.576^{\pm.088}$ | $1.382^{\pm.060}$ |
| | DiverseMotion Lou et al. (2023) | $0.072^{\pm.004}$ | $0.515^{\pm.003}$ | $0.706^{\pm.002}$ | $0.802^{\pm.002}$ | $2.941^{\pm.007}$ | $9.683^{\pm.102}$ | - |
| | MMM Pinyoanuntapong et al. (2024) | $0.080^{\pm.003}$ | $0.504^{\pm.003}$ | $0.696^{\pm.003}$ | $0.794^{\pm.002}$ | $2.998^{\pm.007}$ | $9.411^{\pm.058}$ | $1.164^{\pm.041}$ |
| | MoMask Guo et al. (2024) | $\underline{0.045^{\pm.002}}$ | $0.521^{\pm.002}$ | $0.713^{\pm.002}$ | $0.807^{\pm.002}$ | $2.958^{\pm.008}$ | - | $1.241^{\pm.040}$ |
| | BAMM Pinyoanuntapong et al. (2025b) | $0.055^{\pm.002}$ | $\underline{0.525^{\pm.002}}$ | $\underline{0.720^{\pm.003}}$ | $\underline{0.814^{\pm.003}}$ | $\underline{2.919^{\pm.008}}$ | $9.717^{\pm.089}$ | $1.687^{\pm.051}$ |

Table 1: Quantitative evaluation on HumanML3D. ↑ and ↓ denote that higher and lower values are better, respectively, while → denotes that the values closer to the real motion are better. **Red** face indicates the best result, while underscore refers to the second best.

0.999), and dropout of 0.1 is applied to mitigate overfitting. Batch sizes are set to 64 (HumanML3D) and 16 (KIT-ML). We adopt DDPM Ho et al. (2020) with 1,000 denoising steps and a condition masking probability of 0.1 for CFG. At inference, the CFG scale is set to 2.5 (HumanML3D) and 5.5 (KIT-ML). We generate motion sequences using the SDE variant of DPM-Solver++ (2nd-order)Lu et al. (2022) with Karras SigmasKarras et al. (2022) in **10** steps. All experiments are conducted on NVIDIA A6000 GPUs with 48GB of memory.

**Quantitative Comparisons of Motion Tokenizers.** Consistent with prior work Guo et al. (2022a; 2024), we compare various continuous and discrete motion tokenizers, including several discrete methods such as VQ-VAE and RVQ-VAE, as well as continuous approaches like VAE and MoCMAE (ours). As shown in Table 2, MoCMAE demonstrates superior reconstruction capabilities as a continuous tokenizer, significantly pushing the boundaries of performance and outperforming both discrete and continuous methods reported in existing literature.

| Paradigms | Methods | HumanML3D | | KIT-ML | |
|---|---|---|---|---|---|
| | | FID↓ | MPJPE↓ | FID↓ | MPJPE↓ |
| Discrete | VQVAE(TM2T Guo et al. (2022b)) | 0.307 | 230.1 | - | - |
| | VQVAE(M2DM Kong et al. (2023)) | 0.063 | - | 0.413 | - |
| | VQVAE(T2M-GPT Zhang et al. (2023a)) | 0.070 | 58.0 | 0.472 | - |
| | VQVAE(MMM Pinyoanuntapong et al. (2024)) | 0.075 | - | 0.641 | - |
| | RVQVAE(MoMask Guo et al. (2024)) | 0.019 | 29.5 | 0.112 | 37.2 |
| Continuous | VAE(VPoser-t Pavlakos et al. (2019)) | 1.430 | 75.6 | - | - |
| | VAE(ACTOR Petrovich et al. (2021)) | 0.341 | 65.3 | - | - |
| | VAE(MLD Chen et al. (2023) Dai et al. (2025)) | 0.017 | 14.7 | - | - |
| | MoCMAE (Ours) | **0.002** | **8.8** | **0.023** | **15.7** |

Table 2: Evaluation of motion tokenizer. **Red** face indicates the best result.

**Qualitative Comparisons of Motion Tokenizers.** We compare representative motion tokenizers, including discrete ones, VQ-VAE (used in T2M-GPT, MMM) and RVQ-VAE (used in MoMask, BAMM), and continuous ones, VAE (used in MLD, MotionLCM) and our proposed MoCMAE.

Quantitative results (Tab. 2) show that MoCMAE achieves substantially better reconstruction and generation quality, with nearly ten times higher reconstruction fidelity compared to baselines. Quantitative results (Tab. 7) show that MoCMAE consistently outperforms both continuous and discrete encoders, achieving higher Silhouette Score (SC) and 5-NN accuracy, and lower Davies–Bouldin Index (DBI). Qualitative visualizations (Figs. 5 and 6) further show that MoCMAE produces broader and more coherent latent distributions, covering a larger portion of the motion manifold and forming

| Methods | Reconstruction | | Generation | |
|---|---|---|---|---|
| | FID↓ | MPJPE↓ | FID↓ | MM-Dist↓ |
| MoCMAE | 0.002 | 8.8 | $0.041^{\pm.002}$ | $2.898^{\pm.006}$ |
| w/o TranBlock | 0.005 | 13.8 | $0.068^{\pm.012}$ | $2.962^{\pm.023}$ |
| w/o MMM | 0.005 | 14.2 | $0.064^{\pm.009}$ | $2.993^{\pm.013}$ |
| w/o CL | 0.004 | 9.6 | $0.058^{\pm.012}$ | $2.974^{\pm.017}$ |
| *Masking Proportion* | | | | |
| 10% | 0.004 | 10.2 | $0.062^{\pm.002}$ | $2.979^{\pm.009}$ |
| 20% | 0.004 | 9.7 | $0.058^{\pm.002}$ | $2.962^{\pm.007}$ |
| 30% | 0.003 | 9.2 | $0.055^{\pm.003}$ | $2.937^{\pm.007}$ |
| 40% | 0.002 | 8.9 | $0.047^{\pm.003}$ | $2.916^{\pm.006}$ |
| 50% | 0.002 | 8.8 | $0.041^{\pm.002}$ | $2.898^{\pm.006}$ |
| 60% | 0.002 | 8.7 | $0.046^{\pm.003}$ | $2.906^{\pm.006}$ |
| 70% | 0.002 | 8.9 | $0.058^{\pm.002}$ | $2.937^{\pm.007}$ |
| 80% | 0.001 | 7.6 | $0.089^{\pm.004}$ | $3.034^{\pm.009}$ |
| 90% | 0.001 | 6.8 | $0.103^{\pm.005}$ | $3.062^{\pm.012}$ |
| *Contrastive Loss Weight* | | | | |
| 1 | 0.003 | 9.2 | $0.055^{\pm.002}$ | $2.923^{\pm.004}$ |
| 0.1 | 0.002 | 8.8 | $0.041^{\pm.002}$ | $2.898^{\pm.006}$ |
| 0.01 | 0.002 | 8.9 | $0.048^{\pm.006}$ | $2.916^{\pm.012}$ |
| MoCMVAE | 0.002 | 9.2 | $0.072^{\pm.003}$ | $2.982^{\pm.006}$ |

Table 4: Ablation study result of MoCMAE.

| Paradigms | Methods | FID↓ | R Precision↑ | | | MM-Dist↓ | Diversity→ | MModality↑ |
|---|---|---|---|---|---|---|---|---|
| | | | Top 1 | Top 2 | Top 3 | | | |
| | Ground Truth | $0.031^{\pm.004}$ | $0.424^{\pm.005}$ | $0.649^{\pm.006}$ | $0.779^{\pm.006}$ | $2.788^{\pm.012}$ | $11.080^{\pm.097}$ | - |
| Continuous | Seq2Seq Plappert et al. (2018) | $11.75^{\pm.035}$ | $0.180^{\pm.002}$ | $0.300^{\pm.002}$ | $0.396^{\pm.002}$ | $5.529^{\pm.007}$ | $6.223^{\pm.061}$ | - |
| | JL2P Lucas et al. (2022) | $11.02^{\pm.046}$ | $0.246^{\pm.002}$ | $0.387^{\pm.002}$ | $0.486^{\pm.002}$ | $5.296^{\pm.008}$ | $7.676^{\pm.058}$ | - |
| | T2G Bhattacharya et al. (2021) | $7.664^{\pm.030}$ | $0.165^{\pm.001}$ | $0.267^{\pm.002}$ | $0.345^{\pm.002}$ | $6.030^{\pm.008}$ | $6.409^{\pm.071}$ | - |
| | Hier Ghosh et al. (2021) | $6.523^{\pm.024}$ | $0.301^{\pm.002}$ | $0.425^{\pm.002}$ | $0.552^{\pm.004}$ | $5.012^{\pm.018}$ | $8.332^{\pm.042}$ | - |
| | TEMOS Petrovich et al. (2022) | $3.717^{\pm.028}$ | $0.353^{\pm.002}$ | $0.561^{\pm.002}$ | $0.687^{\pm.002}$ | $3.417^{\pm.008}$ | $10.84^{\pm.100}$ | $0.532^{\pm.018}$ |
| | T2M Guo et al. (2022a) | $3.022^{\pm.107}$ | $0.361^{\pm.005}$ | $0.559^{\pm.007}$ | $0.681^{\pm.007}$ | $3.488^{\pm.028}$ | $10.72^{\pm.145}$ | $2.052^{\pm.107}$ |
| | MDM Tevet et al. (2022) | $0.497^{\pm.021}$ | - | - | $0.396^{\pm.004}$ | $9.191^{\pm.022}$ | $10.85^{\pm.109}$ | $1.907^{\pm.214}$ |
| | MLD Chen et al. (2023) | $0.404^{\pm.027}$ | $0.390^{\pm.008}$ | $0.609^{\pm.008}$ | $0.734^{\pm.007}$ | $3.204^{\pm.027}$ | $10.80^{\pm.117}$ | $2.192^{\pm.071}$ |
| | MotionDiffuse Zhang et al. (2024) | $1.954^{\pm.062}$ | $0.417^{\pm.004}$ | $0.621^{\pm.004}$ | $0.739^{\pm.004}$ | $2.958^{\pm.005}$ | $\underline{11.10}^{\pm.143}$ | $0.730^{\pm.013}$ |
| | Fg-T2M Wang et al. (2023) | $0.571^{\pm.047}$ | $0.418^{\pm.005}$ | $0.626^{\pm.004}$ | $0.745^{\pm.004}$ | $3.114^{\pm.015}$ | $10.93^{\pm.083}$ | $1.019^{\pm.029}$ |
| | AMD Jing et al. (2024) | $0.233^{\pm.068}$ | $0.401^{\pm.005}$ | - | - | $9.165^{\pm.032}$ | $10.971^{\pm.126}$ | $1.600^{\pm.174}$ |
| | ReMoDiffuse Zhang et al. (2023b) | $\mathbf{0.155}^{\pm.006}$ | $0.427^{\pm.014}$ | $0.641^{\pm.004}$ | $0.765^{\pm.055}$ | $2.814^{\pm.012}$ | $10.80^{\pm.105}$ | $1.239^{\pm.028}$ |
| | FineMoGen Zhang et al. (2023c) | $\underline{0.178}^{\pm.007}$ | $0.432^{\pm.006}$ | $0.649^{\pm.005}$ | $0.772^{\pm.008}$ | $2.869^{\pm.014}$ | $10.85^{\pm.115}$ | $1.877^{\pm.093}$ |
| | GraphMotion Jin et al. (2024) | $0.313^{\pm.013}$ | $0.429^{\pm.007}$ | $0.648^{\pm.006}$ | $0.769^{\pm.006}$ | $3.076^{\pm.022}$ | $\mathbf{11.12}^{\pm.135}$ | $3.627^{\pm.113}$ |
| | MotionMamba Zhang et al. (2025b) | $0.307^{\pm.041}$ | $0.419^{\pm.006}$ | $0.645^{\pm.005}$ | $0.765^{\pm.006}$ | $3.021^{\pm.025}$ | $11.02^{\pm.098}$ | $1.678^{\pm.064}$ |
| | COME (Ours) | $0.189^{\pm.018}$ | $\mathbf{0.443}^{\pm.012}$ | $\mathbf{0.666}^{\pm.009}$ | $\mathbf{0.791}^{\pm.008}$ | $\mathbf{2.715}^{\pm.028}$ | $11.042^{\pm.081}$ | $1.791^{\pm.025}$ |
| Discrete | TM2T Guo et al. (2022b) | $3.599^{\pm.153}$ | $0.280^{\pm.005}$ | $0.463^{\pm.006}$ | $0.587^{\pm.005}$ | $4.591^{\pm.026}$ | $9.473^{\pm.117}$ | $\underline{3.292}^{\pm.081}$ |
| | T2M-GPT Zhang et al. (2023a) | $0.514^{\pm.029}$ | $0.416^{\pm.006}$ | $0.627^{\pm.006}$ | $0.745^{\pm.006}$ | $3.007^{\pm.023}$ | $10.86^{\pm.094}$ | $1.570^{\pm.039}$ |
| | M2DM Kong et al. (2023) | $0.515^{\pm.029}$ | $0.416^{\pm.004}$ | $0.628^{\pm.004}$ | $0.743^{\pm.004}$ | $3.015^{\pm.017}$ | $11.417^{\pm.097}$ | $\mathbf{3.325}^{\pm.037}$ |
| | MotionGPT Jiang et al. (2023) | $0.597$ | - | - | - | $3.394$ | $10.54$ | - |
| | ParCo Zou et al. (2025) | $0.453^{\pm.027}$ | $0.430^{\pm.004}$ | $0.649^{\pm.007}$ | $0.772^{\pm.006}$ | $2.820^{\pm.028}$ | $10.95^{\pm.094}$ | $1.245^{\pm.022}$ |
| | CoMo Huang et al. (2025) | $0.332^{\pm.045}$ | $0.422^{\pm.009}$ | $0.638^{\pm.007}$ | $0.765^{\pm.011}$ | $2.873^{\pm.021}$ | $10.95^{\pm.196}$ | $1.249^{\pm.008}$ |
| | AttT2M Zhong et al. (2023) | $0.870^{\pm.039}$ | $0.413^{\pm.006}$ | $0.632^{\pm.006}$ | $0.751^{\pm.006}$ | $3.039^{\pm.021}$ | $10.96^{\pm.123}$ | $2.281^{\pm.047}$ |
| | DiverseMotion Lou et al. (2023) | $0.468^{\pm.098}$ | $0.416^{\pm.005}$ | $0.637^{\pm.008}$ | $0.760^{\pm.011}$ | $2.892^{\pm.041}$ | $10.873^{\pm.101}$ | - |
| | MMM Pinyoanuntapong et al. (2024) | $0.429^{\pm.019}$ | $0.381^{\pm.005}$ | $0.590^{\pm.006}$ | $0.718^{\pm.005}$ | $3.146^{\pm.019}$ | $10.633^{\pm.097}$ | $1.105^{\pm.026}$ |
| | MoMask Guo et al. (2024) | $0.204^{\pm.011}$ | $0.433^{\pm.007}$ | $0.656^{\pm.005}$ | $0.781^{\pm.005}$ | $2.779^{\pm.022}$ | - | $1.131^{\pm.043}$ |
| | BAMM Pinyoanuntapong et al. (2025b) | $0.183^{\pm.013}$ | $\underline{0.438}^{\pm.009}$ | $\underline{0.661}^{\pm.009}$ | $\underline{0.788}^{\pm.005}$ | $\underline{2.723}^{\pm.026}$ | $11.008^{\pm.094}$ | $1.609^{\pm.065}$ |

Table 3: Quantitative evaluation on KIT-ML. ↑ and ↓ denote that higher and lower values are better, respectively, while → denotes that the values closer to the real motion are better. **Red** face indicates the best result, while underscore refers to the second best.

clearer local clusters. This structured latent space enables more effective diffusion-based denoising and results in higher-fidelity motion generation. Detailed analyses and additional results can be found in Appendix 5.3.

Compared to standard VQ-VAE, RVQ-VAE produces a broader and more evenly distributed feature space by leveraging multiple codebooks, while continuous VAEs often lead to overly compact and clustered representations, which can harm both reconstruction and generation quality. This concentration issue is particularly problematic for generation. Overly dense feature distributions reduce inter-sample separability, making it difficult for generative models to synthesize accurate motions. Diffusion models, in particular, struggle to fully denoise features in such tight spaces, which can result in generated motions aligning incorrectly with features of other samples. This challenge is typically more severe than in discrete models, where token-level classification provides stronger constraints. MoCMAE effectively addresses these limitations. Contrastive learning explicitly enhances separability between motion samples, while masked motion modeling improves the encoder's ability to capture essential spatiotemporal patterns. These strategies reduce the impact of noise and misalignment during generation and lead to a more discriminative and well-distributed latent space. As a result, MoCMAE achieves superior performance in both motion discrimination and generation tasks.

**Quantitative Comparisons.** Following prior work Guo et al. (2022a; 2024), we evaluate ccDIT using the standard metrics. We compare with various methods, including continuous and discrete generation approaches. As shown in Tables 1 and 3, our method represents a significant advancement in the continuous generation paradigm, achieving state-of-the-art performance and surpassing both continuous and discrete methods. Additionally, as shown in Fig. 2, our method matches the speed of MotionLCM while achieving $10\times$ better generation quality (FID 0.041 vs. 0.467).

Crucially, unlike previous methods that rely on additional information—such as ground-truth motion

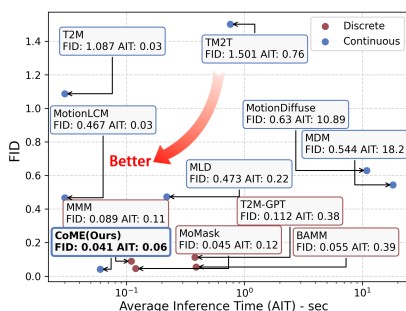

Figure 2: Quality and efficiency comparison between continuous and discrete methods.

lengths Guo et al. (2024), fine-grained annotations Wang et al. (2023); Zhang et al. (2023c), or advanced text encoders Dai et al. (2025)—our method only requires standard CLIP text encodings and basic textual descriptions. Despite this simplicity, our approach achieves SOTA performance across multiple benchmarks. This highlights the effectiveness of diffusion models in T2M generation,

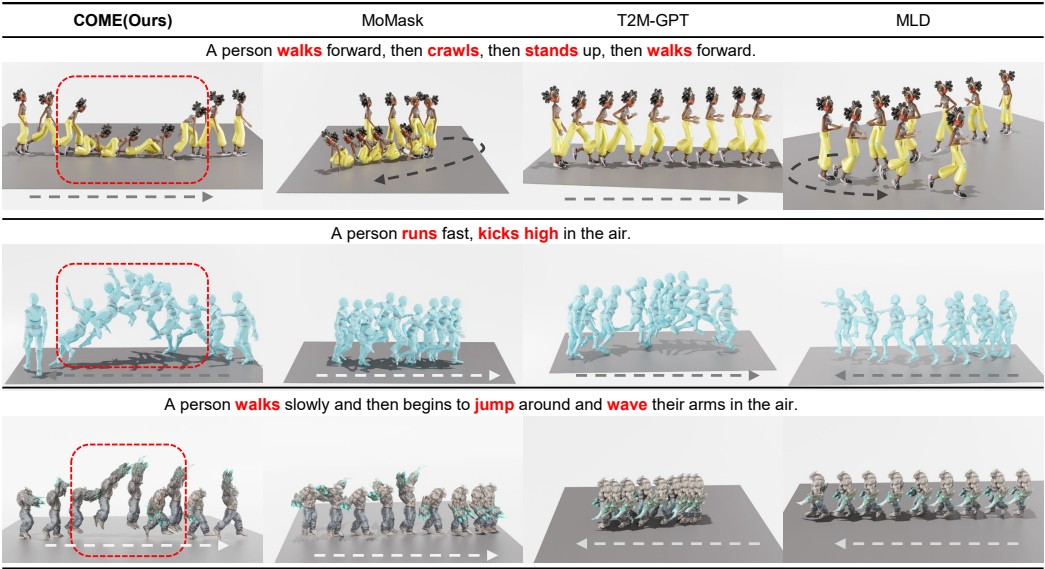

Figure 3: Qualitative results of T2M. **Red** highlight the keywords. When the input text describes multiple actions, existing methods often miss some, while our approach generates more comprehensive motions and better understands details like "kicks high" and "jump and wave". We add temporal arrows to aid understanding, and supplementary videos provide a more intuitive comparison.

showcasing their potential for future research. Additionally, diffusion-based methods offer greater diversity in generated motions and more flexibility in editing and control, making them valuable for the future of T2M generation (see Sec. 5.2).

**Qualitative Comparisons.** Fig. 3 compares our method with MoMask, MLD, and T2M-GPT on prompts involving compound actions (e.g., "jump and wave") or fine-grained semantics (e.g., "kicks high"). While both MoMask and our method perform well on standard benchmarks, we include more challenging cases to better examine their qualitative behaviors. MoMask generally captures the overall semantics but may miss subtle motion details, as it relies solely on sentence-level features without fine-grained grounding mechanisms. In contrast, our method combines global and word-level cues through AdaLN-Zero and cross-attention, resulting in more semantically aligned motion generations. In some cases, such as crawling or footwork-heavy actions, minor interpenetrations may occur due to the lack of explicit hand and foot joint annotations in the dataset. This limitation affects all methods to some extent, as these extremities are approximated rather than precisely supervised.

**User Study.** User studies are vital for evaluating generative models, as they reflect practical effectiveness. We conducted a user study with 50 participants aged 20 to 40 from diverse backgrounds (50% female, 50% male). Participants assessed three core aspects of the generated motions: (1) motion-text alignment, measuring how well the motion aligns with the provided text description, (2) motion naturalness, evaluating the realism and coherence of the motion, and (3) overall quality, reflecting the general impression of the motion's plausibility and engagement. We generated 20 motion videos for each method, including MLD, T2M-GPT, MoMask, and our approach. Participants were asked to compare pairs of videos and select the one that performed better on the three aspects mentioned above. Videos

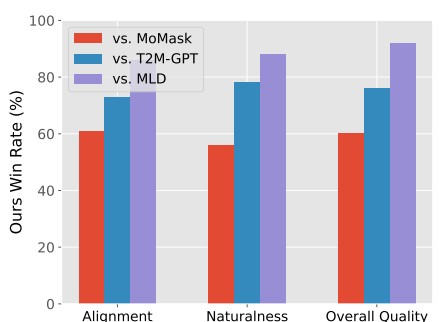

Figure 4: User Study Results. Compared with existing methods, our approach receives the highest preference in terms of alignment, naturalness, and overall quality.

were presented in a randomized order to mitigate positional bias. As shown in Fig. 4 and Fig. 19 , our method outperformed the alternatives in motion-text alignment, naturalness, and overall quality, resulting in the best user experience.

## 5.1 ABLATION STUDIES

We conduct ablations on HumanML3D to analyze the design choices and training strategies of both MoCMAE and ccDIT. *Additional detailed ablation results are provided in the appendix.*

**MoCMAE.** We evaluate the effect of architectural components and training strategies, including the use of transformer blocks, different masking ratios, and contrastive loss weights. As shown in Table 4, a masking ratio of 0.5 and a contrastive loss weight of 0.1 yield the best performance, demonstrating the importance of effective latent perturbation and representation discrimination for improved reconstruction and generalization.

**ccDIT.** We further assess the contribution of key components in ccDIT, including sentence-level semantics (sentence), fine-grained word-level cues (word), skip connections (skip), classifier-free guidance (CFG), SNR-based training techniques (Zero-SNR, Stable-Min-SNR), and different diffusion backbones (DIT, MMDIT Esser et al. (2024), and

| Methods | FID↓ | R Precision↑ | | | Diversity→ |
|---|---|---|---|---|---|
| | | Top 1 | Top 2 | Top 3 | |
| ccDIT | $0.041^{\pm.002}$ | $0.526^{\pm.003}$ | $0.723^{\pm.002}$ | $0.816^{\pm.002}$ | $9.532^{\pm.062}$ |
| w/o sentence | $0.076^{\pm.004}$ | $0.508^{\pm.002}$ | $0.712^{\pm.004}$ | $0.801^{\pm.004}$ | $9.762^{\pm.071}$ |
| w/o word | $0.089^{\pm.003}$ | $0.504^{\pm.002}$ | $0.708^{\pm.002}$ | $0.804^{\pm.003}$ | $9.961^{\pm.073}$ |
| w/o skip | $0.051^{\pm.003}$ | $0.521^{\pm.002}$ | $0.717^{\pm.002}$ | $0.813^{\pm.003}$ | $9.864^{\pm.074}$ |
| w/o cfg | $0.053^{\pm.002}$ | $0.517^{\pm.002}$ | $0.714^{\pm.003}$ | $0.809^{\pm.002}$ | $9.653^{\pm.088}$ |
| w/o Zeros-SNR | $0.092^{\pm.003}$ | $0.511^{\pm.002}$ | $0.704^{\pm.003}$ | $0.801^{\pm.003}$ | $9.784^{\pm.076}$ |
| w/o Stable-Min-SNR-$\gamma$ | $0.096^{\pm.003}$ | $0.509^{\pm.003}$ | $0.701^{\pm.002}$ | $0.802^{\pm.003}$ | $9.691^{\pm.089}$ |
| *Diffusion Block* | | | | | |
| DIT | $0.078^{\pm.002}$ | $0.512^{\pm.003}$ | $0.711^{\pm.002}$ | $0.805^{\pm.002}$ | $9.716^{\pm.073}$ |
| ccDIT | $0.041^{\pm.002}$ | $0.526^{\pm.003}$ | $0.723^{\pm.002}$ | $0.816^{\pm.002}$ | $9.532^{\pm.062}$ |
| MMDIT | $0.054^{\pm.002}$ | $0.518^{\pm.002}$ | $0.719^{\pm.002}$ | $0.809^{\pm.002}$ | $9.899^{\pm.073}$ |
| *Diffusion Target* | | | | | |
| $\epsilon$ | $0.059^{\pm.002}$ | $0.519^{\pm.003}$ | $0.713^{\pm.002}$ | $0.806^{\pm.003}$ | $9.764^{\pm.084}$ |
| $\mathbf{x}_0$ | $0.046^{\pm.002}$ | $0.513^{\pm.003}$ | $0.714^{\pm.002}$ | $0.811^{\pm.002}$ | $9.983^{\pm.078}$ |
| $\mathbf{v}$ | $0.041^{\pm.002}$ | $0.526^{\pm.003}$ | $0.723^{\pm.002}$ | $0.816^{\pm.002}$ | $9.532^{\pm.062}$ |
| *Stable-Min-SNR-$\gamma$: $\gamma$* | | | | | |
| 1 | $0.053^{\pm.003}$ | $0.509^{\pm.003}$ | $0.704^{\pm.003}$ | $0.803^{\pm.003}$ | $9.721^{\pm.096}$ |
| 2 | $0.050^{\pm.002}$ | $0.512^{\pm.003}$ | $0.709^{\pm.003}$ | $0.801^{\pm.003}$ | $9.693^{\pm.073}$ |
| 3 | $0.046^{\pm.002}$ | $0.518^{\pm.003}$ | $0.715^{\pm.003}$ | $0.809^{\pm.002}$ | $9.769^{\pm.069}$ |
| 4 | $0.041^{\pm.002}$ | $0.526^{\pm.003}$ | $0.723^{\pm.002}$ | $0.816^{\pm.002}$ | $9.532^{\pm.062}$ |
| 5 | $0.045^{\pm.002}$ | $0.519^{\pm.002}$ | $0.716^{\pm.003}$ | $0.807^{\pm.002}$ | $9.736^{\pm.072}$ |
| 6 | $0.051^{\pm.002}$ | $0.515^{\pm.002}$ | $0.709^{\pm.002}$ | $0.803^{\pm.002}$ | $9.762^{\pm.084}$ |
| *Stable-Min-SNR-$\gamma$: $\frac{1}{\gamma}$* | | | | | |
| 1000 | $0.071^{\pm.002}$ | $0.507^{\pm.002}$ | $0.704^{\pm.002}$ | $0.801^{\pm.002}$ | $9.641^{\pm.096}$ |
| 100 | $0.066^{\pm.002}$ | $0.510^{\pm.002}$ | $0.708^{\pm.003}$ | $0.804^{\pm.002}$ | $9.693^{\pm.089}$ |
| 10 | $0.059^{\pm.002}$ | $0.516^{\pm.002}$ | $0.714^{\pm.003}$ | $0.806^{\pm.002}$ | $9.762^{\pm.087}$ |
| 5 | $0.054^{\pm.002}$ | $0.519^{\pm.002}$ | $0.718^{\pm.002}$ | $0.810^{\pm.002}$ | $9.794^{\pm.073}$ |
| 4 | $0.050^{\pm.002}$ | $0.518^{\pm.003}$ | $0.716^{\pm.002}$ | $0.809^{\pm.002}$ | $9.893^{\pm.076}$ |
| 3 | $0.044^{\pm.002}$ | $0.521^{\pm.002}$ | $0.721^{\pm.003}$ | $0.813^{\pm.002}$ | $9.986^{\pm.068}$ |
| 2 | $0.041^{\pm.002}$ | $0.526^{\pm.003}$ | $0.723^{\pm.002}$ | $0.816^{\pm.002}$ | $9.532^{\pm.062}$ |
| 1 | $0.046^{\pm.002}$ | $0.520^{\pm.002}$ | $0.719^{\pm.002}$ | $0.812^{\pm.002}$ | $9.883^{\pm.073}$ |
| *Sampler Scheduler* | | | | | |
| DDPM | $0.043^{\pm.002}$ | $0.521^{\pm.002}$ | $0.723^{\pm.002}$ | $0.816^{\pm.002}$ | $9.784^{\pm.052}$ |
| DDIM | $0.046^{\pm.002}$ | $0.518^{\pm.002}$ | $0.717^{\pm.002}$ | $0.809^{\pm.002}$ | $9.872^{\pm.071}$ |
| PNDM | $0.058^{\pm.002}$ | $0.514^{\pm.003}$ | $0.713^{\pm.002}$ | $0.804^{\pm.002}$ | $9.982^{\pm.078}$ |
| Deis | $0.049^{\pm.003}$ | $0.519^{\pm.002}$ | $0.716^{\pm.003}$ | $0.811^{\pm.002}$ | $9.654^{\pm.072}$ |
| UniPC | $0.048^{\pm.002}$ | $0.518^{\pm.002}$ | $0.715^{\pm.002}$ | $0.812^{\pm.002}$ | $9.682^{\pm.069}$ |
| DPMSolver++ | $0.041^{\pm.002}$ | $0.526^{\pm.003}$ | $0.723^{\pm.002}$ | $0.816^{\pm.002}$ | $9.532^{\pm.062}$ |

Table 5: Ablation study result of ccDIT.

our ccDIT). We also compare alternative training objectives ($\epsilon$, $x_0$, $v$), evaluate SNR-based loss reweighting, and examine inference-time schedulers (DDPM, DDIM Song et al. (2020), PNDM Liu et al. (2022), Deis Zhang & Chen (2022), UniPC Zhao et al. (2024), and DPM-Solver++ Lu et al. (2022)). As summarized in Table 5, most components contribute incremental improvements across different metrics, confirming the effectiveness of our architectural design and training strategies. Notably, the joint use of global and fine-grained semantic guidance, along with ccDIT and SNR-based training, proves particularly beneficial.

## 5.2 APPLICATIONS

In addition to T2M, our framework can be extended to various downstream tasks. Here, we demonstrate the applicability of our model to controllable motion generation. *Additional applications are included in the appendix.*

| Methods | FID↓ | R-Precision↑ Top 3 | Diversity→ | Traj. err.↓ (50cm) | Loc. err.↓ (50cm) | Avg. err.↓ |
|---|---|---|---|---|---|---|
| Real | 0.002 | 0.797 | 9.503 | 0.000 | 0.000 | 0.000 |
| MDM Tevet et al. (2022) | 0.698 | 0.602 | 9.197 | 0.4022 | 0.3076 | 0.5959 |
| PriorMDM Shafir et al. (2024) | 0.475 | 0.583 | 9.156 | 0.3457 | 0.2132 | 0.4417 |
| GMD Karunratanakul et al. (2023) | 0.576 | 0.665 | 9.206 | 0.0931 | 0.0321 | 0.1439 |
| OmniControl Xie et al. (2024) | 0.218 | 0.687 | 9.422 | 0.0387 | 0.0096 | 0.0338 |
| MoitonLCM Dai et al. (2025) | 0.531 | 0.752 | 9.253 | 0.1887 | 0.0769 | 0.1897 |
| **COME-ControlNet** | **0.112** | **0.782** | **9.498** | **0.0196** | **0.0072** | **0.0113** |

Table 6: Quantitative results of comparison with existing methods on HumanML3D test set.

**Controllable Motion Generation.** Following the evaluation protocol of OmniControl Xie et al. (2024), we evaluate controllability by integrating our ccDIT backbone into the MotionLCM Dai et al. (2025) framework. Specifically, we replace the original MLD component while retaining the ControlNet architecture, resulting in a controllable variant termed **COME-ControlNet**. Control signals (e.g., trajectories, poses) are injected alongside textual prompts to guide the denoising process. As shown in Table 23, our method achieves improved motion-text alignment and diversity, while significantly reducing trajectory and location errors compared to prior approaches.

## 5.3 MOTION REPRESENTATION QUALITY

To systematically evaluate the latent motion representations, we use internal geometric metrics (Silhouette Score, Calinski–Harabasz Index, Davies–Bouldin Index) and external semantic metrics (5-NN accuracy with keyword-based pseudo-labels). Despite overlapping motion categories and coarse pseudo-labels. Pseudo-labels are constructed using nine motion-related keywords ("chicken", "stair", "crawl", "jump", "pick", "breast stroke", "ballet", "karate", "sit"), with randomly sampled instances per class to ensure a balanced evaluation set. Quantitative results (Tab. 7 show that MoC-MAE consistently outperforms both continuous (e.g., VAE) and discrete (e.g., VQ-VAE, RVQ-VAE) encoders, achieving higher SC and 5-NN accuracy and lower DBI. Qualitative visualizations (Figs. 5 and 6) further show that MoCMAE produces broader, more coherent latent distributions, covering a

Table 7: **Quantitative comparison of motion representations.** MoCMAE outperforms existing encoders on both geometric metrics (SC, CHI, DBI) and semantic alignment (5-NN Acc). Human motion categories naturally overlap, so absolute SC values are low; nonetheless, MoCMAE achieves the best separability and semantic coherence.

| Model | SC ↑ | CHI ↑ | DBI ↓ | 5-NN Acc ↑ |
|---|---|---|---|---|
| VQ-VAE | -0.086093 | 5.42 | 4.241479 | 0.4667 |
| RVQ-VAE | -0.020365 | 7.83 | 4.024332 | 0.5197 |
| VAE | -0.008453 | 1.38 | 6.662875 | 0.3125 |
| **MoCMAE** | **0.047418** | 5.23 | **3.984182** | **0.7250** |
| MoCMAE-w/o CL | 0.025364 | 4.28 | 4.086264 | 0.6425 |

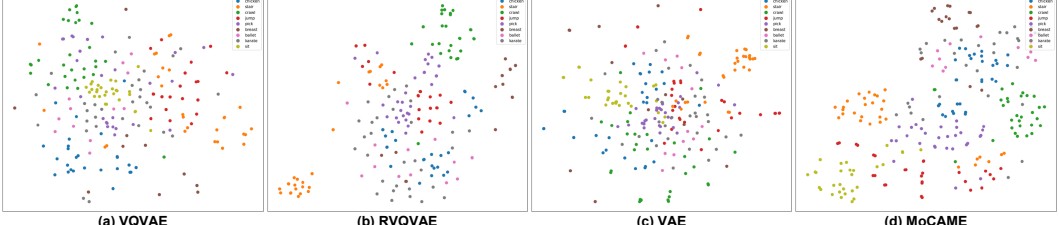

Figure 5: **t-SNE with pseudo-labels.** MoCMAE shows more coherent local groupings and clearer semantic regions compared to baselines, despite naturally overlapping human motion manifolds.

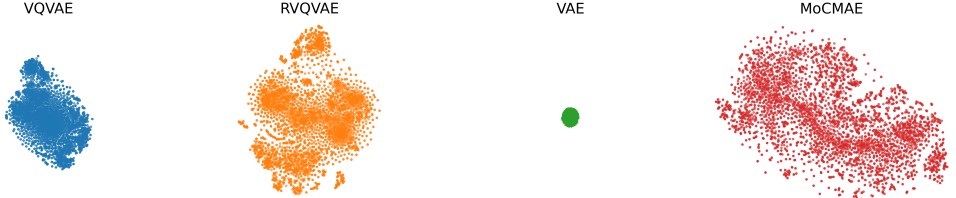

Figure 6: **Global t-SNE on HumanML3D.** VAE features are compact; discrete encoders are more spread out; MoCMAE covers a larger portion of the motion space with higher feature diversity.

larger portion of the motion manifold and forming clearer local clusters. This structured latent space enables more effective diffusion-based denoising and results in higher-fidelity motion generation. These findings demonstrate that MoCMAE learns expressive and semantically meaningful motion representations, which directly support improved generation quality and enhanced downstream task performance. More detailed analysis and experimental results can be found in Appendix 5.3.

## 6 CONCLUSION

We present COME, a continuous diffusion framework for T2M generation that revisits real-valued modeling with modern design. By combining a contrastive masked autoencoder (MoCMAE) for learning structured motion representations and a cross-conditioned diffusion transformer (ccDIT) for semantically aligned generation, COME achieves strong performance across quality, alignment, and efficiency. Unlike prior work that relies on discrete tokenization, our approach demonstrates that continuous models, when equipped with expressive representations and effective generative modeling, can rival and even surpass discrete counterparts.

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

# A APPENDIX

## A.1 MORE DETAILS ON EFFECTIVENESS AND EFFICIENCY COMPARISON

As shown in Fig. 7, our proposed framework, COME, offers significant advantages in both effectiveness and efficiency. Below, we elaborate on these improvements in terms of training cost, inference speed, and generation quality/diversity.

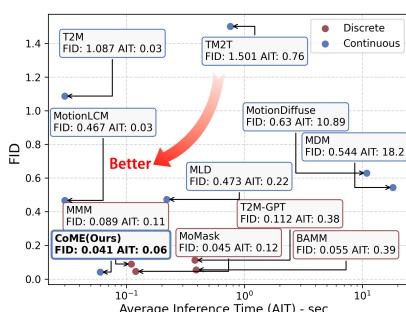

Figure 7: Quality and efficiency comparison between continuous and discrete methods.

**Training Efficiency.** COME converges in just 1,100 epochs (500 for MoCMAE and 600 for diffusion), compared to MLD's 9,000 epochs (6,000 for VAE pretraining and 3,000 for diffusion), with the total training time reduced from 302 GPU-hours to 49 GPU-hours (Table 8), achieving a 6× improvement in training efficiency. In contrast to discrete methods like MoMask and BAMM that rely on cascaded pipelines with at least **two** generative models, COME requires only a **single** diffusion model.

**Inference Speed.** COME generates motions in only 10 denoising steps—5× faster than MLD's 50-step sampling. Unlike MoMask and BAMM, which introduce additional latency due to multi-stage decoding, COME performs fast inference via a streamlined, end-to-end architecture. Notably, it matches the speed of MotionLCM while achieving **10× better quality** (FID 0.041 vs. 0.467).

**Generation Quality and Diversity.** Discrete models (e.g., MoMask, BAMM) are often constrained by deterministic codebook sampling. In contrast, COME leverages the inherent stochasticity of diffusion to generate diverse and semantically faithful motions. It achieves the best overall performance, with an FID of 0.041, outperforming MoMask (0.045), BAMM (0.055), and MLD (0.473).

In summary, the combination of training efficiency, fast inference, and high-quality generation underscores the effectiveness of our architectural and learning design. COME demonstrates that continuous diffusion models—when equipped with strong representation learning—can achieve state-of-the-art T2M synthesis with both speed and fidelity.

| Method | Training Stage | Epochs | GPU-hours | Total GPU-hours |
|---|---|---|---|---|
| **COME (Ours)** | Motion Tokenizer (MoCMAE) | 600 | 18.1 | **49.3** |
| | Diffusion Model (ccDiT) | 500 | 31.2 | |
| **MLD** | Motion Tokenizer (VAE) | 6000 | 196.8 | **301.9** |
| | Diffusion Model | 3000 | 105.1 | |

Table 8: Training cost comparison across methods. We report epochs and GPU-hours for each module (motion tokenizer and diffusion model) and the total GPU-hours using a single NVIDIA A6000 GPU. Our method converges with significantly fewer epochs and GPU-hours compared to MLD, achieving a 6× improvement in training efficiency.

## A.2 APPLICATION TO TEXT-TO-MOTION RETRIEVAL

To further validate the generalization and versatility of our motion tokenizer, we apply the trained MoCMAE to the task of text-to-motion retrieval. Specifically, we integrate MoCMAE 's encoder and decoder into a retrieval-style framework by replacing the corresponding components in TMR (Petrovich et al., 2023).

We compare our method against several strong baselines, including TEMOS (Petrovich et al., 2022), T2M (Guo et al., 2022a), TMR (Petrovich et al., 2023), and MotionPatches (Yu et al., 2024). As shown in Tables 10 and 11, our method achieves the best performance across all metrics on both datasets. Fig. 9 further illustrates that our model retrieves more accurate and semantically aligned motions, even under challenging textual descriptions.

These results highlight that MoCMAE, beyond its strong performance in motion generation, also effectively supports motion understanding tasks such as retrieval—demonstrating its value as a general-purpose motion representation learner.

### A.3    EXPERIMENTS ON ADDITIONAL DATASETS

HumanML3D and KIT-ML have become standard real-world benchmarks for evaluating T2M models. To further assess the generalizability of our framework, we extend our experiments beyond these datasets. However, most existing large-scale datasets are constructed via algorithmic extraction, with motion captions generated by LLMs such as ChatGPT. This raises concerns regarding the quality and reliability of both motion and language annotations, limiting their adoption in the community. For example, although MotionX (Lin et al., 2023) offers large-scale data, it suffers from noisy motion quality and lacks clear usage guidelines. Its enhanced version, MotionX++ (Zhang et al., 2025a), does not release updated train/val/test splits or pretrained weights, making fair comparison infeasible at this stage.

Given these limitations, we conduct additional experiments on the CombatMotionProcessed (CMP) dataset (Liao et al., 2024), a high-quality animation dataset focused on stylized fighting motions from games. CMP contains 8,700 curated animations, each annotated with three levels of text descriptions: a short caption, a caption with sensory details, and a detailed narrative. Unlike daily activity datasets such as HumanML3D or MotionX, CMP textbfasizes dynamic, game-oriented motion patterns. Moreover, it provides full access to data, code, and results from several baselines (T2M-GPT, MDM, MLD, MMM, MoMask, MotionGPT), enabling rigorous benchmarking. As shown in Table 9, our method consistently outperforms all baselines, demonstrating its effectiveness and strong adaptability to domain-shifted, high-action scenarios.

### A.4    ANALYSIS OF SAMPLING STRATEGIES: SAMPLE SCHEDULER, CFG SCALE, AND ITERATION STEPS

To assess the impact of sampling strategies on diffusion-based T2M generation, we conducted experiments on the HumanML3D dataset, evaluating two key metrics: FID and MM Dist. Each experiment was conducted once. Our analysis focused on the effects of different sample schedulers, CFG scales (evaluated at 0.5 intervals), and iteration steps (evaluated at 1 step intervals). The sampling schedulers, including DDPM (Ho et al., 2020), DDIM (Song et al., 2020), PNDM (Liu et al., 2022), Deis (Zhang & Chen, 2022), and DPMSolver++ (Lu et al., 2022), influence noise reduction and diffusion trajectories, directly impacting the trade-off between motion quality, diversity, and computational efficiency. For our experiments, we used DPMSolver++ (Lu et al., 2022) with Karras Sigmas (Karras et al., 2022) (referred to as DPM).

The results, summarized in Fig. 13 and Fig. 14, provide key insights into the optimal configurations. The model showed high sensitivity to the CFG scale, indicating the need for careful tuning to achieve optimal performance. A CFG scale of 2.5 or 3.5 consistently produced strong results across sampling strategies, effectively balancing quality and diversity. In contrast, the model exhibited lower sensitivity to the number of iteration steps, with ranges of 10-20 or 20-30, consistent with recommendations in prior works, yielding satisfactory results. Notably, DPMSolver++ combined with Karras provided superior inference efficiency and generation quality. These findings highlight the significance of fine-tuning sampling strategies to optimize T2M generation, balancing motion quality with computational efficiency.

### A.5    LIMITATIONS

Our work is grounded in a practical design principle: extracting high-quality and discriminative motion representations without increasing generation overhead. To this end, we introduce an asymmetric architecture, discriminative objectives, and reconstruction-oriented training to jointly enhance representation quality and generation performance. While these choices are empirically effective, a deeper theoretical understanding of what defines an optimal motion tokenizer remains an open question. Moreover, although our method shows strong performance on standard benchmarks, further validation is needed across diverse datasets and tasks. Extending our framework to broader mo-

| Paradigms | Methods | FID↓ | R Precision↑ | | | MM-Dist↓ | Diversity→ | MModality↑ |
|---|---|---|---|---|---|---|---|---|
| | | | Top 1 | Top 2 | Top 3 | | | |
| | Ground Truth | $0.006^{\pm.003}$ | $0.335^{\pm.004}$ | $0.513^{\pm.005}$ | $0.628^{\pm.005}$ | $3.850^{\pm.018}$ | $10.098^{\pm.102}$ | - |
| Continuous | T2M (Guo et al., 2022a) | $1.898^{\pm.059}$ | $0.252^{\pm.006}$ | $0.406^{\pm.005}$ | $0.508^{\pm.006}$ | $4.962^{\pm.031}$ | $8.975^{\pm.113}$ | $4.470^{\pm.112}$ |
| | MDM (Tevet et al., 2022) | $9.467^{\pm.217}$ | $0.049^{\pm.003}$ | $0.098^{\pm.005}$ | $0.148^{\pm.004}$ | $8.414^{\pm.048}$ | $7.608^{\pm.100}$ | $5.682^{\pm.203}$ |
| | MLD (Chen et al., 2023) | $0.628^{\pm.038}$ | $0.293^{\pm.004}$ | $0.459^{\pm.004}$ | $0.568^{\pm.004}$ | $4.331^{\pm.029}$ | $9.741^{\pm.093}$ | $3.035^{\pm.138}$ |
| | COME (ours) | $0.119^{\pm.022}$ | $0.357^{\pm.004}$ | $0.519^{\pm.004}$ | $0.638^{\pm.007}$ | $3.432^{\pm.027}$ | $10.116^{\pm.086}$ | $4.127^{\pm.116}$ |
| Discrete | T2M-GPT (Zhang et al., 2023a) | $0.177^{\pm.016}$ | $0.353^{\pm.005}$ | $0.545^{\pm.006}$ | $0.663^{\pm.005}$ | $3.701^{\pm.027}$ | $10.128^{\pm.132}$ | $1.798^{\pm.041}$ |
| | MMM (Pinyoanuntapong et al., 2024) | $0.151^{\pm.013}$ | $0.353^{\pm.004}$ | $0.545^{\pm.004}$ | $0.667^{\pm.005}$ | $3.621^{\pm.020}$ | $10.091^{\pm.086}$ | $0.757^{\pm.042}$ |
| | MoMask (Guo et al., 2024) | $0.383^{\pm.018}$ | $0.301^{\pm.005}$ | $0.481^{\pm.004}$ | $0.597^{\pm.005}$ | $4.138^{\pm.025}$ | $9.689^{\pm.092}$ | $1.968^{\pm.049}$ |
| | MotionGPT (Jiang et al., 2023) | $0.267^{\pm.017}$ | $0.306^{\pm.004}$ | $0.486^{\pm.006}$ | $0.605^{\pm.006}$ | $4.228^{\pm.032}$ | $9.357^{\pm.133}$ | $2.210^{\pm.137}$ |

Table 9: Quantitative comparison of motion generation methods on CMP dataset (Liao et al., 2024). ↓ indicates lower values are better, ↑ higher values are better, and → closer to the ground truth is better.

| Protocol | Methods | Text-motion retrieval | | | | | | Motion-text retrieval | | | | | |
|---|---|---|---|---|---|---|---|---|---|---|---|---|---|
| | | R@1↑ | R@2↑ | R@3↑ | R@5↑ | R@10↑ | MedR↓ | R@1↑ | R@2↑ | R@3↑ | R@5↑ | R@10↑ | MedR↓ |
| All | TEMOS | 2.12 | 4.09 | 5.87 | 8.26 | 13.52 | 173.0 | 3.86 | 4.54 | 6.94 | 9.38 | 14.00 | 183.25 |
| | T2M | 1.80 | 3.42 | 4.79 | 7.12 | 12.47 | 81.00 | 2.92 | 3.74 | 6.00 | 8.36 | 12.95 | 81.50 |
| | TMR | 8.92 | 12.04 | 16.33 | 22.06 | 33.37 | 25.00 | 9.44 | 11.84 | 16.90 | 22.92 | 32.21 | 26.00 |
| | MotionPatches | 10.80 | 14.98 | 20.00 | 26.72 | 38.02 | 19.00 | 11.25 | 13.86 | 19.98 | 26.86 | 37.40 | 20.50 |
| | Ours | **13.38** | **17.81** | **25.53** | **30.92** | **45.28** | **15.50** | **14.23** | **16.68** | **24.19** | **32.46** | **43.61** | **16.50** |

Table 10: Results of text-to-motion and motion-to-text retrieval benchmark on HumanML3D.

tion understanding applications—such as style transfer, motion editing, HOI, HSI, and HHI—also remains a promising direction for future work.

### A.6 EXPERIMENTS ON ADDITIONAL DATASETS

HumanML3D and KIT-ML are standard real-world benchmarks for evaluating T2M models. To further assess the generalizability of our method, we extend our COME to more diverse datasets.

**Fine-grained Semantic Understanding on Complex Descriptions.** To evaluate our method's capability on complex textual descriptions, we conducted experiments on the SnapMoGen dataset (**?**), which features fine-grained, long-form text-motion pairs with descriptions averaging 48 words per caption. As shown in Table 12, COME achieves superior performance across all evaluation metrics, demonstrating exceptional semantic understanding and alignment capabilities. The results validate our model's ability to comprehend nuanced textual descriptions and generate motions that accurately reflect complex semantic requirements, significantly outperforming both continuous and discrete baseline methods.

**Long-horizon Temporal Modeling and Transition Quality.** We evaluated our approach on the BABEL dataset (Punnakkal et al., 2021) to assess performance on long-duration, semantically diverse motion sequences with complex temporal dependencies following FlowMDM (Barquero et al., 2024). The experimental setup focuses on both subsequence generation and transition modeling capabilities. As demonstrated in Table 13, COME achieves the best overall performance in generating temporally coherent long-horizon motions while maintaining smooth transitions between different action phases. This validates our model's superior temporal modeling capacity and its ability to handle complex sequential motion patterns effectively.

**Large-scale Generalization and Robustness.** To assess scalability and generalization across diverse linguistic patterns, we conducted comprehensive evaluation on MotionX++ (Zhang et al., 2025a), a substantially larger dataset containing over 120,000 sequences with linguistically di-

| Protocol | Methods | Text-motion retrieval | | | | | | Motion-text retrieval | | | | | |
|---|---|---|---|---|---|---|---|---|---|---|---|---|---|
| | | R@1↑ | R@2↑ | R@3↑ | R@5↑ | R@10↑ | MedR↓ | R@1↑ | R@2↑ | R@3↑ | R@5↑ | R@10↑ | MedR↓ |
| All | TEMOS | 7.11 | 13.25 | 17.59 | 24.10 | 35.66 | 24.00 | 11.69 | 15.30 | 20.12 | 26.63 | 36.39 | 26.50 |
| | T2M | 3.37 | 6.99 | 10.84 | 16.87 | 27.71 | 28.00 | 4.94 | 6.51 | 10.72 | 16.14 | 25.30 | 28.50 |
| | TMR | 10.05 | 13.87 | 20.74 | 30.03 | 44.66 | 14.00 | 11.83 | 13.74 | 22.14 | 29.39 | 38.55 | 16.00 |
| | MotionPatches | 14.02 | 21.08 | 28.91 | 34.10 | 50.00 | 10.50 | 13.61 | 17.26 | 27.54 | 33.33 | 44.77 | 13.00 |
| | Ours | **16.26** | **24.18** | **34.31** | **39.38** | **55.21** | **7.50** | **16.42** | **21.16** | **32.68** | **38.62** | **50.78** | **10.00** |

Table 11: Results of text-to-motion and motion-to-text retrieval benchmark on KIT-ML.

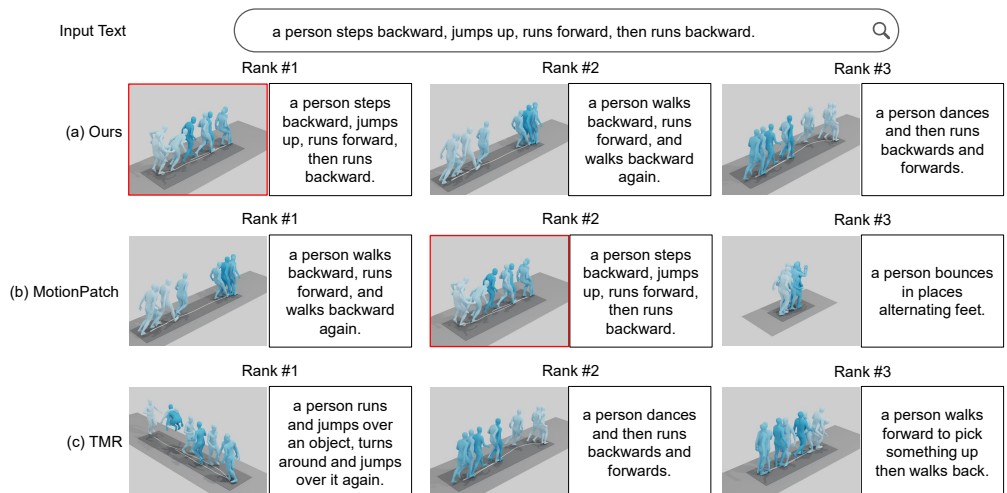

Figure 8: The text to motion retrieval results. The red box means the right sample.

verse annotations. The experimental protocol followed standard text-to-motion evaluation procedures across multiple motion generation paradigms. As shown in Table 14, COME demonstrates exceptional generalization ability, significantly outperforming both discrete and continuous baseline methods. The results confirm our model's robust performance across large-scale, diverse data scenarios and validate its strong scalability properties.

**Motion Editing Precision and Controllability.** We integrated our diffusion framework into the MotionFix benchmark (Athanasiou et al., 2024) to evaluate motion editing capabilities, replacing the original MDM backbone while maintaining the editing protocol. The evaluation focuses on measuring editing precision through retrieval-based metrics that assess both target alignment and source consistency. Table 15 demonstrates that COME achieves substantial improvements in editing precision and controllability, validating the effectiveness of our latent diffusion framework for downstream editing applications requiring fine-grained motion modifications.

**Personalized Motion Generation and Identity Preservation.** To assess personalization capabilities, we integrated our model into the PerMo benchmark (Kim et al., 2025) pipeline, focusing on identity preservation across multiple character attributes while maintaining semantic alignment with textual descriptions. The experimental setup evaluates personalization effectiveness across various character traits including age, emotional expressions, and behavioral characteristics. As shown in Table 16, COME achieves superior personalization performance with remarkable efficiency, demonstrating exceptional identity preservation capabilities while requiring significantly fewer inference steps than competing methods.

**Plug-and-Play Compatibility.** We conducted integration experiments with MotionLCM-V2 (**?**) to validate the modularity and compatibility of our proposed components. The experimental design included incremental integration of our training strategies and framework components to assess their orthogonal contributions. Table 17 demonstrates that the "+Stable", incorporating our Stable-Min-SNR-$\gamma$ training strategy, consistently improves base performance across all metrics, and the "+More" configuration, integrating our full framework including global-local alignment and latent optimization, further enhances performance. These systematic improvements demonstrate that our contributions are orthogonal and complementary to existing methods, validating our plug-and-play design philosophy and confirming that enhanced representation learning and diffusion processes can strengthen any motion generation pipeline.

**Game Scene Motion Generation.** We conduct additional experiments on the CombatMotionProcessed (CMP) dataset (Liao et al., 2024), a high-quality animation dataset focused on stylized fighting motions from games. CMP contains 8,700 curated animations, each annotated with three levels of text descriptions: a short caption, a caption with sensory details, and a detailed narrative. Unlike daily activity datasets such as HumanML3D or MotionX, CMP textbfasizes dynamic, game-oriented motion patterns. Moreover, it provides full access to data, code, and results from several baselines (T2M-GPT, MDM, MLD, MMM, MoMask, MotionGPT), enabling rigorous benchmarking. As

Table 12: Quantitative results on SnapMoGen.

| Methods | R Precision↑ | | | FID↓ | CLIP Score↑ | MModality↑ |
|---|---|---|---|---|---|---|
| | Top 1 | Top 2 | Top 3 | | | |
| Real motions | $0.940^{\pm.001}$ | $0.976^{\pm.001}$ | $0.985^{\pm.001}$ | $0.001^{\pm.000}$ | $0.837^{\pm.000}$ | - |
| MDM | $0.503^{\pm.002}$ | $0.653^{\pm.002}$ | $0.727^{\pm.002}$ | $57.783^{\pm.092}$ | $0.481^{\pm.001}$ | $\mathbf{13.412^{\pm.231}}$ |
| T2M-GPT | $0.618^{\pm.002}$ | $0.773^{\pm.002}$ | $0.812^{\pm.002}$ | $32.629^{\pm.087}$ | $0.573^{\pm.001}$ | $9.172^{\pm.181}$ |
| StableMoFusion | $0.679^{\pm.002}$ | $0.823^{\pm.002}$ | $0.888^{\pm.002}$ | $27.801^{\pm.063}$ | $0.605^{\pm.001}$ | $9.064^{\pm.138}$ |
| MARDM | $0.659^{\pm.002}$ | $0.812^{\pm.002}$ | $0.860^{\pm.002}$ | $26.878^{\pm.131}$ | $0.602^{\pm.001}$ | $9.812^{\pm.287}$ |
| MoMask | $0.777^{\pm.002}$ | $0.888^{\pm.002}$ | $0.927^{\pm.002}$ | $17.404^{\pm.051}$ | $0.664^{\pm.001}$ | $8.183^{\pm.184}$ |
| COME(ours, CLIP) | $0.786^{\pm.002}$ | $0.897^{\pm.002}$ | $0.932^{\pm.001}$ | $16.246^{\pm.070}$ | $0.672^{\pm.001}$ | $12.135^{\pm.182}$ |
| MoMask++$^{in}$(CLIP) | - | - | - | 19.96 | 0.478 | - |
| MoMask++$^{in}$(T5) | $0.805^{\pm.002}$ | $0.904^{\pm.002}$ | $0.938^{\pm.001}$ | $15.56^{\pm.071}$ | $0.684^{\pm.001}$ | $6.556^{\pm.178}$ |
| MoMask++$^{cra}$(T5) | $0.802^{\pm.001}$ | $0.905^{\pm.002}$ | $0.938^{\pm.001}$ | $\mathbf{15.06^{\pm.065}}$ | $0.685^{\pm.001}$ | $7.259^{\pm.180}$ |
| COME(ours, T5) | $\mathbf{0.807^{\pm.002}}$ | $\mathbf{0.908^{\pm.002}}$ | $\mathbf{0.944^{\pm.001}}$ | $15.162^{\pm.068}$ | $\mathbf{0.686^{\pm.001}}$ | $12.452^{\pm.164}$ |

Table 13: Quantitative results on Babel.

| Method | Subsequence | | | | Transition | | | |
|---|---|---|---|---|---|---|---|---|
| | R-prec↑ | FID↓ | Div→ | MM-Dist↓ | FID↓ | Div→ | PJ→ | AUJ↓ |
| GT | $0.715^{\pm0.003}$ | $0.00^{\pm0.00}$ | $8.42^{\pm0.15}$ | $3.36^{\pm0.00}$ | $0.00^{\pm0.00}$ | $6.20^{\pm0.06}$ | $0.02^{\pm0.00}$ | $0.00^{\pm0.00}$ |
| TEACH | $0.655^{\pm0.002}$ | $1.82^{\pm0.02}$ | $7.96^{\pm0.11}$ | $3.72^{\pm0.01}$ | $3.27^{\pm0.04}$ | $\mathbf{6.14^{\pm0.06}}$ | $0.07^{\pm0.00}$ | $0.44^{\pm0.00}$ |
| DoubleTake | $0.668^{\pm0.005}$ | $1.33^{\pm0.04}$ | $7.98^{\pm0.12}$ | $3.67^{\pm0.03}$ | $3.15^{\pm0.05}$ | $6.14^{\pm0.07}$ | $0.17^{\pm0.00}$ | $0.64^{\pm0.01}$ |
| MultiDiffusion | $0.702^{\pm0.005}$ | $1.74^{\pm0.04}$ | $8.37^{\pm0.13}$ | $3.43^{\pm0.02}$ | $6.56^{\pm0.12}$ | $5.72^{\pm0.07}$ | $0.18^{\pm0.00}$ | $0.68^{\pm0.00}$ |
| DiffCollage | $0.671^{\pm0.003}$ | $1.45^{\pm0.05}$ | $7.93^{\pm0.09}$ | $3.71^{\pm0.01}$ | $4.36^{\pm0.09}$ | $6.09^{\pm0.08}$ | $0.19^{\pm0.00}$ | $0.84^{\pm0.01}$ |
| FlowMDM | $0.702^{\pm0.004}$ | $0.99^{\pm0.04}$ | $8.36^{\pm0.13}$ | $3.45^{\pm0.02}$ | $2.61^{\pm0.06}$ | $6.47^{\pm0.05}$ | $0.06^{\pm0.00}$ | $0.13^{\pm0.00}$ |
| COME (ours) | $\mathbf{0.709^{\pm0.003}}$ | $\mathbf{0.86^{\pm0.03}}$ | $\mathbf{8.49^{\pm0.11}}$ | $\mathbf{3.39^{\pm0.01}}$ | $\mathbf{2.26^{\pm0.05}}$ | $6.31^{\pm0.04}$ | $\mathbf{0.04^{\pm0.00}}$ | $\mathbf{0.11^{\pm0.00}}$ |

shown in Table 9, our method consistently outperforms all baselines, demonstrating its effectiveness and strong adaptability to domain-shifted, high-action scenarios.

### A.7 ANALYSIS OF SAMPLING STRATEGIES: SAMPLE SCHEDULER, CFG SCALE, AND ITERATION STEPS

To assess the impact of sampling strategies on diffusion-based T2M generation, we conducted experiments on the HumanML3D dataset, evaluating two key metrics: FID and MM Dist. Each experiment was conducted once. Our analysis focused on the effects of different sample schedulers, CFG scales (evaluated at 0.5 intervals), and iteration steps (evaluated at 1 step intervals). The sampling schedulers, including DDPM (Ho et al., 2020), DDIM (Song et al., 2020), PNDM (Liu et al., 2022), Deis (Zhang & Chen, 2022), and DPMSolver++ (Lu et al., 2022), influence noise reduction and diffusion trajectories, directly impacting the trade-off between motion quality, diversity, and computational efficiency. For our experiments, we used DPMSolver++ (Lu et al., 2022) with Karras Sigmas (Karras et al., 2022) (referred to as DPM).

The results, summarized in Fig. 13 and Fig. 14, provide key insights into the optimal configurations. The model showed high sensitivity to the CFG scale, indicating the need for careful tuning to achieve optimal performance. A CFG scale of 2.5 or 3.5 consistently produced strong results across sampling strategies, effectively balancing quality and diversity. In contrast, the model exhibited lower sensitivity to the number of iteration steps, with ranges of 10-20 or 20-30, consistent with recommendations in prior works, yielding satisfactory results. Notably, DPMSolver++ combined with Karras provided superior inference efficiency and generation quality. These findings highlight the significance of fine-tuning sampling strategies to optimize T2M generation, balancing motion quality with computational efficiency.

Table 14: Quantitative results on MotionX++.

| Methods | R Precision↑ | | | FID↓ | MM Dist↓ | Diversity→ | MModality↑ |
|---|---|---|---|---|---|---|---|
| | Top 1 | Top 2 | Top 3 | | | | |
| Real | $0.578^{\pm0.003}$ | $0.767^{\pm0.005}$ | $0.856^{\pm0.004}$ | - | $2.452^{\pm0.002}$ | $13.263^{\pm0.227}$ | - |
| T2M-GPT(Discrete) | $0.486^{\pm0.004}$ | $0.673^{\pm0.007}$ | $0.786^{\pm0.009}$ | $0.686^{\pm0.014}$ | $3.228^{\pm0.032}$ | $12.132^{\pm0.132}$ | $2.306^{\pm0.042}$ |
| MoMask(Discrete) | $0.503^{\pm0.002}$ | $0.706^{\pm0.006}$ | $0.814^{\pm0.003}$ | $0.352^{\pm0.005}$ | $2.951^{\pm0.016}$ | $11.562^{\pm0.143}$ | $1.812^{\pm0.162}$ |
| MDM(Continuous) | $0.306^{\pm0.016}$ | $0.472^{\pm0.012}$ | $0.594^{\pm0.008}$ | $2.113^{\pm0.201}$ | $6.016^{\pm0.109}$ | $14.185^{\pm0.318}$ | $\mathbf{2.872^{\pm0.084}}$ |
| MLD(Continuous) | $0.456^{\pm0.013}$ | $0.652^{\pm0.008}$ | $0.743^{\pm0.003}$ | $0.906^{\pm0.068}$ | $3.493^{\pm0.036}$ | $12.481^{\pm0.263}$ | $2.634^{\pm0.076}$ |
| COME (Continuous) | $\mathbf{0.511^{\pm0.002}}$ | $\mathbf{0.722^{\pm0.003}}$ | $\mathbf{0.826^{\pm0.004}}$ | $\mathbf{0.268^{\pm0.004}}$ | $\mathbf{2.802^{\pm0.013}}$ | $12.829^{\pm0.116}$ | $2.516^{\pm0.068}$ |

Table 15: Quantitative results on MotionFix.

| Methods | Generated-to-Target Retrieval | | | | Generated-to-Source Retrieval | | | |
|---|---|---|---|---|---|---|---|---|
| | R@1 | R@2 | R@3 | AvgR | R@1 | R@2 | R@3 | AvgR |
| MDM | 4.03 | 7.56 | 10.48 | 15.55 | 2.62 | 6.15 | 9.38 | 15.88 |
| $MDM_s$ | 3.63 | 7.06 | 10.08 | 15.64 | 2.62 | 6.25 | 9.78 | 15.84 |
| $MDM\text{-}BP_s$ | 38.10 | 48.99 | 54.84 | 6.47 | 60.28 | 69.46 | 73.89 | 4.23 |
| MDM-BP | 39.10 | 50.09 | 54.84 | 6.46 | 61.28 | 69.55 | 73.99 | 4.21 |
| TMED | 62.90 | 76.51 | 83.06 | 2.71 | 71.77 | 84.07 | 89.52 | 1.96 |
| COME (Ours) | **73.82** | **86.76** | **89.76** | **2.16** | **72.42** | **84.23** | **89.63** | **1.87** |

Table 16: Quantitative results on PerMo.

| Methods | Steps | FID↓ | Top 1↑ | Top 2↑ | Top 3↑ | Age↑ | Char1↑ | Char2↑ | Cond1↑ | Cond2↑ | Emo1↑ | Emo2↑ | Trait↑ | Sur↑ | Avg.↑ | Diversity↑ |
|---|---|---|---|---|---|---|---|---|---|---|---|---|---|---|---|---|
| MoMo | 100 | 13.91 | 0.05 | 0.09 | 0.13 | 28.23 | 10.23 | 11.30 | **8.73** | 15.97 | 9.97 | 14.57 | 12.60 | 7.93 | 13.47 | 6.74 |
| MCM-LDM | 1000 | 9.16 | 0.13 | 0.23 | 0.30 | 47.38 | 11.44 | 10.34 | 7.63 | 18.90 | 11.69 | 17.20 | 15.71 | 9.22 | 17.00 | 6.70 |
| PersonaBooth | 50 | 3.18 | 0.15 | 0.26 | 0.33 | 48.00 | 13.67 | 11.97 | 6.69 | 18.67 | 14.34 | 19.75 | 17.06 | 8.80 | 18.05 | 7.74 |
| COME (Ours) | 10 | **2.92** | **0.18** | **0.29** | **0.36** | **49.33** | **14.95** | **13.42** | 8.32 | **19.94** | **15.82** | **20.92** | **18.31** | **9.94** | **19.58** | **8.85** |

Table 17: Insert into MotionLCM-v2.

| Method | R Precision↑ | | | FID↓ | MM Dist↓ | Diversity→ | MModality↑ |
|---|---|---|---|---|---|---|---|
| | Top 1 | Top 2 | Top 3 | | | | |
| MLD | $0.481^{\pm.003}$ | $0.673^{\pm.003}$ | $0.772^{\pm.002}$ | $0.473^{\pm.013}$ | $3.196^{\pm.010}$ | $9.724^{\pm.082}$ | $\mathbf{2.413^{\pm.079}}$ |
| MotionLCM | $0.502^{\pm.003}$ | $0.701^{\pm.002}$ | $0.803^{\pm.002}$ | $0.467^{\pm.012}$ | $3.022^{\pm.009}$ | $9.631^{\pm.066}$ | $2.172^{\pm.082}$ |
| MLD-v2 | $0.544^{\pm.003}$ | $0.736^{\pm.002}$ | $0.827^{\pm.002}$ | $0.049^{\pm.002}$ | $2.828^{\pm.007}$ | $9.531^{\pm.087}$ | $1.672^{\pm.051}$ |
| MotionLCM-v2 | $0.546^{\pm.003}$ | $0.743^{\pm.002}$ | $0.837^{\pm.002}$ | $0.072^{\pm.003}$ | $2.767^{\pm.007}$ | $9.577^{\pm.070}$ | $1.858^{\pm.056}$ |
| MLD-v2+ Stable | $0.548^{\pm.002}$ | $0.739^{\pm.002}$ | $0.834^{\pm.002}$ | $0.044^{\pm.003}$ | $2.806^{\pm.006}$ | $9.552^{\pm.084}$ | $1.743^{\pm.046}$ |
| MotionLCM-v2+Stable | $0.551^{\pm.003}$ | $0.748^{\pm.002}$ | $0.845^{\pm.002}$ | $0.064^{\pm.002}$ | $2.742^{\pm.006}$ | $9.592^{\pm.068}$ | $1.892^{\pm.052}$ |
| MLD-v2+ More | $0.558^{\pm.003}$ | $0.751^{\pm.002}$ | $0.846^{\pm.002}$ | $\mathbf{0.039^{\pm.002}}$ | $2.734^{\pm.005}$ | $9.627^{\pm.063}$ | $1.934^{\pm.058}$ |
| MotionLCM-v2 + More | $\mathbf{0.563^{\pm.004}}$ | $\mathbf{0.758^{\pm.003}}$ | $\mathbf{0.852^{\pm.002}}$ | $0.048^{\pm.003}$ | $\mathbf{2.716^{\pm.006}}$ | $9.653^{\pm.071}$ | $1.982^{\pm.053}$ |

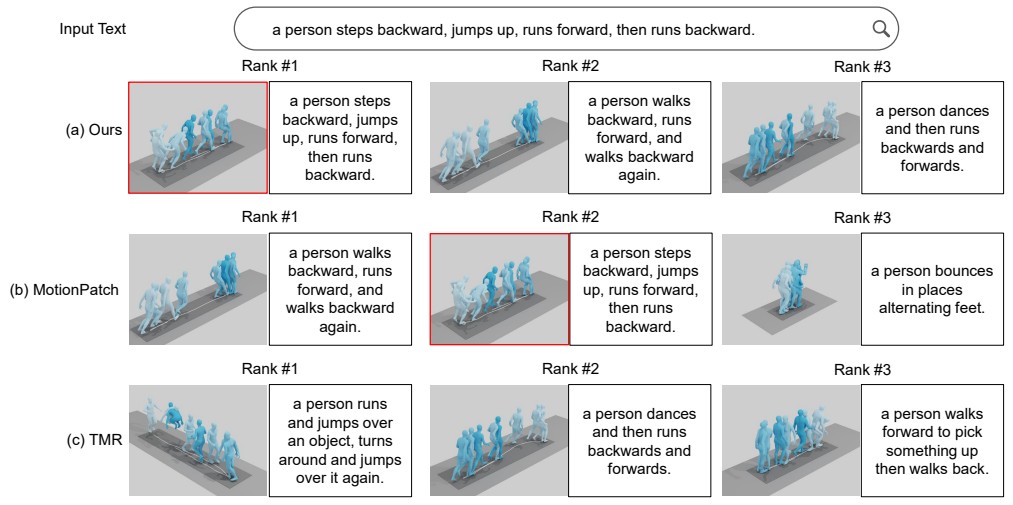

Figure 9: The text to motion retrieval results. The red box means the right sample.

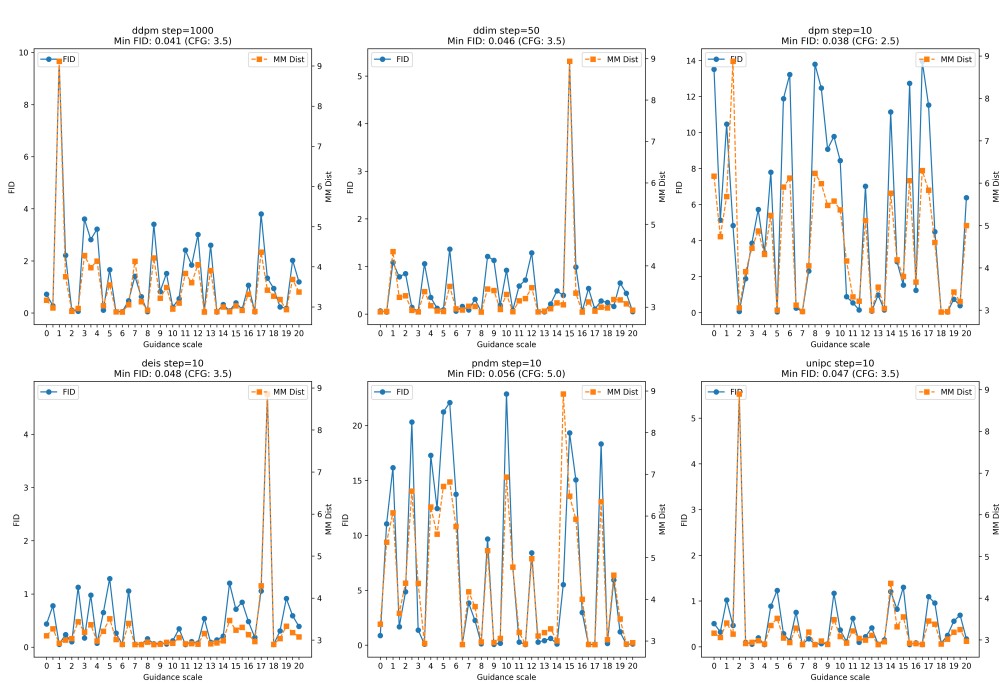

Figure 10: The effect of different CFG scales.

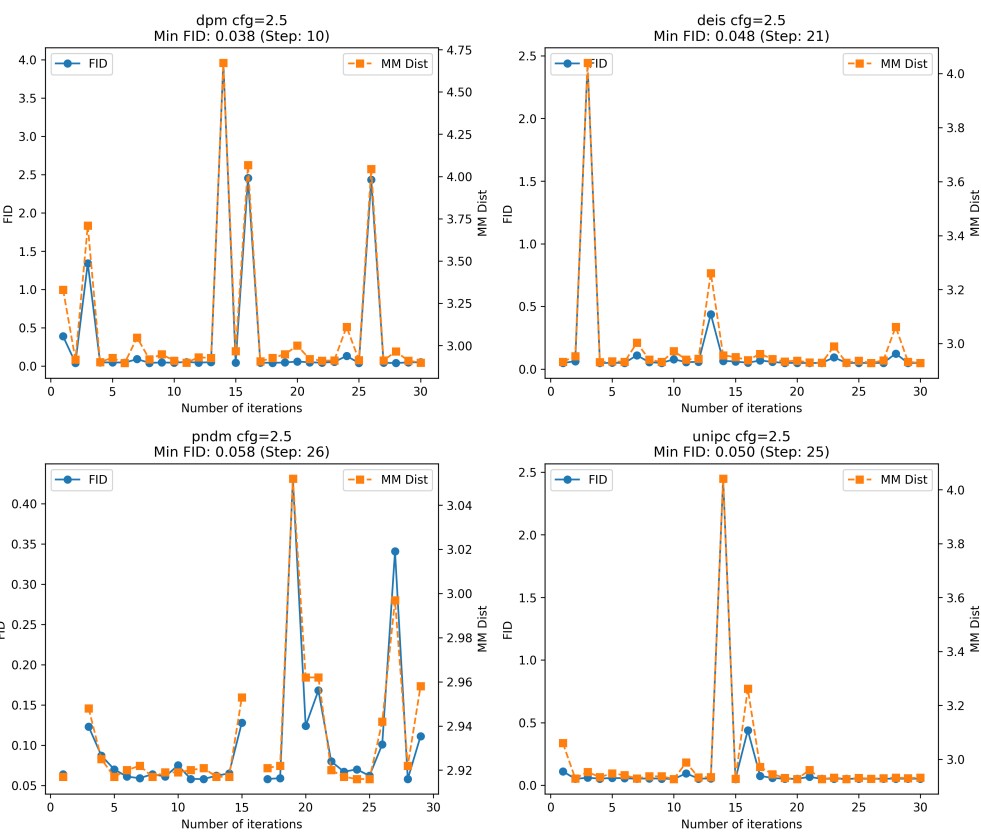

Figure 11: The effect of different iterations. Note that some iteration steps in PNDM led to errors, which are unrelated to our method.

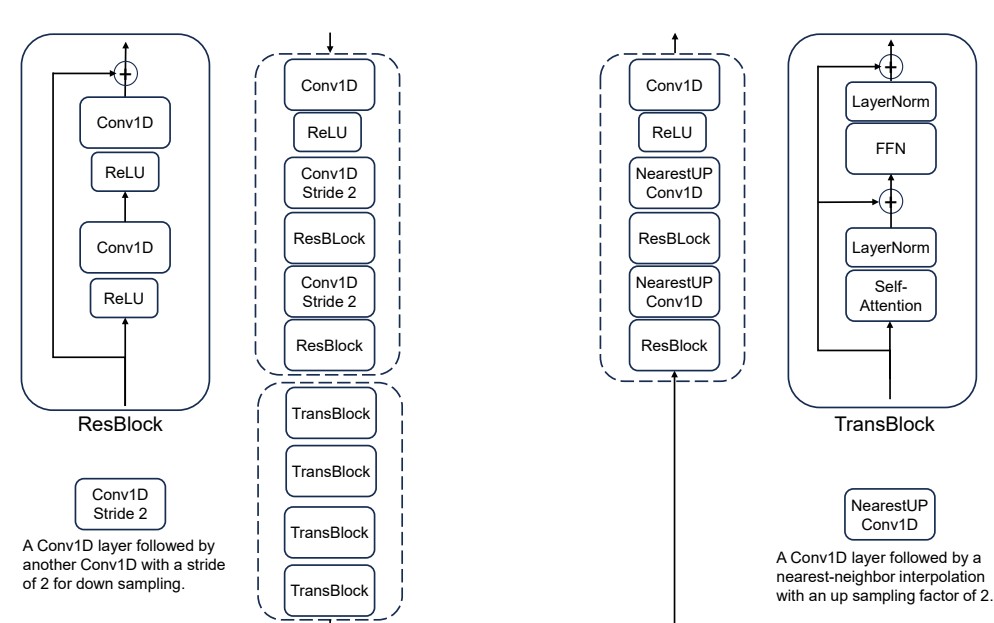

Figure 12: Detailed architecture of MoCMAE. The model adopts an asymmetric design comprising 1D convolutions (Conv1D), residual blocks (ResBlock), ReLU activations, and standard Transformer blocks (TranBlock). The encoder combines convolutional layers with Transformer blocks to extract expressive and compact spatio-temporal representations, while the decoder uses only convolutional layers to maintain fast and stable reconstruction without increasing decoding overhead during generation. Temporal downsampling and upsampling are performed using strided convolutions and nearest-neighbor interpolation, respectively.

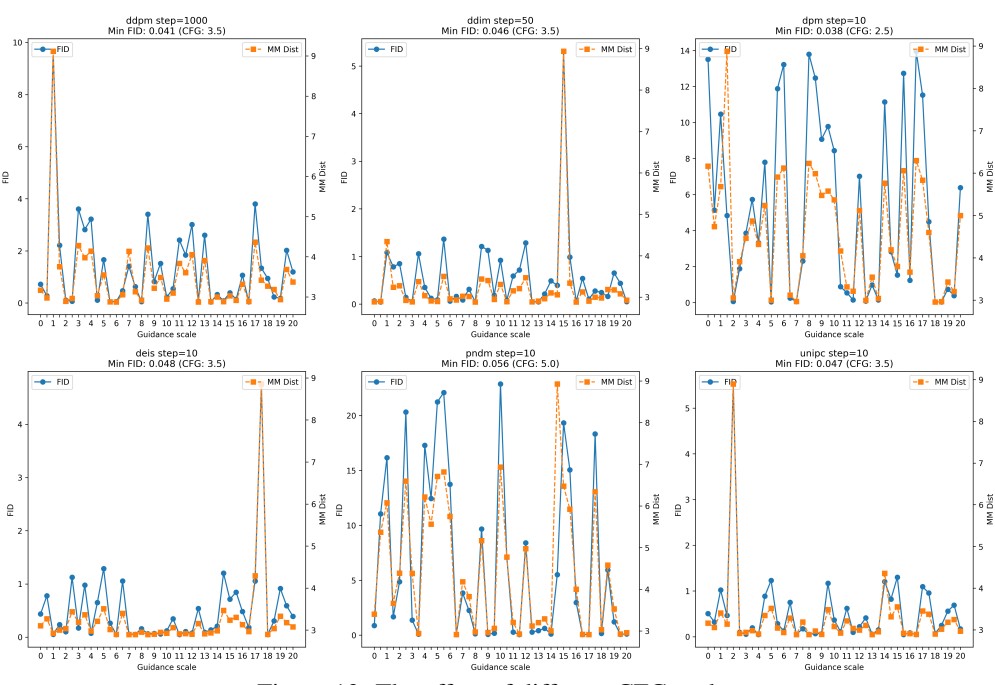

Figure 13: The effect of different CFG scales.

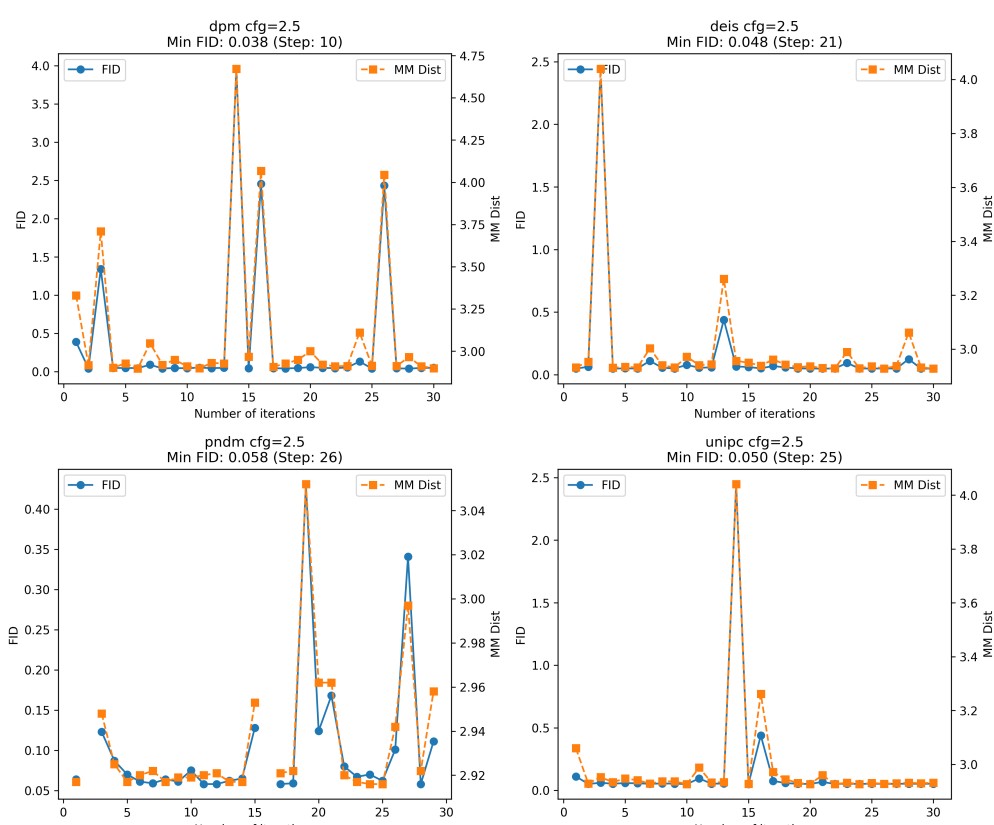

Figure 14: The effect of different iterations. Note that some iteration steps in PNDM led to errors, which are unrelated to our method.

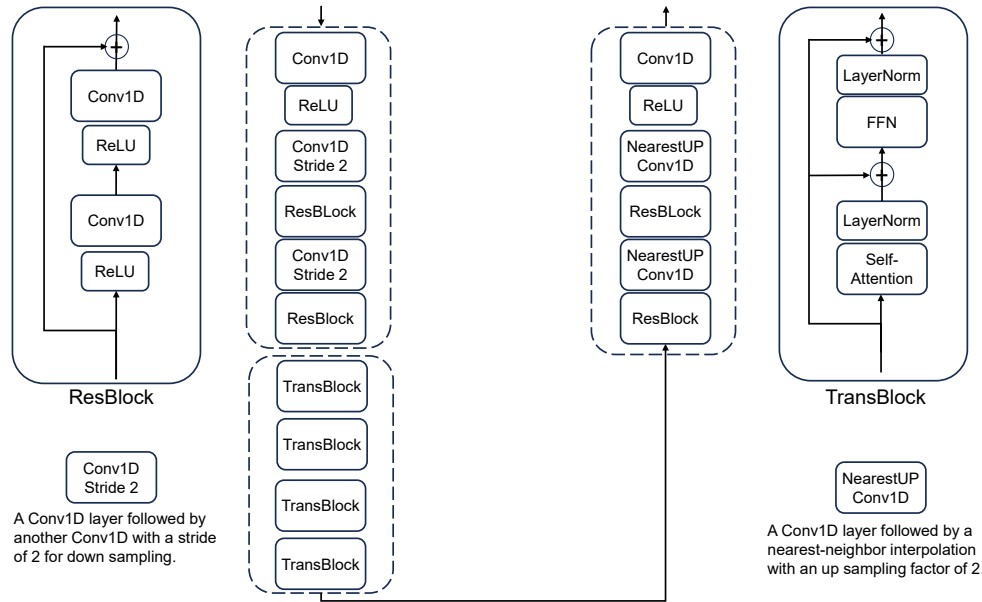

Figure 15: Detailed architecture of MoCMAE. The model adopts an asymmetric design comprising 1D convolutions (Conv1D), residual blocks (ResBlock), ReLU activations, and standard Transformer blocks (TranBlock). The encoder combines convolutional layers with Transformer blocks to extract expressive and compact spatio-temporal representations, while the decoder uses only convolutional layers to maintain fast and stable reconstruction without increasing decoding overhead during generation. Temporal downsampling and upsampling are performed using strided convolutions and nearest-neighbor interpolation, respectively.

# B    REBUTTAL

## B.1    MORE DETAILS OF EVALUATION METRICS

**Text-to-Motion Generation**    We follow standard metrics proposed by T2M (Guo et al., 2022a). **Fréchet Inception Distance (FID)** measures the distributional difference between the high-level features of generated and real motions, reflecting overall motion quality. **R-Precision** and **Multi-modal Distance (MM-Dist)** evaluate semantic alignment between input text and generated motions, with R-Precision reporting Top-1, Top-2, and Top-3 retrieval accuracies. **Diversity** is computed as the average Euclidean distance among 300 randomly sampled motion pairs, indicating variability in generated motions. **Multimodality (MModality)** evaluates the variance of motions generated from the same text prompt.

**Controllable Text-to-Motion Generation**    Following prior works such as **MotionLCM** (Dai et al., 2025), **OmniControl** (Xie et al., 2024), and **MaskControl** (Pinyoanuntapong et al., 2025a), we adopt similar experimental settings for controllable motion generation. We use **Pelvis trajectory** as the primary control signal and evaluate controllability with the following metrics:

- **Trajectory Error (Traj. err.)**: the ratio of unsuccessful trajectories. A trajectory is considered unsuccessful if the position of any control joint exceeds a predefined threshold compared with the reference trajectory.
- **Location Error (Loc. err.)**: counts the number of control joints that fail to follow the desired trajectory within the allowed threshold.
- **Average Error (Avg. err.)**: the mean Euclidean distance error of all control joints across the trajectory, providing an overall measure of tracking performance.

These metrics are consistent with prior works, allowing fair comparisons of controllability performance across methods. They capture both the frequency of failure (Traj. err. and Loc. err.) and the magnitude of deviation (Avg. err.), providing a comprehensive assessment of controllable generation quality.

**Motion Editing**    We follow the **MotionFix** (Athanasiou et al., 2024) experimental setup and adopt motion-to-motion retrieval metrics to evaluate motion editing. Distance-based metrics are unreliable in this setting because multiple plausible motions can correspond to the same text. Therefore, we rely on retrieval-based evaluation:

- **Generated-to-Source Retrieval**: measures how well the edited motion preserves characteristics of the original (source) motion.
- **Generated-to-Target Retrieval**: evaluates how accurately the edited motion aligns with the intended target motion.

Feature embeddings are extracted using **TMR** (Petrovich et al., 2023), retrained on **HumanML3D** to be consistent with our representation. Standard metrics reported include **Recall at rank** $k$ **(R@1, R@2, R@3)** and **Average Recall (AvgR)**, using a gallery size of 32 randomly sampled motions per test batch. This evaluation protocol aligns with previous works and ensures fair, reproducible, and comparable assessment of motion editing quality.

Table 18: **Quantitative comparison of motion representation quality across encoders.** We report both internal geometric metrics (Silhouette Score, Calinski–Harabasz Index, Davies–Bouldin Index) and an external semantic metric (5-NN accuracy). It is worth noting that human motion categories naturally overlap and do not form well-separated clusters; the same action can appear under different textual descriptions, and simple keyword-based pseudo-labels offer only coarse semantic grouping. As a result, the absolute values of the Silhouette Score are close to zero across all encoders, which is expected in this setting. Despite this challenging structure, MoCMAE achieves the strongest performance across nearly all metrics. It attains the highest SC and 5-NN accuracy and the lowest DBI, demonstrating that it learns a latent space with better geometric separability and stronger alignment with semantic distinctions compared to existing encoders.

| Model | SC ↑ | CHI ↑ | DBI ↓ | 5-NN Acc ↑ |
|---|---|---|---|---|
| VQ-VAE | -0.086093 | 5.42 | 4.241479 | 0.4667 |
| RVQ-VAE | -0.020365 | 7.83 | 4.024332 | 0.5197 |
| VAE | -0.008453 | 1.38 | 6.662875 | 0.3125 |
| **MoCMAE** | **0.047418** | 5.23 | **3.984182** | **0.7250** |
| MoCMAE-w/o CL | 0.025364 | 4.28 | 4.086264 | 0.6425 |

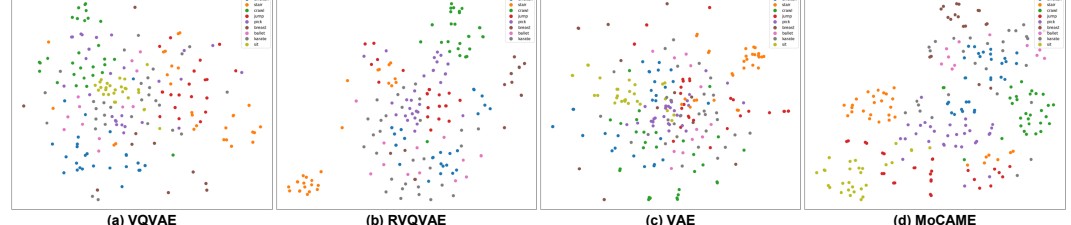

(a) VQVAE     (b) RVQVAE     (c) VAE     (d) MoCAME

Figure 16: **t-SNE visualization using keyword-based pseudo-labels.** Human motion data naturally form continuous and overlapping manifolds. Nonetheless, MoCMAE exhibits visibly more coherent local groupings and clearer semantic regions than baseline encoders. This plot is used as a qualitative reference, while our conclusions rely primarily on the quantitative metrics in Table 18.

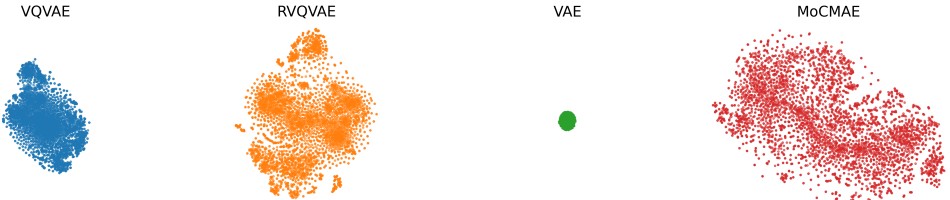

Figure 17: **Global t-SNE visualization over the full HumanML3D dataset.** Continuous encoders such as VAE produce highly compact and densely concentrated feature regions, indicating a limited coverage of the motion manifold. Discrete models (VQ-VAE, RVQ-VAE) spread out more than VAE but still occupy a relatively restricted area. In contrast, MoCMAE yields a substantially broader and more dispersed global distribution, covering a larger portion of the motion space and exhibiting greater feature diversity. As with Figure 16, this visualization is provided as qualitative reference, while the core conclusions rely on the quantitative metrics in Table 18.

## B.2 COMPREHENSIVE EVALUATION OF MOTION REPRESENTATION QUALITY

To systematically assess the quality of the learned latent motion representations, we perform a comprehensive evaluation combining internal geometric metrics, external semantic metrics, and qualitative visualization. Our aim is to determine whether the latent space learned by **MoCMAE** exhibits superior structural organization and semantic coherence compared with existing continuous and discrete encoders.

**Evaluation Protocol.** We adopt two complementary categories of metrics:

- **Internal metrics:** Silhouette Score (SC), Calinski–Harabasz Index (CHI), and Davies–Bouldin Index (DBI), which quantify the geometric structure and compactness of the latent space without requiring labels.
- **External metrics:** semantic alignment is evaluated via the five-nearest-neighbor (5-NN) classification accuracy. Pseudo-labels are constructed using nine motion-related keywords ("chicken", "stair", "crawl", "jump", "pick", "breast stroke", "ballet", "karate", "sit"), with randomly sampled instances per class to ensure a balanced evaluation set.

**Quantitative Analysis.** As shown in Table 18, human motion categories naturally overlap and can be described in multiple ways, making keyword-based grouping only a coarse approximation. As a result, absolute SC values remain close to zero across all encoders. Despite this inherent difficulty, **MoCMAE** consistently outperforms other continuous and discrete encoders on both internal and external metrics. It achieves the highest Silhouette Score (SC) and 5-NN accuracy, as well as the lowest Davies–Bouldin Index (DBI), indicating that the learned latent space is simultaneously more geometrically coherent and semantically meaningful. These results quantitatively confirm the superiority of **MoCMAE** in capturing structured and semantically aligned motion representations.

**Qualitative Visualization.** To complement the quantitative evaluation, we visualize the learned motion representations using t-SNE (Figures 16 and 17). Although human motion inherently forms overlapping and continuous manifolds, **MoCMAE** representations exhibit broader coverage and more coherent local groupings compared with baseline encoders. This wider distribution not only reflects richer latent diversity but also facilitates more effective diffusion-based denoising and higher-fidelity motion generation. Together with the quantitative metrics, these visualizations qualitatively corroborate that **MoCMAE** produces a more informative and well-distributed latent space.

### B.3 Discussion on Motion Encoder Design

The reviewer's suggestion is very valuable, and we will include a more systematic discussion of related work in the final version. In fact, the design of motion encoders has been largely overlooked in recent years. Most methods still adopt early VQ-VAE, VAE, or RVQ architectures, with minimal modifications to the encoder. This stagnation limits the generation quality of diffusion models and also affects downstream tasks such as TMR. Many works directly reuse the VQ-VAE encoder to extract motion features and perform contrastive learning with CLIP text embeddings, without optimizing the motion encoder itself.

Only a few TMR-focused works proposed novel solutions. For example, Guangtao Lyu et al. (Lyu et al., 2025) modeled motion–text alignment using a sparse vocabulary space and employed a 2D skeleton-aware encoder to capture spatio-temporal structure. SALAD (Hong et al., 2025) proposed a 2D skeleton-aware encoder incorporating convolution and pooling to enhance structural modeling, but its inference cost is high and subsequent adoption is limited, appearing only in a few methods such as MaskControl.

The stagnation in motion encoder research mainly stems from two factors:

1. The community has long focused on text-to-motion generation;
2. Caption-style annotations are not conducive to classification or disentangled representations, and the lack of large-scale classification datasets similar to ImageNet makes systematic representation learning difficult.

In this work, we revisit this overlooked core problem. Based on systematic analysis, we find that existing encoders exhibit significant deficiencies in representation quality. To address this, we propose a simple yet effective motion representation learning framework that combines masked motion modeling with contrastive learning.

Through generation performance, reconstruction performance, t-SNE visualization, transfer experiments on TMR, and quantitative latent representation metrics such as Silhouette Score, Calinski–Harabasz Index, Davies–Bouldin Index, and 5-NN classification accuracy, we demonstrate that stronger motion representations significantly improve both diffusion model generation quality and downstream task performance. These results show that integrating discriminative representation learning into motion encoders can effectively enhance generative models.

This is conceptually similar to the core idea of the recent contemporaneous work by Saining Xie et al. (RAE) (Zheng et al., 2025), which replaces the VAE with a pretrained autoencoder to improve latent representations in image generation. The key difference is that, unlike images where pretrained autoencoders are available, in motion generation we train the autoencoder from scratch.

We believe this study highlights an important direction: improving the motion encoder itself is as crucial as refining the diffusion model architecture and is a key avenue for advancing motion generation and multimodal task performance.

### B.4 COMPARISON WITH ADDITIONAL STRONG BASELINES

In the original submission, comparisons with recent strong baselines such as MARDM and MotionStreamer were not included due to differences in data preprocessing and evaluation protocols (MotionLCM v2 results are already reported in Tab. 17). Here, we provide a comprehensive comparison under the respective standard settings of these methods.

Following MARDM's setup, we adopt their motion representation instead of the standard 263-dimensional feature space, and employ their evaluation metrics and protocols. As shown in Table 19 and Table 20, our method, **COME**, consistently achieves state-of-the-art performance on both HumanML3D and KIT datasets, while being approximately 100× faster than MARDM. In terms of generation quality, inference speed, and training efficiency, COME surpasses both continuous and discrete baselines, effectively realizing the goal of reviving diffusion-based motion models. In contrast, MARDM achieves results comparable to MoMask but suffers from significantly lower efficiency, reflecting an initial exploration rather than a full revival.

Similarly, following MotionStreamer's setup, we adopt their motion representation and evaluation protocol. Table 21 shows that COME outperforms MotionStreamer across multiple metrics on HumanML3D, including FID, R-Precision, multimodal distance, and diversity.

Compared with StableMoFusion, our method achieves 1.6× lower FID and 7.3× faster inference, surpassing it in both quality and efficiency. Moreover, COME generates high-quality results directly, without requiring vGRFs as a post-processing step for footskate reduction.

For control-based tasks, we further compare with **MaskControl** under their standard settings (Table 22), evaluating both inference efficiency and motion quality. MaskControl, similar to GMD, relies on iterative optimization to gradually align generated motions to the target control joints. While this allows near-zero joint matching error, it comes at the cost of extremely slow inference, being approximately 1,000 times slower than COME (71.72s compared to 0.058s per sequence). Even when the iteration steps are reduced to 100 (MaskControl-Fast), it still underperforms COME on most metrics and remains roughly 100 times slower (4.94s compared to 0.058s). In contrast, our approach, akin to MotionLCM, directly conditions on the input control joints to generate motions in a single forward pass, achieving substantially higher efficiency. Notably, COME also improves trajectory and local accuracy while maintaining high fidelity and multimodal diversity.

Overall, these results demonstrate that **COME** consistently outperforms recent strong SOTA methods in both text-driven and control-based motion generation, achieving significant improvements in generation quality, multimodal diversity, and inference efficiency. Importantly, COME effectively realizes the full potential of diffusion-based motion models, marking a true revival of this paradigm in motion generation research.

### B.5 EFFECT OF ASYMMETRIC VS. SYMMETRIC MOCMAE DESIGNS

To investigate the impact of decoder design on reconstruction quality, generation performance, and inference efficiency, we compare the original asymmetric MoCMAE with a symmetric variant, MoCMAE*, in which Transformer blocks are also added to the decoder. The goal is to understand whether a stronger decoder can improve latent representation utilization or whether it primarily benefits reconstruction at the expense of generation quality and speed.

**Experimental Setup.** We evaluate both models on reconstruction metrics (FID, MPJPE), generation metrics (FID, MM-Dist), and inference speed. All other model components and training settings are kept identical to ensure a fair comparison.

Table 19: **Quantitative evaluation on HumanML3D and KIT-ML datasets following the MARDM setup.** We repeat each evaluation 20 times and report the average with 95% confidence intervals. **Bold** indicates the best result, while underlined indicates the second best. COME consistently achieves superior performance across R-Precision, FID, Matching, MModality, and CLIP-score, demonstrating both high generation quality and multimodal diversity compared to prior VQ and diffusion-based methods.

| | Methods | Framework | R-Precision↑ | | | FID↓ | Matching↓ | MModality↑ | CLIP-score↑ |
|---|---|---|---|---|---|---|---|---|---|
| | | | Top 1 | Top 2 | Top 3 | | | | |
| HumanML3D | T2M-GPT | VQ | $0.470^{\pm.003}$ | $0.659^{\pm.002}$ | $0.758^{\pm.002}$ | $0.335^{\pm.003}$ | $3.505^{\pm.017}$ | $2.018^{\pm.053}$ | $0.607^{\pm.005}$ |
| | MMM | | $0.487^{\pm.003}$ | $0.683^{\pm.002}$ | $0.782^{\pm.001}$ | $0.132^{\pm.004}$ | $3.359^{\pm.009}$ | $1.241^{\pm.073}$ | $0.635^{\pm.003}$ |
| | MoMask | | $0.490^{\pm.004}$ | $0.687^{\pm.003}$ | $0.786^{\pm.003}$ | $0.116^{\pm.006}$ | $3.353^{\pm.010}$ | $1.263^{\pm.079}$ | $0.637^{\pm.003}$ |
| | MDM-50Step | Diffusion | $0.440^{\pm.007}$ | $0.636^{\pm.006}$ | $0.742^{\pm.004}$ | $0.518^{\pm.032}$ | $3.640^{\pm.028}$ | $3.604^{\pm.031}$ | $0.578^{\pm.003}$ |
| | MotionDiffuse | | $0.450^{\pm.006}$ | $0.641^{\pm.005}$ | $0.753^{\pm.005}$ | $0.778^{\pm.005}$ | $3.490^{\pm.023}$ | $3.179^{\pm.046}$ | $0.606^{\pm.004}$ |
| | MLD | | $0.461^{\pm.004}$ | $0.651^{\pm.004}$ | $0.750^{\pm.003}$ | $0.431^{\pm.014}$ | $3.445^{\pm.019}$ | $3.506^{\pm.031}$ | $0.610^{\pm.003}$ |
| | ReMoDiffuse | | $0.468^{\pm.003}$ | $0.653^{\pm.003}$ | $0.754^{\pm.005}$ | $0.883^{\pm.021}$ | $3.414^{\pm.020}$ | $2.703^{\pm.154}$ | $0.621^{\pm.003}$ |
| | COME(ours) | | $\mathbf{0.508^{\pm.005}}$ | $\mathbf{0.699^{\pm.005}}$ | $\mathbf{0.798^{\pm.004}}$ | $\mathbf{0.074^{\pm.004}}$ | $\mathbf{3.246^{\pm.010}}$ | $\mathbf{3.762^{\pm.051}}$ | $\mathbf{0.648^{\pm.004}}$ |
| | MARDM-DDPM | Autoregressive | $0.492^{\pm.006}$ | $0.690^{\pm.005}$ | $0.790^{\pm.005}$ | $0.116^{\pm.004}$ | $3.349^{\pm.010}$ | $2.470^{\pm.053}$ | $0.637^{\pm.005}$ |
| | MARDM-SiT | Diffusion | $0.500^{\pm.004}$ | $0.695^{\pm.003}$ | $0.795^{\pm.003}$ | $0.114^{\pm.007}$ | $3.270^{\pm.009}$ | $2.231^{\pm.071}$ | $0.642^{\pm.002}$ |
| KIT | T2M-GPT | VQ | $0.359^{\pm.007}$ | $0.553^{\pm.007}$ | $0.690^{\pm.013}$ | $0.593^{\pm.053}$ | $3.765^{\pm.046}$ | $1.798^{\pm.157}$ | $0.651^{\pm.005}$ |
| | MMM | | $0.363^{\pm.006}$ | $0.569^{\pm.006}$ | $0.724^{\pm.006}$ | $0.478^{\pm.034}$ | $3.629^{\pm.028}$ | $1.455^{\pm.106}$ | $0.660^{\pm.003}$ |
| | MoMask | | $0.369^{\pm.005}$ | $0.588^{\pm.005}$ | $0.731^{\pm.005}$ | $0.411^{\pm.026}$ | $3.577^{\pm.021}$ | $1.309^{\pm.058}$ | $0.669^{\pm.002}$ |
| | MDM | Diffusion | $0.333^{\pm.012}$ | $0.561^{\pm.009}$ | $0.689^{\pm.009}$ | $0.585^{\pm.043}$ | $4.002^{\pm.033}$ | $1.681^{\pm.107}$ | $0.605^{\pm.007}$ |
| | MotionDiffuse | | $0.344^{\pm.009}$ | $0.536^{\pm.007}$ | $0.658^{\pm.007}$ | $3.845^{\pm.087}$ | $4.167^{\pm.054}$ | $1.774^{\pm.217}$ | $0.626^{\pm.006}$ |
| | MLD | | $0.351^{\pm.007}$ | $0.536^{\pm.007}$ | $0.658^{\pm.007}$ | $0.492^{\pm.047}$ | $3.746^{\pm.044}$ | $1.803^{\pm.164}$ | $0.646^{\pm.006}$ |
| | ReMoDiffuse | | $0.356^{\pm.004}$ | $0.572^{\pm.007}$ | $0.706^{\pm.009}$ | $1.725^{\pm.053}$ | $3.735^{\pm.036}$ | $1.928^{\pm.127}$ | $0.665^{\pm.005}$ |
| | COME(ours) | | $\mathbf{0.396^{\pm.005}}$ | $\mathbf{0.608^{\pm.006}}$ | $\mathbf{0.758^{\pm.005}}$ | $\mathbf{0.208^{\pm.012}}$ | $\mathbf{3.206^{\pm.016}}$ | $\mathbf{2.062^{\pm.072}}$ | $\mathbf{0.698^{\pm.003}}$ |
| | MARDM-DDPM | Autoregressive | $0.375^{\pm.006}$ | $0.597^{\pm.008}$ | $0.739^{\pm.006}$ | $0.340^{\pm.020}$ | $3.489^{\pm.018}$ | $1.479^{\pm.078}$ | $0.681^{\pm.003}$ |
| | MARDM-SiT | Diffusion | $0.387^{\pm.006}$ | $0.610^{\pm.006}$ | $0.749^{\pm.006}$ | $0.242^{\pm.014}$ | $3.374^{\pm.019}$ | $1.312^{\pm.053}$ | $0.692^{\pm.002}$ |

Table 20: **Average Inference Time Comparison** between our method and baselines under the MARDM setup. COME achieves the fastest inference speed, substantially outperforming prior diffusion-based methods including MARDM, MDM, and MotionDiffuse, highlighting its efficiency advantage in practical deployment.

| Methods | MDM | MotionDiffuse | T2M-GPT | MLD | MMM | MoMask | MARDM | COME(ours) |
|---|---|---|---|---|---|---|---|---|
| AIT | 14.31s | 7.35s | 0.32s | 0.21s | 0.06s | 0.045s | 2.4s | 0.022s |

**Results and Analysis.** Table 24 summarizes the results. Adding Transformer blocks to the decoder slightly improves reconstruction metrics (FID decreases from 0.0024 to 0.0021, MPJPE from 8.8 to 8.2). However, generation quality metrics slightly degrade (FID increases from 0.041 to 0.049, MM-Dist from 2.898 to 2.912), and inference speed is marginally slower (0.041 s vs. 0.043 s). This indicates that a stronger decoder can reconstruct masked motion sequences more accurately but reduces the encoder's pressure to learn high-quality latent representations, which slightly impairs generation performance.

**Conclusion.** The asymmetric design achieves a superior trade-off, maintaining high generation quality while being slightly faster. The symmetric variant mainly benefits reconstruction but does not offer practical improvements for generative motion tasks.

### B.6 USER STUDY ON DIFFERENT BENCHMARKS

To further validate the perceptual quality of our generated motions across different benchmarks, we conduct an extended human user study. Participants evaluate motion naturalness, alignment with text prompts, and overall quality by comparing outputs from our method against several baseline approaches on two datasets: HumanML3D and KIT-ML. Each participant ranks the motions according to the three criteria.

**Results.** Figures 18 and 19 present the aggregated results. On both datasets, Our method consistently achieves the highest preference scores across all evaluation criteria. These findings indicate that the improvements in latent representation and generation fidelity not only improve quantitative metrics but also translate to perceptually superior motions preferred by human observers.

Table 21: **Quantitative comparison on HumanML3D following the MotionStreamer setup.**
MM-D and Div denote Multimodal Distance and Diversity, respectively. COME achieves the best results across all metrics, including FID, R-Precision, MM-D, and Div, demonstrating superior fidelity, relevance to text prompts, and motion diversity compared to MotionStreamer and other baselines.

| Methods | FID ↓ | R@1 ↑ | R@2 ↑ | R@3 ↑ | MM-D ↓ | Div → |
|---|---|---|---|---|---|---|
| Real motion | 0.002 | 0.702 | 0.864 | 0.914 | 15.151 | 27.492 |
| MDM | 23.454 | 0.523 | 0.692 | 0.764 | 17.423 | 26.325 |
| MLD | 18.236 | 0.546 | 0.730 | 0.792 | 16.638 | 26.352 |
| T2M-GPT | 12.475 | 0.606 | 0.774 | 0.838 | 16.812 | 27.275 |
| MotionGPT | 14.375 | 0.456 | 0.598 | 0.628 | 17.892 | 27.114 |
| MoMask | 12.232 | 0.621 | 0.784 | 0.846 | 16.138 | 27.127 |
| AttT2M | 15.428 | 0.592 | 0.765 | 0.834 | 15.726 | 26.674 |
| MotionStreamer | 11.790 | 0.631 | 0.802 | 0.859 | 16.081 | 27.284 |
| COME(ours) | **9.684** | **0.653** | **0.819** | **0.872** | **15.462** | **27.362** |

Table 22: Control motion generation performance comparison: speed and quality metrics following MaskControl. COME achieves a strong balance between accuracy and efficiency: it outperforms MaskControl-Fast on most metrics while being roughly 100× faster, and maintains competitive accuracy compared to MaskControl-Accurate while being over 1,000× faster. This highlights COME's advantage in generating high-quality controlled motions efficiently.

| Methods | Speed ↓ | FID ↓ | R-Precision Top3 ↑ | Diversity → | Traj. err. ↓ | Loc. err. ↓ | Avg. err. ↓ |
|---|---|---|---|---|---|---|---|
| Real motion | – | 0.002 | 0.797 | 9.503 | 0.000 | 0.000 | 0.000 |
| MDM | 10.14 s | 0.698 | 0.602 | 9.197 | 0.402 | 0.308 | 0.596 |
| PriorMDM | 18.11 s | 0.475 | 0.583 | 9.156 | 0.346 | 0.213 | 0.442 |
| GMD | 132.49 s | 0.576 | 0.665 | 9.206 | 0.093 | 0.032 | 0.144 |
| OmniControl | 87.33 s | 0.218 | 0.687 | 9.422 | 0.039 | 0.010 | 0.034 |
| MotionLCM | **0.051 s** | 0.531 | 0.752 | 9.253 | 0.189 | 0.077 | 0.190 |
| **COME(ours)** | 0.058 s | 0.112 | 0.782 | **9.498** | 0.020 | 0.007 | 0.011 |
| MaskControl-Fast | 4.94 s | 0.059 | 0.808 | 9.444 | 0.057 | 0.020 | 0.055 |
| MaskControl-Accurate | 71.72 s | **0.061** | **0.809** | 9.496 | 0.055 | **0.000** | **0.0098** |

## B.7 DETAILS OF TRAINING

We compare the training efficiency of COME with prior methods. Table 25 reports epochs and GPU-hours for each module (motion tokenizer and diffusion model), as well as total GPU-hours, measured on a single NVIDIA A6000 GPU. COME converges in only 1,100 epochs (500 for MoCMAE and 600 for ccDiT), substantially fewer than MLD's 9,000 epochs (6,000 for VAE pretraining and 3,000 for diffusion). This reduces the total training cost from 302 GPU-hours to 49 GPU-hours, achieving a 6× improvement in efficiency while surpassing generation quality. These results further demonstrate the effectiveness and practicality of our approach.

Table 23: Comparison with StableMoFusion in terms of generation quality and efficiency. COME achieves **1.6× lower FID** and **7.3× faster inference**, while producing results directly (without requiring vGRFs post-processing for footskate reduction).

| Method | FID↓ | R Precision (top3)↑ | AITS↓ | Inference Steps↓ |
|---|---|---|---|---|
| StableMoFusion-base (DDPM1000) | 1.251 | 0.760 | 99.060 s | 1000 |
| + Efficient Sampler | 0.076 | 0.836 | 1.004 s | 10 |
| + Embedded-text Cache | 0.076 | 0.836 | 0.690 s | 10 |
| + Parallel CFG | 0.076 | 0.836 | 0.544 s | 10 |
| + FP16 | 0.076 | **0.837** | 0.499 s | 10 |
| COME(ours) | **0.041** | 0.816 | **0.068 s** | 10 |

Table 24: Comparison of asymmetric and symmetric MoCMAE designs. MoCMAE* adds Transformer blocks to the decoder. While reconstruction improves slightly, generation quality decreases modestly and inference is slightly slower. The asymmetric design maintains the best balance between generation fidelity and efficiency.

| Methods | Reconstruction | | Generation | | Speed (s)↓ |
|---|---|---|---|---|---|
| | FID↓ | MPJPE↓ | FID↓ | MM-Dist↓ | |
| MoCMAE | 0.0024 | 8.8 | 0.041 | 2.898 | 0.041 |
| MoCMAE* | 0.0021 | 8.2 | 0.049 | 2.912 | 0.043 |

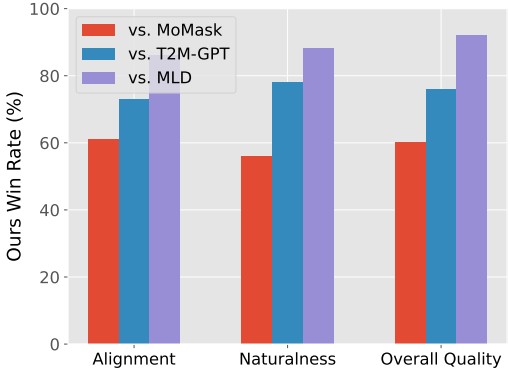

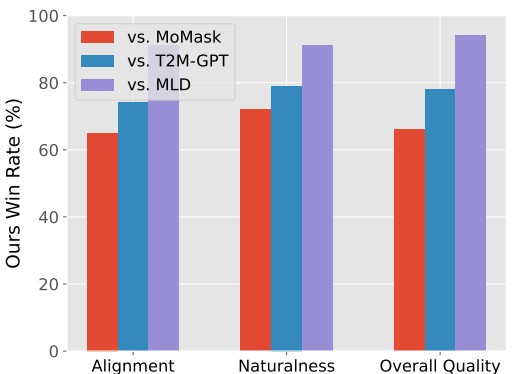

Figure 18: User study results on HumanML3D. Our method receives the highest preference across alignment, naturalness, and overall quality, demonstrating perceptual improvements over baseline methods.

Figure 19: User study results on KIT-ML. Similar to HumanML3D, our method is consistently preferred by participants across all criteria, confirming its perceptual advantages on multiple benchmarks.

| Method | Training Stage | Epochs | GPU-hours | Total GPU-hours |
|---|---|---|---|---|
| **COME (Ours)** | Motion Tokenizer (MoCMAE) | 600 | 18.1 | **49.3** |
| | Diffusion Model (ccDiT) | 500 | 31.2 | |
| **MLD** | Motion Tokenizer (VAE) | 6000 | 196.8 | **301.9** |
| | Diffusion Model | 3000 | 105.1 | |

Table 25: Training cost comparison across methods. We report epochs and GPU-hours for each module (motion tokenizer and diffusion model) and the total GPU-hours using a single NVIDIA A6000 GPU. Our method converges with significantly fewer epochs and GPU-hours compared to MLD, achieving a 6× improvement in training efficiency.

