# OpenReview forum: "COME: Advancing Representation Learning and Generative Modeling for High-Quality Text-to-Motion Generation"
_ICLR.cc/2026/Conference — Submitted to ICLR 2026_

### Official Review · Reviewer_dQuQ · 2025-10-29

**Soundness:** 2
**Presentation:** 1
**Contribution:** 2
**Rating:** 2
**Confidence:** 5

**Summary:**

This paper proposes COME, a continuous diffusion framework for text-to-motion generation. COME includes a motion contrastive masked autoencoder (MoCMAE) serves as a continuous motion tokenizer with better inter-sample separability, and a cross-condition diffusion transformer(ccDIT) which specifically align sentence-level semantics and word-level semantics through different modules. Besides, COME utilizes Stable-Min-SNR-gamma strategy to address training-inference inconsistencies.

**Strengths:**

1. Extensive quantitative results. The authors provide thorough quantitative experiments across multiple datasets and tasks, including HumanML3D, KIT-ML, BABEL, MotionX++, etc. The proposed method COME achieves state-of-the-art quantitative results across standard T2M benchmarks, comparing with both continuous and discrete paradigms.

**Weaknesses:**

1. Insufficient technical novelty. The overall methodology design appears to be a naive ensemble of architectures and training strategies that have existed for quite a while, lacking clear task-driven motivation or theoretical insight.

      - MoCMAE directly employs the preliminary masking strategy from MAE [1] and combine with contrastive learning loss. This combination has been explored in earlier works (e.g., [2]) and is not novel by itself. The paper doesn't include necessary discussions on those similar contrastive masked autoencoder frameworks in previous works.

      - ccDIT have different level of text embeddings injected using AdaLN-Zero and cross attention layer, which is rather a straight-forward extension to existing transformer architecture in text-to-motion methods, without any further analysis nor insights. Comparable DiT-based or hybrid conditioning designs have already been adopted in prior motion generation methods [3–5]. The paper fails to provide sufficient analysis or justification of the claimed advantages or unique contributions of this architecture.

      - Stable-Min-SNR-gamma strategy appears to be a training technique borrowed and slightly revised from an existing diffusion-based image generation paper [6]. Yet, the authors fail to provide any further analysis nor discussion about the technical insights of this technique in the text-to-motion generation tasks specifically.

1. What makes it so important to highlight the asymmetric encoder-decoder architecture? What would be the changes in encoding performance and inference cost respectively, if adding the layers of decoder to make it symmetric comparing to the current MoCMAE design?

2. The details of the text features processing are not clear. How are the word-level language features encoded and processed? How does the word-level features separate from the sentence-level features?


2. The qualitative results in Figure 4 are difficult to interpret as the prompts that the authors provide contain a lots of adjectives that are ambiguous by itself (e.g., fast, high, slowly) without any reference or clear assessments, and the visualizations use frame shots that often overlap with each other.  It would be more convincing if the authors can provide 2-3 sets of video demos for generated samples from the existing methods and from the proposed method for comparison.

    - E.g.  the second prompt in Figure 4, all methods seem to generate generally correct motions as described, but it’s quite hard to tell whose details (speed, height) are better with the overlapped frames without any definition or reference given.

    - Some of the generated examples in the provided demo videos are not well performed. E.g. in demo003, text prompt is “the person runs fast, kicks high in the air”, but the generated motion performs more like  a “jump” instead of a “high kick”.

    - The axe or reference for temporal direction should be mentioned alongside.


3. Ablation studies in Table 4 suggests that the contrastive learning module contributes relatively marginal improvements in generation performance compared to the impact of masking strategy and TransBlock, contradicting the authors’ claim that it is critical for learning a discriminative latent space. A more detailed analysis—such as latent-space visualizations or cluster separability metrics—would help validate this claim.


4. No details about test set settings, control signal settings and the controllability evaluation metrics for the comparison results in Table 6 for motion control evaluation (Similarly,  Table 13 for motion editing), which makes the presentation hard to follow nor evaluate.

5. The demo videos of motion control are confusing. It's not clear how the control is realized in each demo videos. What's the representation for the control signal in each case?  Some of the generated samples are confusing and even look like failure cases, e.g. control_6_joint.mp4.




[1] He, et al. “Masked Autoencoders Are Scalable Vision Learners.” In Proceedings of the IEEE/CVF Conference on Computer Vision and Pattern Recognition (CVPR), June 2022.

[2] Mishra, et al. “A simple, efficient and scalable contrastive masked autoencoder for learning visual representations.” arXiv preprint arXiv:2210.16870 (2022).

[3] Chen, et al. Executing Your Commands via Motion Diffusion in Latent Space. In Proceedings of the IEEE/CVF Conference on Computer Vision and Pattern Recognition (CVPR), 2023.

[4] Tseng, et al. “EDGE: Editable Dance Generation From Music.” In Proceedings of the IEEE/CVF Conference on Computer Vision and Pattern Recognition (CVPR), 2023, pp. 448-458.

[5] Shan, et al. Towards Open Domain Text-Driven Synthesis of Multi-Person Motions. n Proceedings of the European Conference on Computer Vision (ECCV) 2024.

[6] Shanchuan, et al. Common diffusion noise schedules and sample steps are flawed. In Proceedings of the IEEE/CVF winter conference on applications of computer vision, pp. 5404–5411, 2024.

**Questions:**

Please see the weaknesses points.

---

> ### Author Response · Authors · 2025-11-24
> **Response to dQuQ (1/n)**
>
> _Thank you for your constructive and thoughtful comments. They were indeed helpful in improving the paper. We take this opportunity to address your concerns:_
>
> _**List of changes in the manuscript:**_
>
> > 1. `Section B.5 in Appendix` **Asymmetric encoder-decoder design of motion encoder.**
> > 2. `Section B.2 in Appendix` **Comprehensive evaluation of motion representation quality.**
> > 3. `Section B.1 in Appendix` **More details of evaluation metrics.**
> > 4. `t2m_gen_compare in Demo Videos` **More direct comparison in demo videos.**
> > 5. `Figure.4 in Main Paper` **Add temporal arrows in the figure.4.**
> >
>
> ## W1: More clarification about technical novelty
> Our core goal is to revive continuous motion diffusion models for text-to-motion generation under standard experimental settings, with improvements in **generation quality, training efficiency, and inference speed**. This motivation arises from the observation that, despite the strong success of diffusion models in many other domains, they consistently underperform discrete VQ-based approaches in human motion generation across all three dimensions. This persistent gap reveals unique challenges specific to motion data and indicates that continuous motion diffusion models require renewed attention and reconsideration of their core components.
>
>
>
> We conducted a systematic analysis of continuous motion diffusion pipelines and identified fundamental issues in both the motion tokenizer and the diffusion backbone. Because continuous diffusion models have long lagged behind discrete tokenization–based approaches, the community has shown limited motivation to further refine them. In principle, continuous motion tokenizers should offer a higher representational upper bound than VQ-VAEs. Yet existing VAE-based tokenizers perform only on par with RVQ, failing to exploit the advantages of continuous latent spaces or to deliver superior generation quality. At the same time, the diffusion backbones themselves have seen little progress, as their design remains largely unchanged since early MDM and MLD variants, and MotionLCM mainly accelerates inference through LCM distillation instead of addressing the core modeling limitations.
>
>
>
> We performed a comprehensive design analysis of architectures and training strategies to address these limitations, aiming to achieve breakthroughs in generation quality, training efficiency, and inference speed. Importantly, this is achieved without requiring additional complex modules or auxiliary information. Using standard experimental settings, a standard CLIP text encoder, and 263-dimensional motion features, our approach achieves state-of-the-art performance across multiple metrics.

---

> ### Author Response · Authors · 2025-11-24
> **Response to dQuQ (2/n)**
>
> ### 1.1 MoCMAE: improving motion encoder for better motion representation
> **Problem:** In text-to-motion generation, motion encoder design has long been neglected. Most existing methods, such as MDM and MLD, still use continuous VAEs with minimal modifications, while discrete VQ-based methods (e.g., T2M-GPT’s VQ-VAE, MoMask’s RVQ) mainly improve quantization rather than the encoder itself. Consequently, continuous VAEs often achieve reconstruction performance comparable to RVQ, failing to fully exploit the potential of continuous motion encoders. In some newly proposed clustering metrics, continuous VAEs even underperform RVQ. In Text-to-Motion Retrieval (TMR), many works directly reuse the VAE or VQ-VAE encoder for contrastive learning with CLIP text embeddings, without optimizing the encoder for richer representations.
>
> **Related Work:** Only a few TMR works attempt to address this issue. Guangtao Lyu et al. model motion-text correspondences with a sparse vocabulary and a skeleton-aware encoder to capture spatio-temporal structures. SALAD introduces a 2D skeleton-aware encoder with convolution and pooling to enhance structural modeling. However, both works focus on producing discriminative representations for retrieval tasks, and due to slower inference, they have limited adoption in generative motion modeling.
>
> **Our Approach:** To address this overlooked problem, we propose **MoCMAE**, which combines comprehensive improvements in model architecture and training strategy:
>
> 1. **Training Strategy:** We leverage **discriminative representation learning techniques to enhance the latent space for generative modeling** by combining masked motion modeling (MMM) with contrastive learning (CL) to train a motion encoder from scratch. While MAE and CL are well-established for **discriminative tasks**, applying them in this generation context is novel. This approach enables the encoder to produce a structured, high-quality latent space that directly benefits motion generation, alleviating issues such as latent crowding and posterior collapse commonly observed in VAEs.
> 2. **Architectural Improvements:** MoCMAE uses an **asymmetric encoder-decoder design**: a heavy encoder captures rich global semantics from masked motion sequences, while a lightweight decoder ensures fast reconstruction during generation. This allows the encoder to focus on high-quality representation learning without being bottlenecked by decoder capacity.
>
> **Results**: MoCMAE achieves a **10× improvement in reconstruction quality** (Table 2) and surpasses various motion tokenizers including VAE, VQ-VAE, and RVQ-VAE on multiple clustering metrics (Table 18 in `Appendix B.2` , newly added).
>
> **Compared to recent image generation work, such as RAE by Saining Xie et al.,** which replaces the VAE with a pre-trained image autoencoder  to obtain a better representation space, our approach shares a similar philosophy of improving the latent representation space to enhance generative modeling. Since motion generation lacks pre-trained autoencoders, we train a motion autoencoder from scratch using established discriminative representation learning techniques, effectively bridging discriminative priors to generative motion modeling for the first time in this domain. This effectively bridges discriminative priors to generative motion modeling and demonstrates that enhancing the latent space can substantially improve the capability of generative motion models.

---

> ### Author Response · Authors · 2025-11-24
> **Response to dQuQ (3/n)**
>
> ### 1.2 ccDIT: improving continuous motion diffusion models
> **Motivation:**
> Continuous motion diffusion models (MDM → MLD → MotionLCM) have seen minimal structural improvements, and most works focus on efficiency rather than generation quality. Even StableMotionFusion, which explored structural changes, still lags behind MoMask in quality and is ~10× slower. In contrast, discrete models (T2M-GPT → MoMask → BAMM) have evolved rapidly. Inspired by image diffusion advances and discrete T2M progress, we systematically improve both the **architecture** and **training strategy** of continuous motion diffusion models.
>
>
>
> **Training Strategy – Stable-Min-SNR-Gamma:**
> We identify two key issues in motion diffusion models compared to advanced image diffusion models: training-inference inconsistency and multi-step SNR conflicts. These are typically addressed individually using Zero-SNR and Min-SNR-Gamma strategies.
>
> Training-inference inconsistency arises because, during training, noise is added over 1000 steps, but the sequence at the final step is not pure noise. In contrast, inference starts from pure noise at step 1000, leading to a discrepancy. This issue is more pronounced in motion data, which consists of multi-frame sequences similar to video. Due to temporal redundancy and physical constraints of human motion, plausible motions can still be decoded after 1000 steps of training noise (as shown in Figure X), making training relatively easier. Inference from pure noise, however, is more challenging, which degrades performance.
>
> The Zero-Min-SNR strategy addresses this by forcing the final 1000-step sequence to be pure noise. However, this creates a conflict with Min-SNR-Gamma, as the resulting SNR is zero, making its denominator invalid. To resolve this, we propose Stable-Min-SNR-Gamma, which adds a small constant to the denominator. This simple adjustment ensures compatibility between the two strategies without affecting their theoretical foundations.
>
> Results: Ablation studies (Table 5, w/o Zero-SNR and w/o Stable-Min-SNR-Gamma) show that removing either component degrades performance, while using both together yields the best results.
>
>
>
> **Architecture Improvements – ccDIT:**
> Inspired by text-to-image and discrete text-to-motion advances, we analyzed various architectural modifications and distilled only the most effective design choices for ccDIT. Our goal is to improve model quality, training efficiency, and inference speed without introducing extra auxiliary information or task-specific enhancements.
>
> Key features include:
>
> + Global and local conditional feature fusion: global fusion uses AdaLN-zero, while local fusion employs cross-attention. Prior motion diffusion models typically focus on either global or local alignment, both of which are less effective than considering both simultaneously.
> + Skip connections to improve training efficiency.
>
> **Ablation studies** (Table 5, top rows) show that removing any component degrades performance: _w/o sentence_ and _w/o word_ remove global or local features, respectively, while _w/o skip connection_ slightly reduces performance.
>
> **Additional improvement:** Replacing CLIP with the better-performing T5 model (compared to SnapMogen) further improves results, highlighting that diffusion-based motion models still have room for architectural advancement.
>
>
>
> **Summary:**
>
> Overall, our contributions lie in revisiting the core components that have been neglected in prior diffusion-based T2M systems, and in uncovering the underlying reasons for the long-standing underperformance of continuous motion models. We find that continuous models have been overshadowed by discrete tokenization approaches, resulting in limited motivation for architectural innovation, and that motion representation learning itself has been largely overlooked. Our analysis reveals key limitations in both the quality of motion representations and the internal design of continuous diffusion models. To address these issues, we introduce discriminative representation learning techniques to substantially enhance the motion encoder, yielding a richer and more structured latent space for generative modeling. On the generative side, we make targeted improvements to the diffusion architecture and training strategy, resolving intrinsic weaknesses and refining the model design. Together, these advances lead to significant gains in generation quality, training efficiency, and inference speed.

---

> ### Author Response · Authors · 2025-11-24
> **Response to dQuQ (4/n)**
>
> ## Q2: Asymmetric encoder-decoder design of MoCMAE
> The asymmetric encoder–decoder architecture in MoCMAE is intentionally designed to improve both **representation quality** and **inference efficiency**. A strong encoder focuses on extracting expressive motion features from masked inputs, while a lightweight decoder reconstructs motions without diminishing the encoder’s responsibility in shaping the latent space.
>
> The design serves two main purposes:
>
> 1. **Higher-quality latent space.**
> A weaker decoder cannot compensate for poorly structured latents, forcing the encoder to learn discriminative and well-organized representations. This alleviates issues such as latent crowding and partial posterior collapse often observed in generative models.
> 2. **Efficient inference.**
> Since the decoder participates in every reconstruction step during denoising, keeping it lightweight directly reduces inference-time computation.
>
> We further evaluate a **symmetric variant** with a stronger decoder (Table 24) in `Appendix B.5`. The observations are consistent:
>
> + **Weaker latent quality despite better reconstruction.**
> Although the stronger decoder achieves slightly lower reconstruction FID, the **generation quality becomes worse**. A powerful decoder overly compensates during reconstruction, reducing pressure on the encoder and leading to a less structured latent space.
> + **Higher inference cost.**
> The heavier decoder increases computational overhead during reconstruction.
>
> **Conclusion.**
> The asymmetric design improves latent representation quality and generation-time efficiency. Table 24 empirically demonstrates how the decoder strength influences both latent learning and inference speed.
>
>
>
> **Table: Comparison of asymmetric and symmetric MoCMAE designs.**
> MoCMAE* adds Transformer blocks to the decoder. While reconstruction improves slightly, generation quality decreases modestly and inference becomes slightly slower. The asymmetric design maintains the best balance between generation fidelity and efficiency.
>
> | Methods | FID ↓ (Recon) | MPJPE ↓ |  | FID ↓ (Gen) | MM-Dist ↓ | Speed (s) ↓ |
> | --- | --- | --- | --- | --- | --- | --- |
> | MoCMAE | 0.0024 | 8.8 |  | 0.041 | 2.898 | 0.041 |
> | MoCMAE* | 0.0021 | 8.2 |  | 0.049 | 2.912 | 0.043 |
>
>
>
>
> ## **Q3: Details of the word-level and sentence-level text features**
> **A:** The **word-level features** are obtained from the text encoder (e.g., CLIP text encoder), representing individual token embeddings of the input text. The **sentence-level features** are computed by averaging the word-level features to form a global textual representation.
>
> For fusion with motion features:
>
> + **Word-level features** are integrated using **cross-attention**, allowing fine-grained, token-level alignment with motion sequences.
> + **Sentence-level features** are fused using **AdaLN-Zero**, providing global conditioning and enabling consistent modulation of the motion features across the entire sequence.
>
> This design ensures that both local (word-level) and global (sentence-level) textual information effectively guide the motion generation process.

---

> ### Author Response · Authors · 2025-11-24
> **Response to dQuQ (5/n)**
>
> ## Q4: More direct comparison in demo videos
> **A:**
> We thank the reviewer for the valuable suggestions. We agree that overlapping frame visualizations and ambiguous adjectives in text prompts (e.g., "fast", "high", "slowly") can make qualitative comparison difficult.
> To address this, we have added three sets of comparative T2M rendering videos in the folder `t2m_gen_compare of Demo Videos`, showing generated motions from our method and existing baselines side by side. These videos provide a more intuitive assessment of motion details and temporal progression.
>
> ### Q4.1: Overlapping frames make it hard to compare details
> **A:**
> From the new supplementary videos  in the folder `t2m_gen_compare`, it can be observed that compared with MoMask, our method performs higher kicks. MoMask starts the kick early but does not achieve sufficient speed or height, whereas our method first runs and then executes a higher kick.
> Similarly, for prompts with multiple actions, our method produces more complete motions, while MoMask and MLD often miss some actions.
>
> ### Q4.2: Motion performs more like a “jump” instead of a “high kick”
> **A:**
> The supplementary videos  in the folder `t2m_gen_compare` show that Ours and MoMask both execute the kick action, but because it runs first and performs a single-leg kick, it visually resembles a single-leg jump. Single-leg jumps and high kicks can look similar at high speeds, which is a general difficulty in text-to-motion generation. Nevertheless, our method consistently generates higher kicks compared with momask, as demonstrated in the comparative videos.
>
> ### Q4.3: Temporal axis or reference should be indicated
> **A:**
> We have added temporal arrows in the figure.4 to indicate motion direction and progression in the main paper. This provides a clearer reference for temporal interpretation and helps visualize the sequence of actions.

---

> ### Author Response · Authors · 2025-11-24
> **Response to dQuQ (6/n)**
>
> ## **Q5: Further evaluation of contrastive learning**
> A: To further analyze the impact of different components in MoCMAE, we evaluate the quality of the learned latent motion representations. We provide a more rigorous and comprehensive evaluation of the learned representation space and motion encoder  in `Appendix B.2`. While the masking strategy and TransBlock primarily enhance the overall model capability, contrastive learning (CL) can further improve both representation quality and generation performance.
>
> We adopt two complementary categories of metrics to quantify latent space quality:
>
> + **Internal metrics:** Silhouette Score (SC), Calinski–Harabasz Index (CHI), and Davies–Bouldin Index (DBI), which assess the geometric structure and compactness of the latent space without requiring labels.
> + **External metrics:** Semantic alignment is evaluated via the five-nearest-neighbor (5-NN) classification accuracy. Pseudo-labels are constructed from nine motion-related keywords (`chicken`, `stair`, `crawl`, `jump`, `pick`, `breast stroke`, `ballet`, `karate`, `sit`), with 30 randomly sampled instances per class to ensure a balanced evaluation set.
>
> ---
>
> ### Quantitative comparison of latent representations
> | Model | SC ↑ | CHI ↑ | DBI ↓ | 5-NN Acc ↑ |
> | --- | --- | --- | --- | --- |
> | VQ-VAE | -0.086 | 5.42 | 4.241 | 0.467 |
> | RVQ-VAE | -0.020 | 7.83 | 4.024 | 0.520 |
> | VAE | -0.008 | 1.38 | 6.663 | 0.313 |
> | **MoCMAE** | **0.047** | 5.23 | **3.984** | **0.725** |
> | MoCMAE w/o CL | 0.025 | 4.28 | 4.086 | 0.643 |
>
>
> **Analysis:**
> Removing CL results in degraded performance across multiple metrics:
>
> + **SC drop:** indicates weaker local separability among motion samples.
> + **CHI drop:** suggests that intra-cluster cohesion is reduced, and clusters are less well-defined.
> + **DBI increase:** clusters are more overlapping, reflecting noisier latent representations.
> + **5-NN accuracy drop:** semantic alignment between latent features and motion categories is weaker.
>
> Overall, these results demonstrate that CL contributes to both geometric discriminability and semantic coherence.
>
> ---
>
> ### Impact on reconstruction and generation
> | Methods | Reconstruction FID ↓ | MPJPE ↓ | Generation FID ↓ | MM-Dist ↓ |
> | --- | --- | --- | --- | --- |
> | MoCMAE | 0.002 | 8.8 | 0.041 | 2.898 |
> | MoCMAE w/o CL | 0.004 | 9.6 | 0.058 | 2.974 |
>
>
> **Analysis:**
> Removing CL lowers both reconstruction and generation performance. Specifically:
>
> + Higher reconstruction FID and MPJPE indicate reduced fidelity in reconstructing motions from the latent space.
> + Higher generation FID and MM-Dist reflect less accurate and less diverse generated motions.
>
> ---
>
> ### Summary
> Adding CL to MoCMAE improves **latent representation quality**, **reconstruction fidelity**, and **generation quality**.
>
> Importantly, this training adjustment:
>
> + Does **not increase inference cost**,
> + Enhances both reconstruction and generation performance,
> + Demonstrates that discriminative representation learning techniques can effectively improve generative models.
>
> Since representation techniques like MMM and CL have received relatively little attention in text-to-motion generation, exploring them further offers a promising avenue for improving T2M performance.

---

> ### Author Response · Authors · 2025-11-24
> **Response to dQuQ (7/n)**
>
> ## Q6: Controllable Text-to-Motion Generation and Motion Editing Settings
> ### 1. Controllable text-to-motion generation
> Following prior works such as **MotionLCM**, **OmniControl**, and **MaskControl**, we adopt similar experimental settings and evaluation metrics.
>
> + **Control Signal:** Pelvis trajectory is used as the primary control signal.
> + **Evaluation Metrics:**
>     - **Trajectory Error (Traj. err.):** Measures the ratio of unsuccessful trajectories, where any control joint location exceeds a predefined threshold.
>     - **Location Error (Loc. err.):** Counts the number of unsuccessful control joints.
>     - **Average Error (Avg. err.):** Computes the mean location error across all control joints.
>
> These metrics are consistent with previous works, allowing fair comparison of controllability performance.
>
> ---
>
> ### 2. Motion editing
> We follow the **MotionFix** settings and evaluation protocol. Similar to text-to-motion synthesis, direct distance-based metrics are unreliable due to multiple plausible ground-truth motions for the same text.
>
> + **Motion-to-Motion Retrieval Metrics:**
>     - **Generated-to-Source Retrieval:** Measures preservation of characteristics from the original motion.
>     - **Generated-to-Target Retrieval:** Measures how accurately the edited motion matches the intended target motion.
> + **Feature Representation:**
>     - **TMR [Petrovich et al., 2023]** is used as the feature extractor.
>     - The model is retrained on **HumanML3D** following the original setup.
> + **Reported Metrics:**
>     - **Recall at rank k (R@k):** Fraction of times the correct motion is among the top k retrievals.
>     - **Ranks:** R@1, R@2, R@3.
>     - **Average Recall (AvgR):** Average across R@1–R@3.
>     - **Gallery Size:** 32 randomly sampled motions per test batch.
>
> This setup ensures our evaluation is fully aligned with prior works, enabling a fair and comparable assessment of motion editing quality.
>
> For a more detailed introduction, please refer to `Appendix B.1`.
>
> ## Demo Videos of Motion Control
> We thank the reviewer for the comment. The demo videos follow the same visualization scripts as in **MotionLCM**. The **green dots** indicate the control signal joint, meaning the motion is generated based on the given caption and control joint.
>
> Due to the scale and coordinate range used for visualization, the skeleton may occasionally appear slightly distorted; however, the overall motion and trajectory alignment are correctly executed. These videos demonstrate how the control signal guides motion generation, and despite minor visual artifacts, the generated actions generally follow the desired joints and captions.
>
> We are also exploring better visualization methods to make the control effects clearer.

---

### Official Review · Reviewer_9iVt · 2025-10-29

**Soundness:** 3
**Presentation:** 4
**Contribution:** 2
**Rating:** 4
**Confidence:** 4

**Summary:**

The paper proposes COME, including a Contrastive Masked Autoencoder (MoCMAE) and a cross-condition diffusion transformer (ccDiT) to improve the performance of text2motion generation. It first identifies the core limitations of previous works are 1) poor motion representation and 2) suboptimal generative modeling methods. and then address them by proposing a better design motion autoencoder and a diffusion transformer with modern designs. Extensive experiments demonstrated the improved performance using the proposed design across a suite of different motion generation tasks.

**Strengths:**

1. I agree that a better motion representation is important, and the proposed MoCMVAE makes sense along this line.
2. The proposed latent diffusion model adopts several modern designs that should clearly improve performance compared to other old versions of latent diffusion models.
3. I appreciate that the authors have conducted extensive experiments on a wide range of tasks and datasets. These experiments demonstrate that the proposed design could improve motion generation performance for broader tasks and areas.

**Weaknesses:**

1. Although it shows better performance comprehension evaluation across different tasks, the technical novelty is relatively limited.
2. Constrative Masked Autoencoder is interesting, but more closely related work about motion autoencoder and latent representation should be discussed and compared, to better highlight the difference and motivation of the proposed design.
3. ccDiT is rather a common design. Min-SNR-γ is commonly used in other areas, and the design of "Stable-Min-SNR-γ" is relatively minor. According to Table 5, the improvement is subtle.
4. Given that most designs are relatively general or random, that is not very well motivated or introduce motion-specific insight, I'm looking for significantly stronger performance. However, the text2motion generation performance is not good enough. Some sota methods are missing, for example, StableMofusion [1].

Overall, I appreciate the effort to conduct extensive experiments across tasks and datasets. However, I believe ICLR is looking for technical novelty and insights that can have a broader impact to the community. So, I’m on the fence and leaning toward negative.

[1] Huang, Yiheng, et al. "Stablemofusion: Towards robust and efficient diffusion-based motion generation framework." Proceedings of the 32nd ACM International Conference on Multimedia. 2024.

**Questions:**

1. I wonder what the core contributions are in ccDiT, and how it is different from previous related works.

---

> ### Author Response · Authors · 2025-11-24
> **Response to 9iVt(1/n)**
>
> _Thank you for your constructive and thoughtful comments. They were indeed helpful in improving the paper. We take this opportunity to address your concerns:_
>
> _**List of changes in the manuscript:**_
>
> > 1. `Section B.4 in Appendix` **Comparison with additional strong baselines.**
> > 2. `Section B.2 in Appendix` **Comprehensive evaluation of motion representation quality.**
> > 3. `Section B.5 in Appendix` **Asymmetric encoder-decoder design of motion encoder.**
> > 4. `t2m_gen_compare in Demo Videos` **More direct comparison in demo videos.**
>
>
> ## W1: More clarification about technical novelty
> Our core goal is to revive continuous motion diffusion models in text-to-motion generation under standard experimental settings, improving **generation quality, training efficiency, and inference speed**.
>
> Our core motivation comes from the observation that, in human motion generation, diffusion models still underperform discrete VQ-based methods in terms of generation quality, training efficiency, and inference speed, even though diffusion models have achieved strong results in other domains. This highlights several unique challenges of motion generation.
>
>
>
> We conducted a systematic analysis of motion diffusion model components and found that both the motion tokenizer and the diffusion model itself have fundamental issues. Due to the limited performance of diffusion-based approaches, the community has shown little interest in modifying these models. From MDM to MLD and MotionLCM, the changes to tokenizers and diffusion models have been minimal, far less than the variations explored in discrete methods.
>
>
>
> Specifically, continuous motion tokenizers should, in principle, provide higher upper bounds than VQ-VAEs. However, existing continuous motion VAE tokenizers perform almost on par with discrete RVQ, failing to realize their potential and not offering higher generation quality. Meanwhile, the diffusion models themselves are not capable enough for motion generation. As research focus shifted toward discrete approaches, the architecture of motion diffusion models has changed very little since MDM and MLD; even MotionLCM simply applies LCM distillation on top of MLD to accelerate inference.
>
>
>
> We performed a comprehensive design analysis of architectures and training strategies to address these limitations, aiming to achieve breakthroughs in generation quality, training efficiency, and inference speed. Importantly, this is achieved without requiring additional complex modules or auxiliary information. Using standard experimental settings, a standard CLIP text encoder, and 263-dimensional motion features, our approach achieves state-of-the-art performance across multiple metrics.

---

> ### Author Response · Authors · 2025-11-24
> **Response to 9iVt(2/n)**
>
> ### MoCMAE: improving motion encoder for better motion representation
> **Problem:** In text-to-motion generation, motion encoder design has long been neglected. Most existing methods, such as MDM and MLD, still use continuous VAEs with minimal modifications, while discrete VQ-based methods (e.g., T2M-GPT’s VQ-VAE, MoMask’s RVQ) mainly improve quantization rather than the encoder itself. Consequently, continuous VAEs often achieve reconstruction performance comparable to RVQ, failing to fully exploit the potential of continuous motion encoders. In some newly proposed clustering metrics, continuous VAEs even underperform RVQ. In Text-to-Motion Retrieval (TMR), many works directly reuse the VAE or VQ-VAE encoder for contrastive learning with CLIP text embeddings, without optimizing the encoder for richer representations.
>
> **Related Work:** Only a few TMR works attempt to address this issue. Guangtao Lyu et al. model motion-text correspondences with a sparse vocabulary and a skeleton-aware encoder to capture spatio-temporal structures. SALAD introduces a 2D skeleton-aware encoder with convolution and pooling to enhance structural modeling. However, both works focus on producing discriminative representations for retrieval tasks, and due to slower inference, they have limited adoption in generative motion modeling.
>
> **Our Approach:** To address this overlooked problem, we propose **MoCMAE**, which combines comprehensive improvements in model architecture and training strategy:
>
> 1. **Training Strategy:** We leverage **discriminative representation learning techniques to enhance the latent space for generative modeling** by combining masked motion modeling (MMM) with contrastive learning (CL) to train a motion encoder from scratch. While MAE and CL are well-established for **discriminative tasks**, applying them in this generation context is novel. This approach enables the encoder to produce a structured, high-quality latent space that directly benefits motion generation, alleviating issues such as latent crowding and posterior collapse commonly observed in VAEs.
> 2. **Architectural Improvements:** MoCMAE uses an **asymmetric encoder-decoder design**: a heavy encoder captures rich global semantics from masked motion sequences, while a lightweight decoder ensures fast reconstruction during generation. This allows the encoder to focus on high-quality representation learning without being bottlenecked by decoder capacity.
>
> **Results**: MoCMAE achieves a **10× improvement in reconstruction quality** (Table 2) and surpasses various motion tokenizers including VAE, VQ-VAE, and RVQ-VAE on multiple clustering metrics (Table 18 in the Appendix B.2, newly added).
>
> **Compared to recent image generation work, such as RAE by Saining Xie et al.,** which replaces the VAE with a pre-trained image autoencoder  to obtain a better representation space, our approach shares a similar philosophy of improving the latent representation space to enhance generative modeling. Since motion generation lacks pre-trained autoencoders, we train a motion autoencoder from scratch using established discriminative representation learning techniques, effectively bridging discriminative priors to generative motion modeling for the first time in this domain. This effectively bridges discriminative priors to generative motion modeling and demonstrates that enhancing the latent space can substantially improve the capability of generative motion models.

---

> ### Author Response · Authors · 2025-11-24
> **Response to 9iVt(3/n)**
>
> ### ccDIT: improving continuous motion diffusion models
> **Problem:**
> Continuous motion diffusion models (MDM → MLD → MotionLCM) have seen minimal structural improvements, and most works focus on efficiency rather than generation quality. Even StableMotionFusion, which explored structural changes, still lags behind MoMask in quality and is ~10× slower. In contrast, discrete models (T2M-GPT → MoMask → BAMM) have evolved rapidly. Inspired by image diffusion advances and discrete T2M progress, we systematically improve both the **architecture** and **training strategy** of continuous motion diffusion models.
>
>
>
> **Training Strategy – Stable-Min-SNR-Gamma:**
> We identify two key issues in motion diffusion models compared to advanced image diffusion models: training-inference inconsistency and multi-step SNR conflicts. These are typically addressed individually using Zero-SNR and Min-SNR-Gamma strategies.
>
> Training-inference inconsistency arises because, during training, noise is added over 1000 steps, but the sequence at the final step is not pure noise. In contrast, inference starts from pure noise at step 1000, leading to a discrepancy. This issue is more pronounced in motion data, which consists of multi-frame sequences similar to video. Due to temporal redundancy and physical constraints of human motion, plausible motions can still be decoded after 1000 steps of training noise (as shown in Figure X), making training relatively easier. Inference from pure noise, however, is more challenging, which degrades performance.
>
> The Zero-Min-SNR strategy addresses this by forcing the final 1000-step sequence to be pure noise. However, this creates a conflict with Min-SNR-Gamma, as the resulting SNR is zero, making its denominator invalid. To resolve this, we propose Stable-Min-SNR-Gamma, which adds a small constant to the denominator. This simple adjustment ensures compatibility between the two strategies without affecting their theoretical foundations.
>
> **Results:** Ablation studies (Table 5, w/o Zero-SNR and w/o Stable-Min-SNR-Gamma) show that removing either component degrades performance, while using both together yields the best results.
>
>
>
> **Architecture Improvements – ccDIT:**
> Inspired by text-to-image and discrete text-to-motion advances, we analyzed various architectural modifications and distilled only the most effective design choices for ccDIT. Our goal is to improve model quality, training efficiency, and inference speed without introducing extra auxiliary information or task-specific enhancements.
>
> Key features include:
>
> + Global and local conditional feature fusion: global fusion uses AdaLN-zero, while local fusion employs cross-attention. Prior motion diffusion models typically focus on either global or local alignment, both of which are less effective than considering both simultaneously.
> + Skip connections to improve training efficiency.
>
> **Ablation studies** (Table 5, top rows) show that removing any component degrades performance: _w/o sentence_ and _w/o word_ remove global or local features, respectively, while _w/o skip connection_ slightly reduces performance.
>
> **Additional improvement:** Replacing CLIP with the better-performing T5 model (compared to SnapMogen) further improves results, highlighting that diffusion-based motion models still have room for architectural advancement.
>
>
>
> ### Summary:
> Overall, our contributions lie in revisiting the core components that have been neglected in prior diffusion-based T2M systems, and in uncovering the underlying reasons for the long-standing underperformance of continuous motion models. We find that continuous models have been overshadowed by discrete tokenization approaches, resulting in limited motivation for architectural innovation, and that motion representation learning itself has been largely overlooked. Our analysis reveals key limitations in both the quality of motion representations and the internal design of continuous diffusion models. To address these issues, we introduce discriminative representation learning techniques to substantially enhance the motion encoder, yielding a richer and more structured latent space for generative modeling. On the generative side, we make targeted improvements to the diffusion architecture and training strategy, resolving intrinsic weaknesses and refining the model design. Together, these advances lead to significant gains in generation quality, training efficiency, and inference speed.

---

> ### Author Response · Authors · 2025-11-24
> **Response to 9iVt (4/n)**
>
> ## W2: **More discussion about motion encoder**
> The reviewer’s suggestion is very valuable, and we will include a more systematic discussion of related work in the final version. In fact, the design of **motion encoders has been largely overlooked** in recent years. Most methods still adopt early VQ-VAE, VAE, or RVQ architectures, with minimal modifications to the encoder. This stagnation limits the generation quality of diffusion models and also affects downstream tasks such as TMR. Many works directly reuse the VQ-VAE encoder to extract motion features and perform contrastive learning with CLIP text embeddings, without optimizing the motion encoder itself.
>
> Only a few TMR-focused works proposed novel solutions. For example, Guangtao Lyu et al. modeled motion–text alignment using a sparse vocabulary space and employed a 2D skeleton-aware encoder to capture spatio-temporal structure. SALAD proposed a 2D skeleton-aware encoder incorporating convolution and pooling to enhance structural modeling, but its inference cost is high and subsequent adoption is limited, appearing only in a few methods such as MaskControl.
>
> The **stagnation in motion encoder research** mainly stems from two factors:
>
> 1. The community has long focused on **text-to-motion generation**;
> 2. **Caption-style annotations** are not conducive to classification or disentangled representations, and the lack of **large-scale classification datasets** similar to ImageNet makes systematic **representation learning difficult**.
>
> In this work, we **revisit this overlooked core problem**. Based on systematic analysis, we find that **existing encoders exhibit significant deficiencies in representation quality**. To address this, we propose a simple yet effective motion representation learning framework that combines masked motion modeling with contrastive learning.
>
> We provide a more rigorous and comprehensive evaluation of the learned representation space and motion encoder  in `Appendix B.2 and B.5`. Through generation and reconstruction performance in Table 2, t-SNE visualization in Figures 16 and 17, transfer experiments on TMR in Tables 10 and 11, and quantitative latent representation metrics such as Silhouette Score, Calinski–Harabasz Index, Davies–Bouldin Index, and 5-NN classification accuracy in Table 18, we demonstrate that **stronger motion representations significantly improve both diffusion model generation quality and downstream task performance**. These results show that **integrating discriminative representation learning into motion encoders can effectively enhance generative models**.
>
> This is conceptually similar to the core idea of the recent contemporaneous work by **Saining Xie et al. (RAE)**, which replaces the VAE with a **pretrained autoencoder** to improve latent representations in image generation. The key difference is that, unlike images where pretrained autoencoders are available, in motion generation we train the autoencoder from scratch.
>
> We believe this study highlights an **important direction**: **improving the motion encoder itself** is as crucial as refining the diffusion model architecture and is a **key avenue for advancing motion generation and multimodal task performance**.

---

> ### Author Response · Authors · 2025-11-24
> **Response to 9iVt (5/n)**
>
> ## W3 & Q1: **More discussion about generative models**
> **Motivation:**
> Our core goal is to revive continuous motion diffusion models in text-to-motion generation under standard experimental settings, improving **generation quality, training efficiency, and inference speed**.
>
> Continuous motion diffusion models (MDM → MLD → MotionLCM) have seen minimal structural improvements.  Even StableMotionFusion, which explored architectural changes, still underperforms MoMask in quality and is ~10× slower. In contrast, discrete motion models (T2M-GPT → MoMask → BAMM) have evolved rapidly. Inspired by advances in **image diffusion** and discrete T2M, we systematically improve both the **architecture and training strategy** of continuous motion diffusion models.
>
> **Training Strategy – Stable-Min-SNR-Gamma:**
> We identify two key issues compared to image diffusion: **training-inference inconsistency** and **multi-step SNR conflicts**. Zero-SNR and Min-SNR-Gamma strategies are typically used individually to address these problems, but combining them is problematic because Zero-SNR produces a zero denominator for Min-SNR-Gamma.
> We propose **Stable-Min-SNR-Gamma**, which adds a small constant to the denominator, ensuring compatibility between the two strategies without altering their theoretical foundations. Ablation studies (Table 5, w/o Zero-SNR and w/o Stable-Min-SNR-Gamma) show that removing either component degrades performance, while using both yields the best results.
>
> **Architecture Improvements – ccDIT:**
> Inspired by text-to-image and discrete T2M advances, we systematically analyzed architectural modifications and retained only the most effective design choices.
>
> Key features include:
>
> + **Global and local conditional feature fusion:** global fusion uses AdaLN-zero, while local fusion employs cross-attention. Prior motion diffusion models often focus on either global or local alignment, which is less effective than considering both simultaneously.
> + **Skip connections** to improve training efficiency.
>
> Ablation studies (Table 5, top rows) demonstrate the importance of each component: removing global features (w/o sentence) or local features (w/o word) significantly degrades performance, while removing skip connections slightly reduces performance.
>
> **Additional Improvements:**
> Replacing CLIP with the better-performing **T5 model** (compared to SnapMogen) further improves results (Table 12), highlighting that diffusion-based motion models still have room for architectural enhancement. Moreover, adopting the **redundancy-reduced representations proposed by MAR-DM** allows for further gains (Table 19), demonstrating that **better motion representations complement architectural improvements**.
>
> Under standard experimental settings, our approach achieves **state-of-the-art performance** in **generation quality, training efficiency, and inference speed**, demonstrating the significant untapped potential of continuous motion diffusion models. Furthermore, the improved architecture and motion representations provide a **stronger foundational text-to-motion model**, enabling enhanced performance across various downstream tasks.
>
> ## W4. Comparison with StableMoFusion and Stronger Performance
> **4.1 Comparison with StableMoFusion**
>
> We provide a more rigorous and comprehensive comparison between StableMoFusion and other strong baselines in `Appendix B.4`. Compared with StableMoFusion, our method achieves **1.6× lower FID** and **7.3× faster inference**, surpassing it in both quality and efficiency. Moreover, COME generates high-quality results **directly**, without requiring vGRFs as a post-processing step for footskate reduction.
>
> | Method | FID ↓ | R Precision (top3) ↑ | AITS ↓ | Inference Steps ↓ |
> | --- | --- | --- | --- | --- |
> | StableMoFusion-base (DDPM1000) | 1.251 | 0.760 | 99.060 s | 1000 |
> | + Efficient Sampler | 0.076 | 0.836 | 1.004 s | 10 |
> | + Embedded-text Cache | 0.076 | 0.836 | 0.690 s | 10 |
> | + Parallel CFG | 0.076 | 0.836 | 0.544 s | 10 |
> | + FP16 | 0.076 | **0.837** | 0.499 s | 10 |
> | **COME (ours)** | **0.041** | 0.816 | **0.068 s** | 10 |

---

> ### Author Response · Authors · 2025-11-24
> **Response to 9iVt (6/n)**
>
> **4.2 Significantly Stronger Performance**
>
> In the main paper, all our results follow **standard data formats and training setups**, which we intentionally maintain to ensure fair and meaningful comparisons. A key motivation of our work is to challenge the widespread perception that **diffusion models are inherently slow or inferior** in motion generation.
>
> + **MotionLCM** restores diffusion efficiency through LCM acceleration but still suffers from limited quality.
> + **MARDM** reduces motion dimensionality drastically (from the standard 263D to 67D) and combines autoregression with diffusion to approach MoMask-level metrics; however, its inference is **~60× slower**.
>
> In contrast, **COME achieves state-of-the-art performance in quality, inference speed, and training efficiency simultaneously**, _under the standard 263-dim representation and standard evaluation protocol_—demonstrating that diffusion models can be both **high-quality and highly efficient** without altering the data format.
>
> While our main comparison focuses on standard settings for fairness, we note that **COME can achieve even stronger results when combined with more powerful components**:
>
> + In SnapMoGen, replacing CLIP-Text with the stronger **T5 encoder (same as MoMask++)** yields further improvements (see Table 12).
> + MARDM also reports that diffusion models perform better on their redundant representations; on these settings, COME achieves **even larger gains** (see Table 19).
>
> We did not include these boosted settings in the main paper to maintain fairness, but they confirm that **COME scales effectively and can reach even higher performance when stronger modules are introduced**.

---

### Official Review · Reviewer_KCkb · 2025-11-01

**Soundness:** 3
**Presentation:** 3
**Contribution:** 2
**Rating:** 4
**Confidence:** 4

**Summary:**

This paper proposes COME, a continuous text to motion model with a masked motion autoencoder (MoCMAE) for robust continuous latents and a conditioning-aware diffusion transformer (ccDiT) with AdaLN-Zero + token-level cross-attention. This paper reports strong FID/R-precision on HumanML3D and KIT-ML with 10-step DPM-Solver++ sampling.

**Strengths:**

1. Strong quantitative numbers on HumanML3D and competitive results on KIT-ML under standard metrics (FID, R-Precision).

2. One framework supports text-to-motion, motion editing, and text-to-motion control.

**Weaknesses:**

1. The motivation of closing the large performance gap between discrete and continuous is fine, but it’s quite similar to MARDM [1]. I understand the authors address it in a different way, but they do not compare with MARDM, neither methodologically nor in experimental results.

2. The authors claim “features encoded by existing continuous models (e.g., VAEs) are often overly concentrated” and rely on a t-SNE plot to support this. t-SNE does not preserve global distances or density, so that figure is not meaningful. The authors should use better diagnostics to show limitations of current continuous feature extractors and quantify “separability” in experiments comparing existing continuous motion encoders with the proposed masked motion encoder.

3. The model architecture and training are incremental. The authors just combine standard components and training recipes, and there is no strong technical improvement here.

4. Missing strong baseline results. Compare against recent, stronger works (MARDM [1], MotionStreamer [2], MotionLCM v2 [3]) for the text-to-motion generation task, and MaskControl [4] for the control task, with both qualitative and quantitative results. Current visualizations are not apples-to-apples.

5. Weak visual results in the supplement. For T2M generation, many cases show severe foot floating and foot–ground penetration. Compared with MoMask [5] (from the MoMask official website) on the same prompts (e.g., demo009, demo010), the motions look clearly less natural. For motion editing, I don’t see any real difference for edit_3. For T2M control, the human body shape (bone lengths) appears to change over time. In summary, most visualizations are not reasonable to me.

Reference

[1] Rethinking Diffusion for Text-Driven Human Motion Generation

[2] MotionStreamer: Streaming Motion Generation via Diffusion-based Autoregressive Model in Causal Latent Space

[3] MotionLCM-V2: Improved Compression Rate for Multi-Latent-Token Diffusion

[4] MaskControl: Spatio-Temporal Control for Masked Motion Synthesis

[5] MoMask: Generative Masked Modeling of 3D Human Motions

**Questions:**

Refer to Weaknesses.

---

> ### Author Response · Authors · 2025-11-24
> **Response to KCkb (1/n)**
>
> _Thank you for your constructive and thoughtful comments. They were indeed helpful in improving the paper. We take this opportunity to address your concerns:_
>
> _**List of changes in the manuscript:**_
>
> > 1. `Section B.4 in Appendix` **Comparison with additional strong baselines.**
> > 2. `Section B.2 in Appendix` **Comprehensive evaluation of motion representation quality.**
> >
>
> ## W1: Comparison and discussion about MARDM
> **Differences:**
>
> Although MARDM also reexamines diffusion models, **it improves performance at the cost of substantially slower inference.** Specifically, it begins with the assumption that diffusion models underperform discrete methods under standard configurations. To address this limitation, MARDM modifies the motion representation, reduces feature dimensionality, retrains its own evaluation model, and adopts a MAR-style autoregressive-plus-diffusion architecture. While this design achieves MoMask-level quality, it results in inference that is approximately 60× slower.
>
> In contrast, we redesign both the tokenizer and the generation model, from architecture to training strategy, and **achieve consistent improvements in generation quality, training efficiency, and inference speed.**
>
> Speciffically, we perform a systematic analysis of motion diffusion models and identify issues in both the motion tokenizer and the generation model. Continuous tokenizers should theoretically outperform VQ-based ones, yet existing continuous VAEs offer no real advantage over RVQ and fail to provide a higher quality ceiling. At the same time, previous diffusion backbones are weak because diffusion has long lagged behind discrete models and has not absorbed recent architectural advances.
>
> We redesign both the tokenizer and the generation model, from architecture to training strategy, and achieve consistent improvements in generation quality, training efficiency, and inference speed. Under standard experimental settings, using the standard CLIP text encoder and 263-D motion features, our method achieves state-of-the-art performance across all metrics.
>
>
>
> **Experimental Comparisons:**
>
> The experimental results are shown in following tables. Following the MARDM experimental setup, we conducted a comprehensive comparison with their methods. The results demonstrate that while maintaining significantly **higher generation quality**, our model achieves approximately **100× faster inference speed**. This indicates that, under the same data format and evaluation protocol, our design offers substantial advantages in both efficiency and performance. More detailed analyses and additional experiments can be found in `Appendix C.1`.
>
> ### **Table R.1: HumanML3D**
> | Method | Framework | Top1 ↑ | Top2 ↑ | Top3 ↑ | FID ↓ | Matching ↓ | MModality ↑ | CLIP ↑ |
> | --- | --- | --- | --- | --- | --- | --- | --- | --- |
> | T2M-GPT | VQ | 0.470 | 0.659 | 0.758 | 0.335 | 3.505 | 2.018 | 0.607 |
> | MMM | VQ | 0.487 | 0.683 | 0.782 | 0.132 | 3.359 | 1.241 | 0.635 |
> | MoMask | VQ | 0.490 | 0.687 | 0.786 | 0.116 | 3.353 | 1.263 | 0.637 |
> | MDM-50Step | Diffusion | 0.440 | 0.636 | 0.742 | 0.518 | 3.640 | _3.604_ | 0.578 |
> | MotionDiffuse | Diffusion | 0.450 | 0.641 | 0.753 | 0.778 | 3.490 | 3.179 | 0.606 |
> | MLD | Diffusion | 0.461 | 0.651 | 0.750 | 0.431 | 3.445 | 3.506 | 0.610 |
> | ReMoDiffuse | Diffusion | 0.468 | 0.653 | 0.754 | 0.883 | 3.414 | 2.703 | 0.621 |
> | **COME (ours)** | Diffusion | **0.508** | **0.699** | **0.798** | **0.074** | **3.246** | **3.762** | **0.648** |
> | MARDM-DDPM | Autoreg. | 0.492 | 0.690 | 0.790 | 0.116 | 3.349 | 2.470 | 0.637 |
> | MARDM-SiT | Diffusion | _0.500_ | _0.695_ | _0.795_ | _0.114_ | _3.270_ | 2.231 | _0.642_ |
>
>
> ### **Table R.2: KIT**
> | Method | Framework | Top1 ↑ | Top2 ↑ | Top3 ↑ | FID ↓ | Matching ↓ | MModality ↑ | CLIP ↑ |
> | --- | --- | --- | --- | --- | --- | --- | --- | --- |
> | T2M-GPT | VQ | 0.359 | 0.553 | 0.690 | 0.593 | 3.765 | 1.798 | 0.651 |
> | MMM | VQ | 0.363 | 0.569 | 0.724 | 0.478 | 3.629 | 1.455 | 0.660 |
> | MoMask | VQ | 0.369 | 0.588 | 0.731 | 0.411 | 3.577 | 1.309 | 0.669 |
> | MDM | Diffusion | 0.333 | 0.561 | 0.689 | 0.585 | 4.002 | 1.681 | 0.605 |
> | MotionDiffuse | Diffusion | 0.344 | 0.536 | 0.658 | 3.845 | 4.167 | 1.774 | 0.626 |
> | MLD | Diffusion | 0.351 | 0.536 | 0.658 | 0.492 | 3.746 | 1.803 | 0.646 |
> | ReMoDiffuse | Diffusion | 0.356 | 0.572 | 0.706 | 1.725 | 3.735 | _1.928_ | 0.665 |
> | **COME (ours)** | Diffusion | **0.396** | **0.608** | **0.758** | **0.208** | **3.206** | **2.062** | **0.698** |
> | MARDM-DDPM | Autoreg. | 0.375 | 0.597 | 0.739 | 0.340 | 3.489 | 1.479 | 0.681 |
> | MARDM-SiT | Diffusion | _0.387_ | _0.610_ | _0.749_ | _0.242_ | _3.374_ | 1.312 | _0.692_ |
>
>
> ### **Table R.3: Inference Speed**
> | Method | MDM | MotionDiffuse | T2M-GPT | MLD | MMM | MoMask | MARDM | **COME (ours)** |
> | --- | --- | --- | --- | --- | --- | --- | --- | --- |
> | AIT (s) | 14.31 | 7.35 | 0.32 | 0.21 | 0.06 | 0.045 | 2.4 | **0.022** |

---

> ### Author Response · Authors · 2025-11-24
> **Response to KCkb (2/n)**
>
> ## W2: Comprehensive evaluation of motion representation quality
> Thank you for your valuable suggestion. We provide a more rigorous and comprehensive evaluation of the learned representation space in `Appendix B.2`.
>
> To systematically assess the quality of the learned latent motion representations, we perform a comprehensive evaluation combining internal geometric metrics, external semantic metrics, and qualitative visualization. Our aim is to determine whether the latent space learned by _MoCMAE_ exhibits superior structural organization and semantic coherence compared with existing continuous and discrete encoders.
>
> ### Evaluation Protocol
> We adopt two complementary categories of metrics:
>
> + **Internal metrics:** Silhouette Score (SC), Calinski–Harabasz Index (CHI), and Davies–Bouldin Index (DBI), which quantify the geometric structure and compactness of the latent space without requiring labels.
> + **External metrics:** Semantic alignment is evaluated via five-nearest-neighbor (5-NN) classification accuracy. Pseudo-labels are constructed using nine motion-related keywords ("chicken", "stair", "crawl", "jump", "pick", "breast stroke", "ballet", "karate", "sit"), with randomly sampled instances per class to ensure a balanced evaluation set.
>
> ### Quantitative Analysis
> | Model | SC | CHI | DBI | 5-NN Accuracy |
> | --- | --- | --- | --- | --- |
> | VQ-VAE | -0.086093 | 5.42 | 4.241479 | 0.4667 |
> | RVQ-VAE | -0.020365 | 7.83 | 4.024332 | 0.5197 |
> | VAE | -0.008453 | 1.38 | 6.662875 | 0.3125 |
> | **MoCMAE** | **0.047418** | 5.23 | **3.984182** | **0.7250** |
>
>
> As shown in table, human motion categories naturally overlap and can be described in multiple ways, making keyword-based grouping only a coarse approximation. As a result, absolute SC values remain close to zero across all encoders. Despite this inherent difficulty, _MoCMAE_ consistently outperforms other continuous and discrete encoders on both internal and external metrics. It achieves the highest Silhouette Score (SC) and 5-NN accuracy, as well as the lowest Davies–Bouldin Index (DBI), indicating that the learned latent space is simultaneously more geometrically coherent and semantically meaningful. These results quantitatively confirm the superiority of _MoCMAE_ in capturing structured and semantically aligned motion representations.
>
> ### Qualitative Visualization
> To complement the quantitative evaluation, we visualize the learned motion representations using t-SNE. Although human motion inherently forms overlapping and continuous manifolds, _MoCMAE_ representations exhibit broader coverage and more coherent local groupings compared with baseline encoders. This wider distribution reflects richer latent diversity, which facilitates more effective diffusion-based denoising and higher-fidelity motion generation. For detailed visualization, please refer to `Appendix B.2`, Figures 16 and 17.

---

> ### Author Response · Authors · 2025-11-24
> **Response to KCkb (3/n)**
>
> ## W3: **More clarification about motivation and innovation**
> **Motivation:**
> Our core goal is to revive continuous motion diffusion models in text-to-motion generation under standard experimental settings, improving **generation quality, training efficiency, and inference speed**.
>
> Our core motivation comes from the observation that, in human motion generation, diffusion models still underperform discrete VQ-based methods in terms of generation quality, training efficiency, and inference speed, even though diffusion models have achieved strong results in other domains. This highlights several unique challenges of motion generation.
>
>
>
> While MAR-DM attempted to rethink diffusion models, their approach still focused on the standard observation that diffusion models underperform discrete methods. They modified the data format, reduced the dimensionality of motion data, and combined autoregressive and diffusion model architectures in a MAR-like manner. Although this allowed them to achieve performance close to MoMask on their new evaluation metrics, inference speed dropped by nearly 60×.
>
>
>
> In contrast, we conducted a systematic analysis of motion diffusion model components and found that both the motion tokenizer and the diffusion model itself have fundamental issues. Due to the limited performance of diffusion-based approaches, the community has shown little interest in modifying these models. From MDM to MLD and MotionLCM, the changes to tokenizers and diffusion models have been minimal, far less than the variations explored in discrete methods.
>
>
>
> Specifically, continuous motion tokenizers should, in principle, provide higher upper bounds than VQ-VAEs. However, existing continuous motion VAE tokenizers perform almost on par with discrete RVQ, failing to realize their potential and not offering higher generation quality. Meanwhile, the diffusion models themselves are not capable enough for motion generation. As research focus shifted toward discrete approaches, the architecture of motion diffusion models has changed very little since MDM and MLD; even MotionLCM simply applies LCM distillation on top of MLD to accelerate inference.
>
>
>
> We performed a comprehensive design analysis of architectures and training strategies to address these limitations, aiming to achieve breakthroughs in generation quality, training efficiency, and inference speed. Importantly, this is achieved without requiring additional complex modules or auxiliary information. Using standard experimental settings, a standard CLIP text encoder, and 263-dimensional motion features, our approach achieves state-of-the-art performance across multiple metrics.

---

> ### Author Response · Authors · 2025-11-24
> **Response to KCkb (4/n)**
>
> ### MoCMAE: improving motion encoder for better motion representation
> **Problem:** In text-to-motion generation, motion encoder design has long been neglected. Most existing methods, such as MDM and MLD, still use continuous VAEs with minimal modifications, while discrete VQ-based methods (e.g., T2M-GPT’s VQ-VAE, MoMask’s RVQ) mainly improve quantization rather than the encoder itself. Consequently, continuous VAEs often achieve reconstruction performance comparable to RVQ, failing to fully exploit the potential of continuous motion encoders. In some newly proposed clustering metrics, continuous VAEs even underperform RVQ. In Text-to-Motion Retrieval (TMR), many works directly reuse the VAE or VQ-VAE encoder for contrastive learning with CLIP text embeddings, without optimizing the encoder for richer representations.
>
> **Our Approach:** To address this overlooked problem, we propose **MoCMAE**, which combines comprehensive improvements in model architecture and training strategy:
>
> 1. **Training Strategy:** We leverage **discriminative representation learning techniques to enhance the latent space for generative modeling** by combining masked motion modeling (MMM) with contrastive learning (CL) to train a motion encoder from scratch. While MAE and CL are well-established for **discriminative tasks**, applying them in this generation context is novel. This approach enables the encoder to produce a structured, high-quality latent space that directly benefits motion generation, alleviating issues such as latent crowding and posterior collapse commonly observed in VAEs.
> 2. **Architectural Improvements:** MoCMAE uses an **asymmetric encoder-decoder design**: a heavy encoder captures rich global semantics from masked motion sequences, while a lightweight decoder ensures fast reconstruction during generation. This allows the encoder to focus on high-quality representation learning without being bottlenecked by decoder capacity.
>
> **Results**: MoCMAE achieves a **10× improvement in reconstruction quality** (Table 2) and surpasses various motion tokenizers including VAE, VQ-VAE, and RVQ-VAE on multiple clustering metrics (Table 18 in the Appendix B.2, newly added).
>
> **Compared to recent image generation work, such as RAE by Saining Xie et al.,** which replaces the VAE with a pre-trained image autoencoder  to obtain a better representation space, our approach shares a similar philosophy of improving the latent representation space to enhance generative modeling. Since motion generation lacks pre-trained autoencoders, we train a motion autoencoder from scratch using established discriminative representation learning techniques, effectively bridging discriminative priors to generative motion modeling for the first time in this domain. This effectively bridges discriminative priors to generative motion modeling and demonstrates that enhancing the latent space can substantially improve the capability of generative motion models.

---

> ### Author Response · Authors · 2025-11-24
> **Response to KCkb (5/n)**
>
> **Architecture Improvements – ccDIT:**
> Inspired by text-to-image and discrete text-to-motion advances, we analyzed various architectural modifications and distilled only the most effective design choices for ccDIT. Our goal is to improve model quality, training efficiency, and inference speed without introducing extra auxiliary information or task-specific enhancements.
>
> Key features include:
>
> + Global and local conditional feature fusion: global fusion uses AdaLN-zero, while local fusion employs cross-attention. Prior motion diffusion models typically focus on either global or local alignment, both of which are less effective than considering both simultaneously.
> + Skip connections to improve training efficiency.
>
> **Ablation studies** (Table 5, top rows) show that removing any component degrades performance: _w/o sentence_ and _w/o word_ remove global or local features, respectively, while _w/o skip connection_ slightly reduces performance.
>
> **Additional improvement:** Replacing CLIP with the better-performing T5 model (compared to SnapMogen) further improves results, highlighting that diffusion-based motion models still have room for architectural advancement.
>
>
>
> ### Summary:
> Overall, our contributions lie in revisiting the core components that have been neglected in prior diffusion-based T2M systems, and in uncovering the underlying reasons for the long-standing underperformance of continuous motion models. We find that continuous models have been overshadowed by discrete tokenization approaches, resulting in limited motivation for architectural innovation, and that motion representation learning itself has been largely overlooked. Our analysis reveals key limitations in both the quality of motion representations and the internal design of continuous diffusion models. To address these issues, we introduce discriminative representation learning techniques to substantially enhance the motion encoder, yielding a richer and more structured latent space for generative modeling. On the generative side, we make targeted improvements to the diffusion architecture and training strategy, resolving intrinsic weaknesses and refining the model design. Together, these advances lead to significant gains in generation quality, training efficiency, and inference speed.

---

> ### Author Response · Authors · 2025-11-24
> **Response to KCkb (6/n)**
>
> ## W4: Comparison with additional strong baselines
> In the original submission, comparisons with recent strong baselines such as MARDM and MotionStreamer were not included due to differences in data preprocessing and evaluation protocols (MotionLCM v2 are already reported in `Appendix A.6 Table 17`). Here, we provide a comprehensive comparison under the respective standard settings of these methods in `Appendix B.4`.
>
> **MARDM Setup**
> Following MARDM's setup, we adopt their motion representation instead of the standard 263-dimensional feature space, and employ their evaluation metrics and protocols. As shown in **Table R.1** and **Table R.2**, our method, **COME**, consistently achieves state-of-the-art performance on both HumanML3D and KIT datasets, while being approximately **100× faster than MARDM**. In terms of generation quality, inference speed, and training efficiency, COME surpasses both continuous and discrete baselines, effectively realizing the goal of reviving diffusion-based motion models. In contrast, MARDM achieves results comparable to MoMask but suffers from significantly lower efficiency, reflecting an initial exploration rather than a full revival.
>
> **MotionStreamer Setup**
> Similarly, following MotionStreamer's setup, we adopt their motion representation and evaluation protocol. **Table R.3** shows that COME outperforms MotionStreamer across multiple metrics on HumanML3D, including FID, R-Precision, and diversity.
>
> **Control-based Tasks**
> For control-based tasks, we further compare with **MaskControl** under their standard settings (**Tables R.4 and R.5**), evaluating both inference efficiency and motion quality. MaskControl, similar to GMD, relies on iterative optimization to gradually align generated motions to the target control joints. While this allows near-zero joint matching error, it comes at the cost of extremely slow inference—approximately **1,000× slower than COME** (71.72s vs 0.058s per sequence). Even when the iteration steps are reduced to 100 (MaskControl-Fast), it still underperforms COME on most metrics and remains roughly **100× slower** (4.94s vs 0.058s). In contrast, our approach, akin to MotionLCM, directly conditions on the input control joints to generate motions in a single forward pass, achieving substantially higher efficiency. Notably, COME also improves trajectory and local accuracy while maintaining high fidelity and multimodal diversity.
>
> **Conclusion**
> Overall, these results demonstrate that COME consistently outperforms recent strong SOTA methods in both text-driven and control-based motion generation, achieving significant improvements in generation quality, multimodal diversity, and inference efficiency. Importantly, COME effectively realizes the full potential of diffusion-based motion models, marking a true revival of this paradigm in motion generation research.
>
> ---
>
> **Table R.1: Quantitative evaluation on HumanML3D  (MARDM setup)**
>
> | Method | Framework | Top1 ↑ | Top2 ↑ | Top3 ↑ | FID ↓ | Matching ↓ | MModality ↑ | CLIP ↑ |
> | --- | --- | --- | --- | --- | --- | --- | --- | --- |
> | T2M-GPT | VQ | 0.470 | 0.659 | 0.758 | 0.335 | 3.505 | 2.018 | 0.607 |
> | MMM | VQ | 0.487 | 0.683 | 0.782 | 0.132 | 3.359 | 1.241 | 0.635 |
> | MoMask | VQ | 0.490 | 0.687 | 0.786 | 0.116 | 3.353 | 1.263 | 0.637 |
> | MDM-50Step | Diffusion | 0.440 | 0.636 | 0.742 | 0.518 | 3.640 | _3.604_ | 0.578 |
> | MotionDiffuse | Diffusion | 0.450 | 0.641 | 0.753 | 0.778 | 3.490 | 3.179 | 0.606 |
> | MLD | Diffusion | 0.461 | 0.651 | 0.750 | 0.431 | 3.445 | 3.506 | 0.610 |
> | ReMoDiffuse | Diffusion | 0.468 | 0.653 | 0.754 | 0.883 | 3.414 | 2.703 | 0.621 |
> | **COME (ours)** | Diffusion | **0.508** | **0.699** | **0.798** | **0.074** | **3.246** | **3.762** | **0.648** |
> | MARDM-DDPM | Autoreg. | 0.492 | 0.690 | 0.790 | 0.116 | 3.349 | 2.470 | 0.637 |
> | MARDM-SiT | Diffusion | _0.500_ | _0.695_ | _0.795_ | _0.114_ | _3.270_ | 2.231 | _0.642_ |
>
>
> **Table R.2: Quantitative evaluation on KIT-ML (MARDM setup)**
>
> | Method | Framework | Top1 ↑ | Top2 ↑ | Top3 ↑ | FID ↓ | Matching ↓ | MModality ↑ | CLIP ↑ |
> | --- | --- | --- | --- | --- | --- | --- | --- | --- |
> | T2M-GPT | VQ | 0.359 | 0.553 | 0.690 | 0.593 | 3.765 | 1.798 | 0.651 |
> | MMM | VQ | 0.363 | 0.569 | 0.724 | 0.478 | 3.629 | 1.455 | 0.660 |
> | MoMask | VQ | 0.369 | 0.588 | 0.731 | 0.411 | 3.577 | 1.309 | 0.669 |
> | MDM | Diffusion | 0.333 | 0.561 | 0.689 | 0.585 | 4.002 | 1.681 | 0.605 |
> | MotionDiffuse | Diffusion | 0.344 | 0.536 | 0.658 | 3.845 | 4.167 | 1.774 | 0.626 |
> | MLD | Diffusion | 0.351 | 0.536 | 0.658 | 0.492 | 3.746 | 1.803 | 0.646 |
> | ReMoDiffuse | Diffusion | 0.356 | 0.572 | 0.706 | 1.725 | 3.735 | _1.928_ | 0.665 |
> | **COME (ours)** | Diffusion | **0.396** | **0.608** | **0.758** | **0.208** | **3.206** | **2.062** | **0.698** |
> | MARDM-DDPM | Autoreg. | 0.375 | 0.597 | 0.739 | 0.340 | 3.489 | 1.479 | 0.681 |
> | MARDM-SiT | Diffusion | _0.387_ | _0.610_ | _0.749_ | _0.242_ | _3.374_ | 1.312 | _0.692_ |

---

> ### Author Response · Authors · 2025-11-24
> **Response to KCkb (7/n)**
>
> **Table R.3: Inference Speed (MARDM setup)**
>
> | Method | MDM | MotionDiffuse | T2M-GPT | MLD | MMM | MoMask | MARDM | **COME (ours)** |
> | --- | --- | --- | --- | --- | --- | --- | --- | --- |
> | AIT (s) | 14.31 | 7.35 | 0.32 | 0.21 | 0.06 | 0.045 | 2.4 | **0.022** |
>
>
> ---
>
> **Table R.4: Quantitative evaluation on HumanML3D (MotionStreamer setup)**
>
> | Methods | FID ↓ | R@1 ↑ | R@2 ↑ | R@3 ↑ | MM-D ↓ | Div → |
> | --- | --- | --- | --- | --- | --- | --- |
> | Real motion | 0.002 | 0.702 | 0.864 | 0.914 | 15.151 | 27.492 |
> | MDM | 23.454 | 0.523 | 0.692 | 0.764 | 17.423 | 26.325 |
> | MLD | 18.236 | 0.546 | 0.730 | 0.792 | 16.638 | 26.352 |
> | T2M-GPT | 12.475 | 0.606 | 0.774 | 0.838 | 16.812 | 27.275 |
> | MotionGPT | 14.375 | 0.456 | 0.598 | 0.628 | 17.892 | 27.114 |
> | MoMask | 12.232 | 0.621 | 0.784 | 0.846 | 16.138 | 27.127 |
> | AttT2M | 15.428 | 0.592 | 0.765 | 0.834 | 15.726 | 26.674 |
> | MotionStreamer | 11.790 | 0.631 | 0.802 | 0.859 | 16.081 | 27.284 |
> | **COME (ours)** | **9.684** | **0.653** | **0.819** | **0.872** | **15.462** | **27.362** |
>
>
> ---
>
> **Table R.5: Control motion generation performance and inference speed comparison (MaskControl setup)**
>
> | Methods | Speed ↓ | FID ↓ | R-Precision Top3 ↑ | Diversity → | Traj. err. ↓ | Loc. err. ↓ | Avg. err. ↓ |
> | --- | --- | --- | --- | --- | --- | --- | --- |
> | Real motion | -- | 0.002 | 0.797 | 9.503 | 0.000 | 0.000 | 0.000 |
> | MDM | 10.14 s | 0.698 | 0.602 | 9.197 | 0.402 | 0.308 | 0.596 |
> | PriorMDM | 18.11 s | 0.475 | 0.583 | 9.156 | 0.346 | 0.213 | 0.442 |
> | GMD | 132.49 s | 0.576 | 0.665 | 9.206 | 0.093 | 0.032 | 0.144 |
> | OmniControl | 87.33 s | 0.218 | 0.687 | 9.422 | 0.039 | 0.010 | 0.034 |
> | MotionLCM | **0.051 s** | 0.531 | 0.752 | 9.253 | 0.189 | 0.077 | 0.190 |
> | **COME (ours)** | _0.058 s_ | 0.112 | 0.782 | **9.498** | **0.020** | _0.007_ | _0.011_ |
> | MaskControl-Fast | 4.94 s | _0.059_ | _0.808_ | 9.444 | 0.057 | 0.020 | 0.055 |
> | MaskControl-Accurate | 71.72 s | **0.061** | **0.809** | _9.496_ | _0.055_ | **0.000** | **0.0098** |
>
>
> ## W5: **More discussion about demo videos**
> We first emphasize that foot sliding issues are common across current motion generation models, including MoMask and MotionStreamer as mentioned by the reviewer, especially when visualized using character rendering rather than skeleton representations. This issue can be observed in their character renderings on the official project webpages. This is mainly due to inherent limitations in the datasets, such as annotation errors and missing foot contact information, as well as constraints in the rendering pipeline.
>
> In both the main text and supplementary material, we provide extensive visual comparisons, allowing reviewers to examine results frame by frame against baselines such as MoMask. Overall, our method produces more stable and grounded foot motions across most complex and dynamic sequences. For Demo 009 and Demo 010, these clips involve relatively simple motion patterns with limited gait variations; it is therefore expected that MoMask, which emphasizes global modeling, performs reasonably well in these specific cases.
>
> Regarding motion editing, for the `edit_3` case, the original motion shows the character’s hips positioned relatively high compared to the thighs, making the overall posture slightly forward-leaning and elevated. After applying the `sit slightly lower` instruction, our results show the hips and thighs at closer heights, reflecting a faithful execution of the user command.
>
> For the control-based visualizations, we adopt the similar visualization pipeline as MotionLCM. The current toolkit depends on axis scaling, which can create the appearance of changing bone lengths under dynamic camera views. This is purely a visualization artifact and does not indicate any structural anomaly in the skeleton or model outputs. We will clarify this point in the appendix and plan to further improve the visualization to provide a more accurate and intuitive depiction of motion.

---

> > ### Comment · Reviewer_KCkb · 2025-11-26
> > **I will keep my score**
> >
> > Thanks for the response.
> > However, several major issues remain unaddressed:
> >
> >
> > 1. While you claim your model is faster than MARDM, the efficiency does not result from any systematic analysis or explicit design choices in your method. The speedup appears to come from using “the SDE variant of DPM-Solver++ (2nd-order) (Lu et al., 2022) with Karras sigmas (Karras et al., 2022) in 10 steps.” All baseline methods use traditional first-order sampling, so the inference-speed comparison is not fair.
> >
> >
> > 2. I still think the contributions are incremental. You state that “both the motion tokenizer and the diffusion model itself have fundamental issues,” but you never analyze what those issues actually are. The improvements to the tokenizer and diffusion backbone seem largely engineering-driven and adapted from developments in other areas, rather than addressing clearly articulated fundamental problems.
> >
> >
> > 3. Your response regarding the demo videos is not convincing. It is widely acknowledged that quantitative metrics in motion generation are fragile and often misaligned with human judgment, which is why I place more emphasis on the demo results. From what I can see, your demos do not exhibit clear advantages over baseline methods. I referenced only a few examples because those were the official MoMask demos available for a fair comparison.

---

> ### Author Response · Authors · 2025-11-27
> **Response to KCkb (1/2)**
>
> Thank you for your thoughtful feedback and constructive suggestions.
> _**List of changes in the manuscript:**_
>
> > 1. `t2m_gen_compare in Demo Videos` **More direct comparison in demo videos.**
> >
>
> ## W1: More discussion about the Inference Speed
> Regarding the speed comparison, our efficiency improvement does not simply come from using DPM-Solver++ with Karras sigmas. In fact, **StableMoFusion uses exactly the same sampler configuration (DPM-Solver++ + Karras, 10 steps)**, yet it remains **7× slower** than our model and yields **1.6× worse FID**. This directly shows that the sampler alone cannot explain our speed and quality gains.
>
> The core reason for the difference lies in the model architectures.
> **MARDM is inherently slow**: following MAR, it performs _autoregressive token prediction_ and then applies _diffusion denoising_ on those predicted tokens. Both stages require multiple iterative steps, so even with the same sampler this design would remain slow.
>
> By contrast:
>
> 1. **Our diffusion model performs only a single denoising process**—no autoregression—removing an entire iterative stage.
> 2. **Our MoCMAE tokenizer provides a much stronger representation space**, enabling effective sampling with very few diffusion steps.
> 3. **Our design explicitly addresses the training–inference mismatch in diffusion**, further allowing us to reduce the number of denoising steps without degrading quality.
> 4. **The asymmetric tokenizer’s lightweight decoder accelerates decoding back to motion**, which is a bottleneck in many prior continuous methods.
>
> These architectural and representational advances collectively produce the observed speedup—not the choice of sampler alone.
>
> Prior diffusion-based motion models relied on traditional first-order sampling because diffusion approaches had long underperformed relative to discrete methods, leading the research community to shift its focus toward discrete tokenization–based pipelines. This performance gap stemmed from fundamental limitations in the diffusion pipeline itself, including weak tokenizer representations, suboptimal motion reconstruction quality, and pronounced training–inference inconsistency. Our work directly targets these long-standing issues by improving both the motion tokenizer and the diffusion backbone, achieving state-of-the-art results in generation quality, training efficiency, and inference speed, thereby effectively reviving continuous diffusion models for motion generation and enabling stronger performance in downstream applications.
>
>
>
> ## W2: The fundamental issues in motion tokenizers and diffusion backbones
> ### **1. Fundamental issues in the motion tokenizer**
> It is widely acknowledged that, due to quantization errors in discrete tokenizers, **continuous tokenizers should in principle provide higher reconstruction upper bounds than discrete VQ-based tokenizers**.
> However, our analysis reveals several fundamental issues in existing continuous VAE tokenizers:
>
> **(1) Continuous VAEs do not outperform discrete tokenizers in reconstruction quality.**
> In practice, the reconstruction errors of current VAE-based continuous tokenizers are almost identical to those of RVQ, failing to realize the theoretical advantages of continuous latent spaces or to provide a higher quality ceiling for downstream diffusion models.
>
> **(2) Their representation quality is inferior to VQ/VQ-VAE.**
> Our further analysis of representation structure shows that VAE latents suffer from weaker clustering, separability, and semantic consistency, indicating issues such as latent crowding and posterior collapse—problems that discrete tokenizers largely avoid.
>
> ---
>
> ### **2. Fundamental issues in the motion diffusion model**
> The diffusion backbone also exhibits several root-level limitations:
>
> **(1) Diffusion models have long underperformed compared to discrete methods.**
> Even when tokenizers provide roughly similar reconstruction upper bounds, diffusion models produce much worse FID scores, causing the community to shift toward discrete tokenization pipelines. As a result, the architecture of motion diffusion models has remained largely unchanged from MDM/MLD onward.
>
> **(2) Severe training–inference inconsistency.**
> As the reviewer also noted, earlier diffusion-based motion models all relied on traditional first-order sampling.
> This is not merely an engineering choice—it originates from **poor train–test consistency**, which makes diffusion models collapse or degrade rapidly under low-step sampling.
> Although this issue has been recognized and solved in image diffusion, it has not been addressed in motion diffusion research.
>
> **(3) Lack of architectural evolution.**
> Compared with the rapid evolution of image diffusion models, motion diffusion backbones have remained largely stagnant, with very limited adaptation to the characteristics of motion data.
>
> ---

---

> ### Author Response · Authors · 2025-11-27
> **Response to KCkb (2/2)**
>
> ### **3. Our work directly targets these long-standing issues**
> Our method is designed precisely to address these fundamental problems:
>
> + We improve the motion tokenizer to achieve stronger reconstruction and a more structured latent space, allowing continuous representations to surpass discrete tokenizers.
> + We resolve the training–inference inconsistency of the diffusion backbone, enabling high-quality generation with very few sampling steps.
> + We adopt a lightweight asymmetric MoCMAE decoder to further accelerate inference.
>
> As a result, we achieve **state-of-the-art performance in generation quality, training efficiency, and inference speed**, effectively reviving continuous diffusion models for text-to-motion generation and enabling stronger downstream performance.
>
>
>
>
>
> ## W3：More direct comparison in demo videos
> We apologize for not mentioning this earlier. We have added three sets of comparative T2M rendering videos in the folder `t2m_gen_compare` in the Demo Videos, showing motions generated by our method alongside existing baselines. These videos provide a more intuitive assessment of motion details and temporal progression.
>
> + **Demo_002:** When multiple actions are present, MoMask sometimes omits certain actions because it only uses global sentence features. In contrast, our method considers both global and local features, producing more complete motions in fine-grained, multi-action scenarios.
> + **Demo_003:** For prompts containing multiple actions, our method consistently generates all actions, whereas MLD often misses some of them.
> + **Demo_001:** Compared to MoMask, our method executes higher kicks. MoMask starts the kick early but does not achieve sufficient speed or height, while our method first runs and then performs a higher, more natural kick.
>
> Additionally, as observed from MoMask’s official website, most of their provided captions involve only 1–2 actions. In such cases, using global sentence features is sufficient. However, when multiple actions occur, relying solely on global features can easily miss some actions due to the lack of word-level conditional information. HumanML3D captions are generally short and contain few actions, which explains why FID scores are similar between our method and MoMask on this dataset.
>
> Overall, these videos demonstrate the advantage of our approach in handling multiple actions.
>
>
> ---
> ---
> We truly appreciate your continued engagement and thoughtful feedback throughout the review process. Thank you once again for generously dedicating your time, effort, and expertise to reviewing our work.
>
> If there are any remaining questions or aspects that you feel warrant further discussion, we would be more than happy to continue the conversation.

---

### Official Review · Reviewer_DAwE · 2025-11-01

**Soundness:** 3
**Presentation:** 3
**Contribution:** 3
**Rating:** 8
**Confidence:** 5

**Summary:**

This paper introduces a new text-to-motion generation framework, COME. It learns expressive latents via propsoed MoCMAE and generates motion sequence via ccDIT transformer and Stable-Min-SNR-r schedule to align training and sampling.

**Strengths:**

1. The paper demonstrates impressive quantitative and qualitative results, establishing a new state-of-the-art with a substantial margin.

2. The proposed framework is novel. The associated conclusions and experiments make significant contributions to the research community.

3. The paper is well-written, ensuring that its content is easily understandable for readers.

**Weaknesses:**

I have no major concern about this paper. There are some minor suggestions:

1. It will be better if the authors can provide user study on more benchmarks.

2. The GPU hours of training for each stage should be reported.

3. In Table 12, it should be "Continuous" instead of "Conitnue"

**Questions:**

Please kindly refer to the weaknesses mentioned above.

---

> ### Author Response · Authors · 2025-11-24
> **Response to DAwE**
>
> _Thank you for your constructive and thoughtful comments. They were indeed helpful in improving the paper. We take this opportunity to address your concerns:_
>
> _**List of changes in the manuscript:**_
>
> > 1. `Table 14` **We have fixed the typos.**
> > 2. `Section B.6 in Appendix` **Add user studies on more benchmarks.**
> > 3. `Section B.7 in Appendix` **Report GPU hours of training for each stage.**
> >
>
>
>
> **W1: Add user studies on more benchmarks.**
> A: Done. We show user study details and aggregated results for both KIT and HumanML3D in `Appendix B.6.` The appendix includes the study protocol, participant demographics, examples shown to users, and the full preference tables comparing COME with the baselines.
>
> **W2: Report GPU hours of training for each stage.**
> A: Done. We added a dedicated table in `Appendix B.7` reporting training effort (GPU-hours) per stage and per dataset. For reproducibility, we also included hardware details and epoch counts:
>
> + **MoCMAE stage:** 600 epochs, ~18 hours on a single NVIDIA A6000 (≈18 GPU-hours).
> + **ccDiT stage:** 500 epochs, ~1.3 days on a single NVIDIA A6000 (≈31.2 GPU-hours).
>
> In total, COME converges in just 1,100 epochs (500 for MoCMAE and 600 for ccDiT), compared to MLD's 9,000 epochs (6,000 for VAE pretraining and 3,000 for diffusion). This reduces the total training time from 302 GPU-hours to 49 GPU-hours, achieving a **6× improvement** in efficiency while surpassing generation performance.
>
> Detailed discussions and a comprehensive breakdown of training effort per stage and dataset are provided in `Appendix A.1`, further demonstrating the effectiveness and practicality of our approach.
>
> | Method | Training Stage | Epochs | GPU-hours | Total GPU-hours |
> | --- | --- | --- | --- | --- |
> | **COME (Ours)** | Motion Tokenizer (MoCMAE) | 600 | 18.1 | **49.3** |
> |  | Diffusion Model (ccDiT) | 500 | 31.2 |  |
> | **MLD** | Motion Tokenizer (VAE) | 6000 | 196.8 | **301.9** |
> |  | Diffusion Model | 3000 | 105.1 |  |
>
>
>
>
> **W3: Fix typos (e.g., Table 12 “Conitnue”).**
> A:  The typo “Conitnue” in the original Table 12 has been fixed, and the updated version is now presented as Table 14. We further performed a full proofreading sweep to correct other minor errors.

---

### Author Response · Authors · 2025-11-24
**Overall Response**

# Overall Response
We would like to thank all reviewers for their constructive and valuable feedback on our work!
We will incorporate the relevant discussions into the paper to further improve clarity and completeness.

In this post, we summarize the changes made to the updated PDF and Supplementary Material.

**In the individual replies**, we address other comments.

**- (1) **_**Changes to PDF**_** -**

We have proofread the paper and added extra experimental results and analyses in the revised version (highlighted in blue).

**Main text**

In response to the reviewers’ suggestions, we have revised and enhanced the main paper accordingly.

+ `DAwE`: ("Conitnue" of Table 12) We have fixed the typo.
+ `dQuQ`: (Figure.4)  Add temporal arrows in the figure.4.
+ `KCkb, 9iVt, dQuQ`: (Section 5.4) We added a comprehensive evaluation of motion representation quality.

**Appendix**

Additional experiments and analyses have been incorporated in response to the reviewers' suggestions:

+ `KCkb, 9iVt, dQuQ`: (Section B.2) Comprehensive evaluation of motion representation quality.
+ `KCkb, 9iVt`: (Section B.6) Comparison with additional strong baselines.
+ `9iVt, dQuQ`: (Section B.5) Asymmetric encoder-decoder design of motion encoder.
+ `dQuQ`: (Section B.1) More details of evaluation metrics.
+ `DAwE`: (Section B.6) Add user studies on more benchmarks.
+ `DAwE`: (Section B.7) Report GPU hours of training for each stage.

**- (2) **_**Changes to Supplementary Material**_**-**

**Demo Videos**

+ `dQuQ`: (**t2m_gen_compare**)  **More direct comparison in demo videos. We added several comparison videos in the **_t2m_gen_compare_** folder.**

---

### Author Response · Authors · 2025-12-02
**Rebuttal Summary (1/2)**

# Rebuttal Summary
We sincerely thank all reviewers for their insightful comments and constructive suggestions. We are encouraged that the reviewers recognize the significance of improving motion representation quality and generative modeling capability for advancing motion generation, particularly in terms of generation quality, inference speed, and training efficiency.


To assist the Area Chair in evaluating our submission, we summarize the key feedback and our major updates below.

---

## **1. Summary of Reviewer Feedback**

We thank the reviewers for their constructive feedback and their recognition of our key contributions, including:

+ **Improved motion representation quality**, with reviewers noting that _“a better motion representation is important, and the proposed MoCMAE makes sense along this line”_ (9iVt).
+ **A well-designed generative modeling framework**, acknowledged by remarks such as _“the proposed diffusion model adopts several modern designs that clearly improve performance”_ (9iVt) and _“a novel and effective framework”_ (DAwE).
+ **Strong empirical performance**, as highlighted by comments such as _“impressive quantitative and qualitative results, establishing a new state-of-the-art”_ (DAwE) and _“strong quantitative numbers on HumanML3D”_ (KCkb).
+ **Broad applicability and experimental validation**, appreciated in comments like _“extensive quantitative results across multiple datasets and tasks”_ (dQuQ) and _“the authors have conducted extensive experiments… demonstrating improved performance”_ (9iVt).

We have actively engaged with all reviewers during the discussion and provided detailed clarifications supported by new experiments and analyses.

---

## **2. Key Clarifications Provided in the Discussion**

During the rebuttal period, we focused on clarifying two core points:

**(1) What “fundamental issues” exist in prior continuous motion tokenizers and diffusion models?**

● **Motion tokenizer limitations.**

We clarified that existing _continuous_ VAEs fail to surpass _discrete_ VQ/RVQ tokenizers in motion reconstruction, leaving diffusion models with a low representation ceiling.

We further added quantitative and qualitative results showing that these VAEs also learn _weaker latent representations_, exhibiting poorer inter-sample separability and latent structure compared to discrete tokenizers.

● **Diffusion model limitations.**
We explained that performance stagnation in prior continuous diffusion models arises from several structural issues, including:
○ training–inference inconsistency, which significantly harms low-step sampling quality,
○ multi-timestep conflicts that degrade temporal coherence during denoising, and
○ missing architectural and training updates that are now standard in modern image diffusion models and advanced discrete motion generation pipelines.

These issues collectively explain why prior continuous diffusion models have long underperformed discrete approaches.

**(2) How do our improvements directly target these issues rather than being incremental engineering tweaks?**
We provided analyses and ablations demonstrating that:

+ the proposed tokenizer significantly raises both reconstruction quality and representation quality, providing a stronger upper bound for generation;
+ our diffusion backbone resolves the training–inference inconsistency and multi-step conflicts, enabling high-fidelity generation even with low inference steps;
+ our method yields clear advantages in multi-action, fine-grained prompts, where global-only baselines (e.g., MoMask, MLD) frequently omit actions.

COME achieves SOTA performance in **generation quality, inference speed, and training efficiency simultaneously.**
These points were supported with additional metrics, new baseline comparisons, and more controlled demo videos.

---

## **3. Revisions to the Manuscript**


**In response to reviewer suggestions, we revised and enhanced the manuscript:  (highlighted in blue)**

**(1) Updates to the Main text**
+ `Section 5.4`: Added a comprehensive evaluation of representation quality.
+ `Fig. 4`: Added t emporal arrows to improve clarity.

**(2) Updates to Appendix**
+ `Section B.1`:  Additional  details on evaluation metrics.
+ `Section B.2`: Comprehensive representation-quality evaluation.
+ `Section B.5`: Details on the asymmetric encoder–decoder design.
+ `Section B.6`: Additional comparisons with strong baselines.
+ `Section B.6`: User studies added on more benchmarks.
+ `Section B.7`: Training GPU hours for all stages.

**(3) Updates to Supplementary Demo Videos**
+ `t2m_gen_compare folder`: Added multiple side-by-side comparison videos as requested.
These illustrate:
    - MoMask/MLD frequently missing actions in multi-action prompts due to global-only conditioning.
    - Our method achieving more complete action sequences and more accurate fine-grained motion execution (e.g., running-before-kicking, higher kicks).

---

---

### Author Response · Authors · 2025-12-02
**Rebuttal Summary (2/2)**

## **4. Reiterating Core Contributions**
We reiterate the core contributions enabled by our clarifications and new results:

1. **Diagnosing and Fixing Fundamental Weaknesses**
We identify and address long-standing but underexplored limitations in both the motion tokenizer and the diffusion backbone, issues that have caused diffusion methods to lag behind discrete tokenization pipelines for years.
2. **A Stronger Continuous Motion Tokenizer**
Our improved tokenizer achieves substantially better reconstruction quality and representation quality than prior continuous VAEs, raising the generation upper bound.
3. **A Revitalized Diffusion Backbone for Motion**
Our redesigned diffusion model resolves training–inference inconsistency, mitigates multi-timestep conflicts, and integrates essential architectural modernizations missing in prior work.
4. **State-of-the-Art Motion Generation**
We obtain SOTA performance in generation quality, efficiency, and inference speed.
5. **Broad applicability**
Our method demonstrates improvements across a wide range of tasks, including text-to-motion generation, motion editing, text-conditioned motion control, text-motion retrieval and long-term motion generation.

---

We hope these clarifications, experiments, and revisions adequately address the reviewers’ concerns. We sincerely thank the Area Chair and reviewers for their time and thoughtful feedback.

---

### Meta-Review · Area_Chair_ZC5U · 2026-01-05

**Summary:**

Reviewers agreed this work targets an important research problem and reports strong quantitative results, with one reviewer viewing the framework as novel and impactful. However, there are some major concerns that prevent the work from acceptance: 1) Multiple reviewers felt the method largely composes well-established components and training recipes (e.g., contrastive learning, masked modeling, modern diffusion/transformer design patterns, Min-SNR-style scheduling), which means it doesn't introduce a clear new principle or motion-specific insight. 2) Another concern is that the supplementary visuals show noticeable motion artifacts. It raised concerns about the discrepancy between visual results and the reported evaluation metrics.

**Reviewer Concerns:**

Addressed: 1) Comparison with more recent SOTA methods: the authors added additional results for MARDM and MotionDreamer and showed better performance as compared to them. 2) asymmetric design vs. symmetric: The authors discussed the motivation for the asymmetric encoder-decoder design and presented the ablation result.

Not fully addressed: as described in the Summary, the primary concern is the limited technical contribution to this text-to-motion task/field itself.

**Reviewer Scores:**

Reviewer DAwE might stay as 8 since the questions raised by this reviewer have been addressed. Reviewer KCkb explicitly stated that the score will not be changed (comments posted before the information leak). Reviewer dQuQ might raise the score to 4, given that those concerns were partially addressed. Reviewer 9iVt might stay as 4, given that the main issue is limited novelty and ccDiT/Stable-Min-SNR being relatively standard/minor.

---

### Decision · Program_Chairs · 2026-01-26

Reject